# Prefrontal signals precede striatal signals for biased credit assignment in motivational learning biases

Johannes Algermissen [1] ✉, Jennifer C. Swart[1], René Scheeringa[1,2], Roshan Cools [1,3] & Hanneke E. M. den Ouden [1] ✉

Actions are biased by the outcomes they can produce: Humans are more likely to show action under reward prospect, but hold back under punishment prospect. Such motivational biases derive not only from biased response selection, but also from biased learning: humans tend to attribute rewards to their own actions, but are reluctant to attribute punishments to having held back. The neural origin of these biases is unclear. Specifically, it remains open whether motivational biases arise primarily from the architecture of subcortical regions or also reflect cortical influences, the latter being typically associated with increased behavioral flexibility and control beyond stereotyped behaviors. Simultaneous EEG-fMRI allowed us to track which regions encoded biased prediction errors in which order. Biased prediction errors occurred in cortical regions (dorsal anterior and posterior cingulate cortices) before subcortical regions (striatum). These results highlight that biased learning is not a mere feature of the basal ganglia, but arises through prefrontal cortical contributions, revealing motivational biases to be a potentially flexible, sophisticated mechanism.

Human action selection is biased by potential action outcomes: reward prospect drives us to invigorate action, while threat of punishment holds us back[1–3]. These motivational biases have been evoked to explain why humans are tempted by reward-related cues signaling the chance to gain food, drugs, or money, as they elicit automatic approach behavior. Conversely, punishment-related cues suppress action and lead to paralysis, which may even lie at the core of mental health problems such as phobias and mood disorders[4,5]. While such examples highlight the potential maladaptiveness of biases in some situations, they confer benefits in other situations: Biases could provide sensible "default" actions before context-specific knowledge is acquired[1,6]. They may also provide ready-made alternatives to more demanding action selection mechanisms, especially when speed has to be prioritized[7].

Previous research has assumed that motivational biases arise because the valence of prospective outcomes influences action

selection[8]. However, we have recently shown that not only action selection, but also the updating of action values based on obtained outcomes is subject to valence-dependent biases:[3,9,10] humans are more inclined to ascribe rewards to active responses, but have problems with attributing punishments to having held back. On the one hand, such biased learning might be adaptive in combining the flexibility of instrumental learning with somewhat rigid "priors" about typical action-outcome relationships. Exploiting lifetime (or evolutionary) experience might lead to learning that is faster and more robust to environmental "noise". On the other hand, biases might be responsible for phenomena of "animal superstition" like negative automaintenance. Studies of this phenomenon used strict omission schedules in which reward were never delivered on trials on which animals showed an action (key peck, button press), but only when animals inhibited responding over a given time period. Still, animals showed

[1]Radboud University, Donders Institute for Brain, Cognition and Behaviour, Nijmegen, The Netherlands. [2]Erwin L. Hahn Institute for Magnetic Resonance Imaging, University of Duisburg-Essen, Essen, Germany. [3]Department of Psychiatry, Radboud University Medical Centre, Nijmegen, The Netherlands. ✉e-mail: johannes.algermissen@donders.ru.nl; hanneke.denouden@donders.ru.nl

continued key picking in such paradigms, which might either reflect a strong "prior belief" that any situation in which rewards were available requires active work to obtain those, or vice versa an inability to attribute rewards to having held back one's actions[1,11,12]. While reward attainment can lead to an illusory sense of control over outcomes, control is underestimated under threat of punishment: Humans find it hard to comprehend how inactions can cause negative outcomes, which makes them more lenient in judging harms caused by others' inactions[13,14]. Taken together, also credit assignment is subject to motivational biases, with enhanced credit for rewards given to actions, but diminished credit for punishments given to inactions.

While evident in behavior, the neural mechanisms subserving such biased credit assignment remain elusive. Previous fMRI studies have studied neural correlates of motivational biases in action selection at the time of cue presentation, finding that the striatal BOLD signal is dominated by the action rather than the cue valence[8,15,16]. More recently, we have reported evidence for cue valence signals in ventromedial prefrontal cortex (vmPFC) and anterior cingulate cortex (ACC), which putatively bias action selection processes in the striatum[17]. The same regions might be involved in motivational biases in learning during outcome processing, given the prominent role of the basal ganglia system not only in action selection, but also learning. Influential computational models of basal ganglia function[18,19] (henceforth called "asymmetric pathways model") predict such motivational learning biases: Positive prediction errors, elicited by rewards, lead to long-term potentiation in the striatal direct "Go" pathway (and long term depression in the indirect pathway), allowing for a particularly effective acquisition of Go responses after rewards. Conversely, negative prediction errors, elicited by punishments, lead to long term potentiation in the "NoGo" pathway, impairing the unlearning of NoGo responses after punishments. This account suggests that motivational biases arise within the same pathways involved in standard reinforcement learning (RL). An alternative candidate model is that biases arise through the modulation of these RL systems by external areas that also track past actions, putatively the prefrontal cortex (PFC). Past research has suggested that standard RL can be biased by information stored in PFC, such as explicit instructions[20,21] or cognitive map-like models of the environment[22-24]. Most notably, the ACC has been found to reflect the impact of explicit instructions[21] and of environmental changes[25,26] on prediction errors.

Both candidate models predict that BOLD signal in striatum should be better described by biased compared with "standard" prediction errors. In addition, the model proposing a prefrontal influence on striatal processing makes a notable prediction about the timing of signals: information about the selected action and the obtained outcome should be present first in prefrontal circuits to then later affect processes in the striatum. While fMRI BOLD recordings allow for unequivocal access to striatal activity, the sluggish nature of the BOLD signal prevents clear inferences about temporal precedence of signals from different regions. We thus combined BOLD with simultaneous EEG recordings which allowed us to precisely characterize learning signals in both space and time.

The key question is whether biased credit assignment arises directly from biased RL through the asymmetric pathways in the striatum, or whether striatal RL mechanisms are biased by external prefrontal sources, with the dACC as likely candidate. To this end, participants performed a motivational Go/NoGo learning task that is well-established to evoke motivational biases[3,9,27]. We expected to observe biased PEs in striatum and frontal cortical areas. By simultaneously recording fMRI and EEG and correlating trial-by-trial BOLD signal with EEG time-frequency power, we were able to time-lock the peaks of EEG-BOLD correlations for regions reflecting biased PEs and infer their relative temporal precedence. We focused on two well-established electrophysiological signatures of RL, namely theta and delta power[28-33] as well as beta power[28,34] over midfrontal electrodes.

Here, we show that motivational biases in behavior are best described by biased learning as predicted by the asymmetric cortico-striatal pathways model[18,19], which predicts better learning of actions following reward and failure to unlearn inaction following a loss. This finding provides a putative computational mechanism for how motivational action biases can arise through learning and aggravate with increased experience. We further show that BOLD signal in a range of cortical and subcortical regions is better explained by biased than by standard prediction errors. Notably, electrophysiological correlates of cortical prediction errors arise earlier than correlates of subcortical prediction errors, consistent with an influence of cortical over subcortical regions in biasing the learning of actions and inactions. Taken together, this work provides evidence for a cortico-striatal basis of biased learning of action-outcome contingencies that may drive the formation of motivational action biases.

## Results

Thirty-six participants performed a motivational Go/NoGo learning task[3,9] in which required action (Go/NoGo) and potential outcome (reward/punishment) were orthogonalized (Fig. 1A–D). They learned by trial-and-error for each of eight cues whether to perform a left button press (Go$_{LEFT}$), right button press (Go$_{RIGHT}$), or no button press (NoGo), and whether a correct action increased the chance to win a

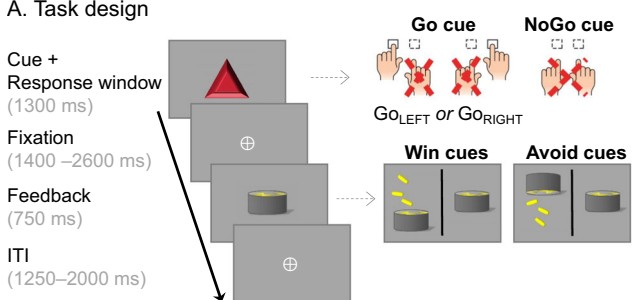

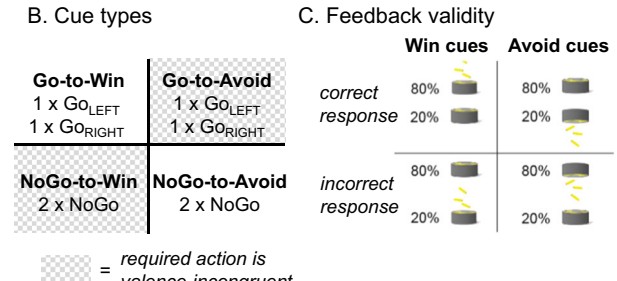

**Fig. 1 | Motivational Go/ NoGo learning task design. A** On each trial, a Win or Avoid cue appeared; valence of the cue was not signaled but should be learned. Cue offset was also the response deadline. Response-dependent feedback followed after a jittered interval. Each cue had only one correct action (Go$_{LEFT}$, Go$_{Right}$, or NoGo), which was followed by the positive outcome 80% of the time. For Win cues, actions could lead to rewards or neutral outcomes; for Avoid cues, actions could lead to neutral outcomes or punishments. Rewards and punishments were represented by money falling into/ out of a can. **B** There were eight different cues,

orthogonalizing cue valence (Win versus Avoid) and required action (Go versus NoGo). The motivationally incongruent cues (for which the motivational action tendencies were incongruent with the instrumental requirements) are highlighted in gray. **C** Feedback was probabilistic: Correct actions to Win cues led to rewards in 80% of cases, but neutral outcomes in 20% of cases. For Avoid cues, correct actions led to neutral outcomes in 80% of cases, but punishments in 20% of cases. For incorrect actions, these probabilities were reversed. Figure previously published in[17].

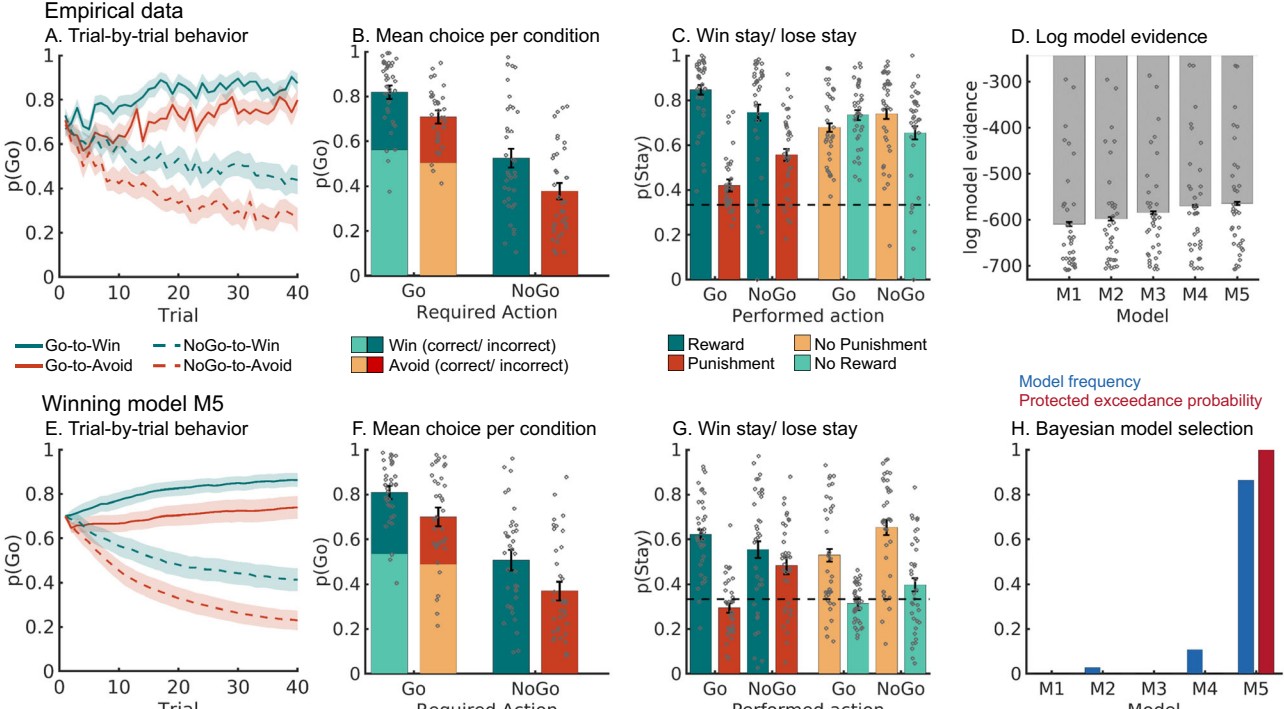

**Fig. 2 | Behavioral performance. A** Trial-by-trial proportion of Go responses (error bands are ±SEM across participants, dots indicate individual participants, $n = 36$) for Go cues (solid lines) and NoGo cues (dashed lines). The motivational bias was already present from very early trials onwards, as participants made more Go responses to Win than Avoid cues (i.e., green lines are above red lines). Additionally, participants clearly learn whether to make a Go response or not (proportion of Go responses increases for Go cues and decreases for NoGo cues). **B** Mean (error bars are ±SEM across participants, $n = 36$) proportion Go responses per cue condition (points are individual participants' means). **C** Probability to repeat a response ("stay") on the next encounter of the same cue as a function of action and outcome (error bars are ±SEM across participants, $n = 36$). Learning was reflected in higher probability of staying after positive outcomes than after negative outcomes (main

effect of outcome valence). Biased learning was evident in learning from salient outcomes, where this valence effect was stronger after Go responses than NoGo responses. Dashed line indicates chance level choice ($p_{Stay} = 0.33$). **D** Log-model evidence favors the asymmetric pathways model (M5) over simpler models (M1-M4; error bars are ±SEM across participants, $n = 36$). **E–G** Trial-by-trial proportion of Go responses, mean proportion Go responses, and probability of staying based on one-step-ahead predictions using parameters (hierarchical Bayesian inference) of the winning model (asymmetric pathways model, M5; error bars are ±SEM across simulated agents, $n = 36$). **H** Model frequency and protected exceedance probability indicate best fit for model M5 (asymmetric pathways model), in line with log model evidence. Source data are provided as a Source Data file.

reward (Win cues) or to avoid a punishment (Avoid cues). Correct actions led to 80% positive outcomes (reward, no punishment), with only 20% positive outcomes for incorrect actions. Participants performed two sessions of 320 trials with separate cue sets, which were counterbalanced across participants.

### Regression analyses of behavior

We performed regression analyses to test whether (a) responses were biased by the valence of prospective outcomes (Win/Avoid), reflecting biased responding and/or learning, and (b) whether response repetition after positive vs. negative outcomes was biased by whether a Go vs. NoGo response was performed, selectively reflecting biased learning.

For the first purpose, we analyzed choice data (Go/NoGo) using mixed-effects logistic regression that included the factors required action (Go/NoGo; note that this approach collapses across Go$_{LEFT}$ and Go$_{RIGHT}$ responses), cue valence (Win/Avoid), and their interaction (also reported in)[17]. Participants learned the task, i.e., they performed more Go responses towards Go than NoGo cues (main effect of required action: $\chi^2(1) = 32.008$, $p < 0.001$, $b = 0.815$, *95%-CI* [0.594, 1.036], two-tailed; for all reported logistic regression models, assumptions of independence of residuals, low regressor collinearity, and linearity of log odds in regressors were not violated). In contrast to previous studies[3,9], learning did not asymptote (Fig. 2A), which provided greater dynamic range for the biased learning effects to surface. Furthermore, participants showed a motivational bias, i.e., they

performed more Go responses to Win than Avoid cues (main effect of cue valence, $\chi^2(1) = 23.695$, $p < 0.001$, $b = 0.423$, *95%-CI* [0.280, 0.566], two-tailed). Replicating other studies with this task, there was no significant interaction between required action and cue valence ($\chi^2(1) = 0.196$, $p = 0.658$, $b = 0.030$, *95%-CI* [−0.103, 0.163], two-tailed; Fig. 2A, B), i.e., there was no evidence for the effect of cue valence (motivational bias) differing in size between Go or NoGo cues.

Secondly, as a proxy of (biased) learning, we analyzed cue-based response repetition (i.e., the probability of repeating a response on the next encounter of the same cue) as a function of outcome valence (positive vs negative outcome), performed action (Go vs. NoGo), and outcome salience (salient: reward or punishment vs. neutral: no reward or no punishment). As expected, participants were more likely to repeat the same response following a positive outcome (main effect of outcome valence: $\chi^2(1) = 45.595$, $p < 0.001$, $b = 0.504$, *95%-CI* [0.4000, 0.608], two-tailed). Most importantly, after salient outcomes, participants adjusted their responses to a larger degree following Go responses than NoGo responses, revealing the presence of a learning bias (Fig. 2C; interaction of valence x action x salience: $\chi^2(1) = 19.732$, $p < 0.001$, $b = 0.248$, *95%-CI* [0.154, 0.342], two-tailed). When selectively analyzing trials with salient outcomes only, rewards (compared to punishments) led to a higher proportion of choice repetitions following Go relative to NoGo responses (valence x response: $\chi^2(1) = 17.798$, $p < 0.001$, $b = 0.308$, *95%-CI* [0.183, 0.433], two-tailed; valence effect for Go only: $\chi^2(1) = 53.932$, $p < 0.001$, $b = 1.276$, *95%-CI* [1.051, 1.501], two-tailed; valence effect for NoGo only: $\chi^2(1) = 18.228$,

$p < 0.001$, $b = 0.637$, $95\%\text{-}CI$ [0.388, 0.886], two-tailed; see full results in Supplementary Table 1).

Taken together, these results suggested that behavioral adaptation following rewards and punishments was biased by the type of action that led to this outcome (Go or NoGo). However, this analysis only considered behavioral adaptation on the next trial, and could not pinpoint the precise algorithmic nature of this learning bias. More importantly, it did not provide trial-by-trial estimates of action values as required for model-based fMRI and EEG analyses to test for regions or time points that reflected biased learning. We thus analyzed the impact of past outcomes on participants' choices using computational RL models.

## Computational modeling of behavior

In line with previous work[3,9], we fitted a series of increasingly complex RL models. We started with a simple Q-learning model featuring learning rate and feedback sensitivity parameters (M1). We next added a Go bias, capturing participants' overall propensity to make Go responses (M2), and a Pavlovian response bias (M3), reflecting participants' propensity to adjust their likelihood of emitting a Go response in response to Win vs. Avoid cues[3]. Alternatively, we added a learning bias (M4), amplifying the learning rate after rewarded Go responses and dampening it after punished NoGo responses[3], in line with the asymmetric pathways model. In the final model (M5), we added both the response bias and the learning bias. For the full model space (M1-M5) and model definitions, see the Methods section.

Model comparison showed clear evidence in favor of the full asymmetric pathways model featuring both response and learning biases (M5; model frequency: 86.43%, protected exceedance probability: 100%, see Fig. 2D, H; for model parameters and fit indices, see Supplementary Table 2; for parameter recovery analyses, see Supplementary Note 6 with Supplementary Fig. 5). Posterior predictive checks involving one-step-ahead predictions and model simulations showed that this model captured key behavioral features (Fig. 2E, F), including motivational biases and a greater behavioral adaptation after Go responses followed by salient outcomes than after NoGo responses followed by salient outcomes (Fig. 2G). This pattern could not be captured by an alternative learning bias model based on the idea that active responses generally enhance credit assignment[35] (Supplementary Note 7 with Supplementary Fig. 6).

One feature of the behavioral data that was not well captured by the asymmetric pathways model was a high tendency of participants to repeat responses ("stay") to the same cue irrespective of outcomes (see Fig. 2C, G). This tendency was stronger for Win than Avoid cues. We explored three additional models featuring supplementary mechanisms to account for this behavioral pattern (Supplementary Note 8 with Supplementary Fig. 7). All these models fitted the data well and captured the propensity of staying better than M5; however, these models overestimated the proportion of incorrect Go responses. Model-based fMRI analyses based on these models led to results largely identical to those obtained with M5 (Supplementary Note 9 with Supplementary Fig. 8). We thus focused on M5, which relied on only a single mechanism (i.e., biased learning from rewarded Go and punishment NoGo actions).

## fMRI: Basic quality control analyses

First, we performed a GLM as a quality-check to test which regions encoded positive (rewards, no punishments) vs. negative (no reward/ punishment) outcomes in a "model-free" way, independent of any model-based measure derived from a RL model (for full description of the GLM regressors and contrasts, see Supplementary Table 4). Positive outcomes elicited a higher BOLD response in regions including vmPFC, ventral striatum, and right hippocampus, while negative outcomes elicited higher BOLD in bilateral dorsolateral PFC (dlPFC), left

ventrolateral PFC, and precuneus (Fig. 3A, see full report of significant clusters in Supplementary Table 6).

We also assessed which regions encoded Go vs. NoGo as well as $Go_{LEFT}$ vs. $Go_{RIGHT}$ responses. There was higher BOLD for Go than NoGo responses at the time of response in dorsal ACC (dACC), striatum, thalamus, motor cortices, and cerebellum, while BOLD was higher for NoGo than Go responses in right IFG (see below; Supplementary Table 6)[17]. For lateralized Go responses, there was higher BOLD signal in contralateral motor cortex and operculum as well as ipsilateral cerebellum when contrasting hand responses against each other (see below). These results are in line with previous results on outcome processing and response selection and thus assure the general data quality.

## fMRI: Biased learning in prefrontal cortex and striatum

To test which brain regions were involved in biased learning, we performed a model-based GLM featuring the trial-by-trial PE update as a parametric regressor (see GLM notation in Supplementary Table 3). We used the group-level parameters of the best fitting computational model (M5) to compute trial-by-trial belief updates (i.e., prediction error * learning rate) for every trial for every participant. In assessing neural signatures of biased learning, we faced the complication that standard (Q-learning in M1) and biased PEs (winning model M5) were highly correlated. A mean correlation of 0.92 across participants (range 0.88–0.95) made it difficult to neurally distinguish biased from standard learning. To circumvent this collinearity problem, we decomposed the biased PE (computed using model M5) into the standard PE (computed using model M1) plus a difference term:[22,36] $PE_{BIAS} = PE_{STD} + PE_{DIF}$. A neural signature of biased learning should significantly—and with the same sign—encode both components of this biased PE term. Standard PEs and the difference term were uncorrelated (mean correlation of −0.02 across participants; range −0.33–0.24; see Supplementary Note 10 with Supplementary Figs. 9 and 10 for a graphical illustration of this procedure). We tested for biased prediction errors $PE_{BIAS}$ by testing which regions significantly encoded the conjunction of both its components, i.e., the significant encoding of both $PE_{STD}$ and $PE_{DIF}$. Dissociating two alternative learning signals by decomposing one into the other plus a difference term is an established procedure to disentangle the contributions of two highly correlated signals[22,36]. It has an effect highly similar to orthogonalizing regressors[37] while maintaining interpretability in that both regressors ($PE_{STD}$ and $PE_{DIF}$) add up to the term of interest ($PE_{BIAS}$). Significant encoding of both components (with the same sign) provides strong evidence for encoding of biased prediction errors $PE_{BIAS}$. The $PE_{DIF}$ term itself has no substantive neural interpretation; it is merely an implicit model comparison of a null model ($PE_{STD}$) against a full model ($PE_{BIAS}$). Intuitively, for voxels for which both $PE_{STD}$ and $PE_{DIF}$ are significant, one can conclude that the BOLD signal correlates with the full biased prediction error term $PE_{BIAS}$, and that this correlation is significantly stronger than for the baseline prediction error term $PE_{STD}$.

While $PE_{STD}$ was encoded in a range of cortical and subcortical regions (Fig. 3B) previously reported in the literature[38], significant encoding of both $PE_{STD}$ and $PE_{DIF}$ (conjunction) occurred in striatum (caudate, nucleus accumbens), dACC (area 23/24), perigenual ACC (pgACC; area 32d bordering posterior vmPFC), posterior cingulate cortex (PCC), left motor cortex, left inferior temporal gyrus, and early visual regions (Fig. 3C; see full report of significant clusters in Supplementary Table 5). Thus, BOLD signal in these regions was better described (i.e., more variance explained) by biased learning than by standard prediction error learning.

## EEG: Biased learning in midfrontal delta, theta, and beta power

Similar to the fMRI analyses, we next tested whether midfrontal power encoded biased PEs rather than standard PEs. While fMRI provides

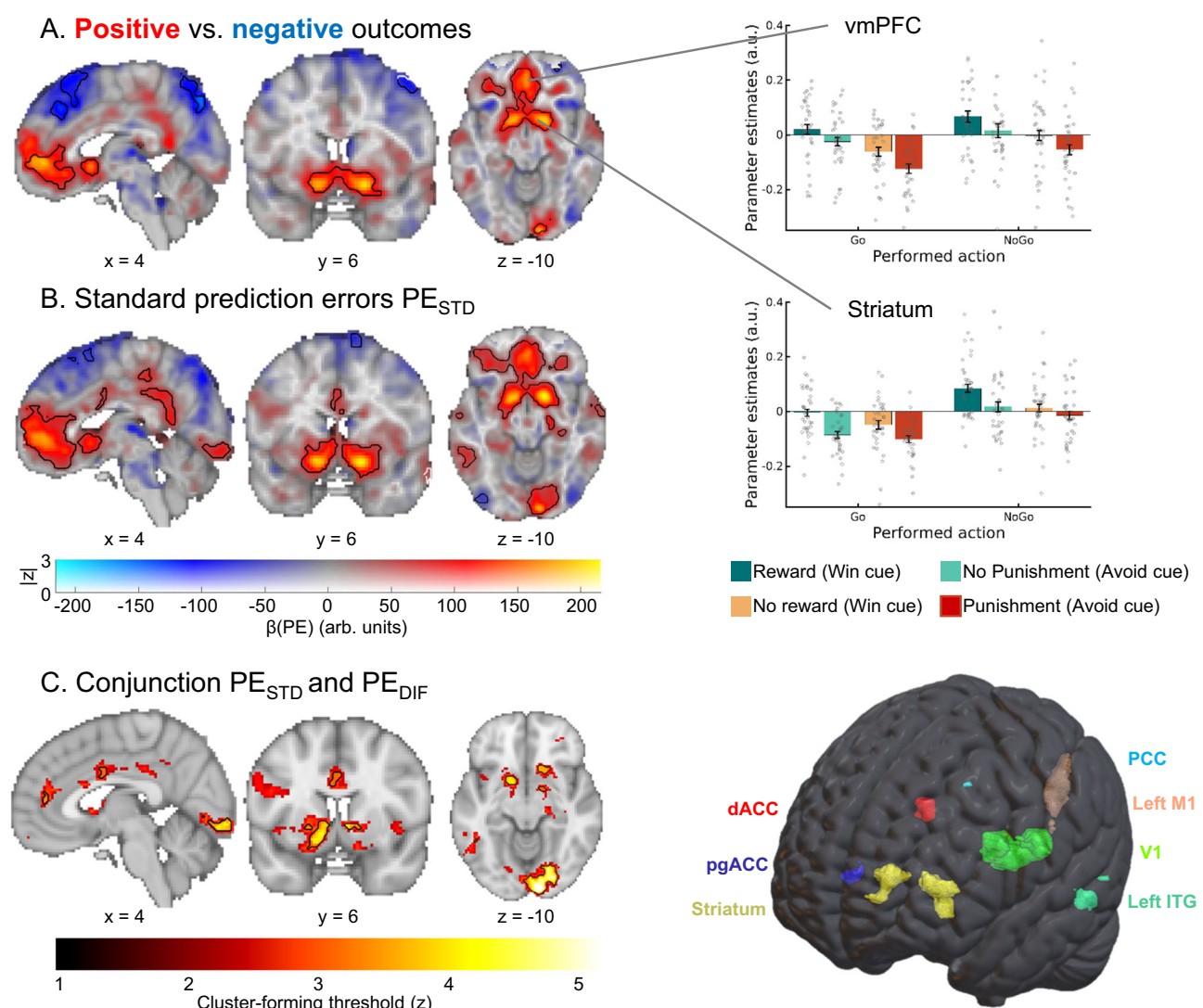

**Fig. 3 | BOLD signal reflecting outcome processing.** BOLD effects displayed using a dual-coding visualization: color indicates the parameter estimates and opacity the associated z-statistics. Significant clusters are surrounded by black edges. **A** Significantly higher BOLD signal for positive outcomes (rewards, no punishments) compared with negative outcomes (no rewards, punishments) was present in a range of regions including bilateral ventral striatum and vmPFC. Bar plots show mean parameter estimates per condition (error bars are ±SEM across participants; dots indicate individual participants, $n = 34$). Source data are provided as a Source Data file. **B** BOLD signals correlated positively to "standard" RL prediction errors in several regions, including the ventral striatum, dACC, vmPFC, and PCC. **C** Left panel: Regions encoding both the standard PE term and the difference term to biased PEs (conjunction) at different cluster-forming thresholds ($1 < z < 5$, color coding; opacity constant). Clusters significant at a threshold of $z > 3.1$ are surrounded by black edges. In bilateral striatum, dACC, pgACC, PCC, left motor cortex, left inferior temporal gyrus, and primary visual cortex, BOLD was significantly better explained by biased learning than by standard learning. Right panel: 3D representation with all seven regions encoding biased learning (and used in fMRI-informed EEG analyses).

spatial specificity of where PEs are encoded, EEG power provides temporal specificity of when signals encoding prediction errors occur[29,34]. In line with our fMRI analysis, we used the standard PE term $PE_{STD}$ and the difference to the biased PE term $PE_{DIF}$ as trial-by-trial regressors for EEG power at each channel-time-frequency bin for each participant and then performed cluster-based permutation tests across the b-maps of all participants. Note that differently from BOLD signal, EEG signatures of learning typically do not encode the full prediction error. Instead, PE valence (better vs. worse than expected) and PE magnitude (saliency, surprise) have been found encoded in the theta and delta band, respectively, but with opposite signs[31-33]. When testing for parametric correlates of PE magnitude, we therefore controlled for PE valence, thereby effectively testing for correlations with the absolute PE magnitude (i.e., degree of surprise). Note that PE valence was identical for standard and biased PEs. Thus, only PE magnitude could distinguish both learning models.

Both midfrontal theta and beta power reflected outcome (PE) valence: Theta power was higher for negative (non-reward and punishment) than for positive (reward and non-punishment) outcomes (225–475 ms, $p = 0.006$, two-tailed; Fig. 4A, B), while beta power was higher for positive than for negative outcomes (300–1,250 ms, $p = .002$, two-tailed; Fig. 4A, C). Differences in theta power were clearly strongest over frontal channels, while differences in the beta range were more diffuse, spreading over frontal and parietal channels (Fig. 4B, C). All results held when the condition-wise ERP was removed from the data (see Supplementary Note 12 with Supplementary Fig. 13), suggesting that differences between conditions were due to induced (rather than evoked) activity (for results in the time domain, see Supplementary Note 13 with Supplementary Figs. 14 and 15).

When testing for correlates of PE magnitude, we controlled for PE valence given that previous studies have reported TF correlates of both PE valence and PE magnitude in a similar time and frequency

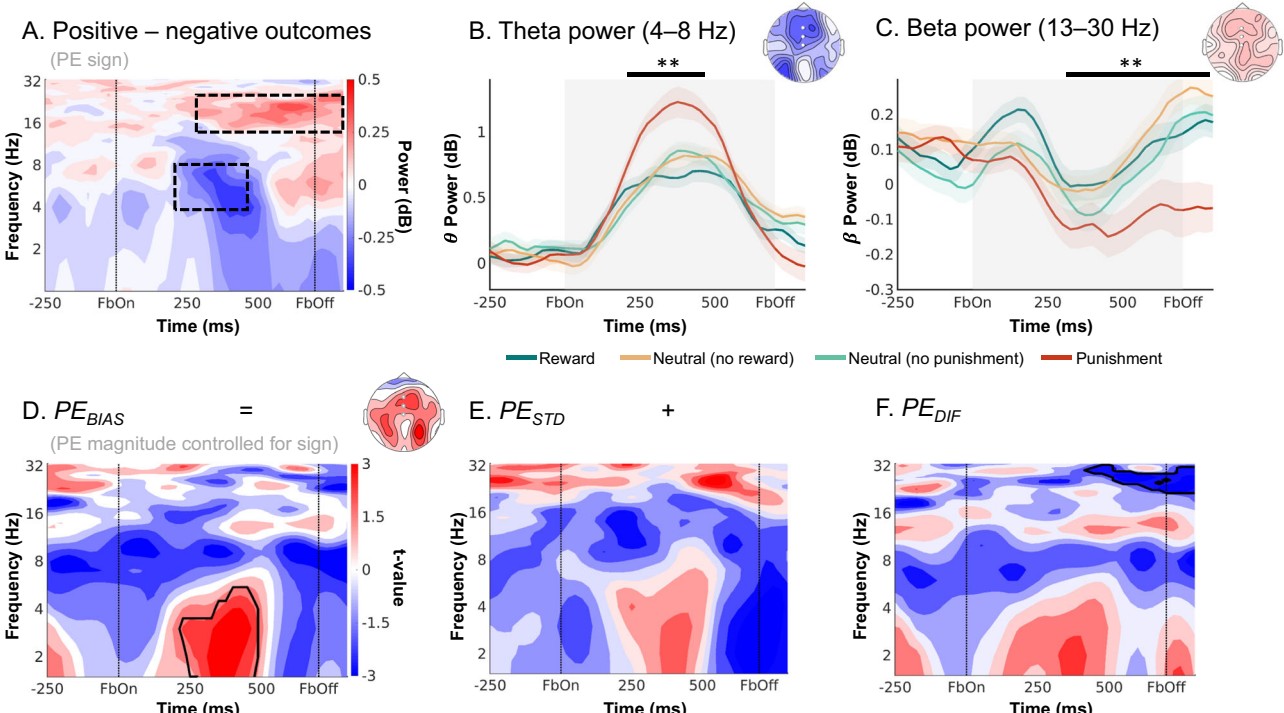

**Fig. 4 | EEG time-frequency power over midfrontal electrodes (Fz/ FCz/ Cz) reflecting outcome processing. A** Time-frequency plot (logarithmic y-axis) displaying higher theta (4–8 Hz) power for negative (non-reward for Win cues and punishment for Avoid cues) outcomes and higher beta power (16–32 Hz) for positive (reward and non-punishment) outcomes. This contrast reflects EEG correlates of PE valence (better vs. worse than expected). Black square dot boxes indicate clusters above threshold that drive significance in a-priori defined frequency ranges. **B** Theta power transiently increases for any outcome, but more so for negative outcomes (especially punishments) around 225–475 ms ($p = 0.006$, two-tailed) after feedback onset (error bands are ±SEM across participants, $n = 32$). Black horizontal lines indicate the time range for which the cluster driving significance was above threshold. **C** Beta power was higher for positive than negative

outcomes over a long time period around 300–1250 ms ($p = 0.002$, two-tailed) after feedback onset (error bands are ±SEM across participants, $n = 32$). **D–F** Correlations between midfrontal EEG power and model-based trial-by-trial PE magnitudes controlling for PE valence (thus effectively testing for correlates of "absolute" PEs). **D** displays the correlates of biased prediction errors $PE_{BIAS}$, which are decomposed into (**E**) $PE_{STD}$ based on the non-biased learning model M1, and (**F**) their difference $PE_{DIF}$. Solid black lines indicate clusters above threshold. Biased PEs were significantly positively correlated with midfrontal delta power (**D**). The correlations of delta with the standard PEs (**E**) and the difference term to biased PEs (**F**) were positive as well, though not significant. Beta power only significantly encoded the difference term to biased PEs (**F**). **$p < 0.01$, cluster-based permutation test. Source data are provided as a Source Data file.

range, but with opposite signs[31–33]. Midfrontal delta power was indeed positively correlated with the $PE_{BIAS}$ term (225–475 ms; $p = 0.017$, two-tailed; Fig. 4D). Decomposition of the $PE_{BIAS}$ term into its constituent terms showed that this correlation was not significant for the $PE_{STD}$ term ($p = 0.074$, two-tailed, Fig. 4E) nor for the $PE_{DIF}$ term ($p = 0.185$, two-tailed; Fig. 4F). This result does not imply that the $PE_{BIAS}$ term explained delta power significantly better than the $PE_{STD}$ term; it only implies significant encoding of the $PE_{BIAS}$ term as suggested by the model that best fitted the behavioral data, with no significant evidence for a similar encoding of the conventional $PE_{STD}$ term. For a similar observation in the time-domain EEG signal, see Supplementary Note 14 with Supplementary Fig. 16. Beyond delta power, beta power correlated positively, though not significantly with $PE_{STD}$ ($p = 0.110$, two-tailed, Fig. 4E) and significantly negatively with $PE_{DIF}$ ($p = 0.001$, two-tailed, 425–850 ms). Given these oppositely-signed correlations of its constituents, the $PE_{BIAS}$ term did not significantly correlate with beta power ($p = 0.550$, two-tailed, Fig. 4D).

In sum, both midfrontal theta power (negatively) and beta power (positively) encoded PE valence. In addition, delta power encoded PE magnitude (positively). This encoding was only significant for biased PEs, but not standard PEs. Taken together, as was the case for BOLD signal, midfrontal EEG power also reflected biased learning. As a next step, we tested whether the identified EEG phenomena were correlated with trial-by-trial BOLD signal in identified regions. Crucially, this allowed us to test whether EEG correlates of cortical learning precede EEG correlates of subcortical learning.

## Combined EEG-fMRI: prefrontal cortex signals precede striatum during biased outcome processing

The observation that also cortical areas (dACC, pgACC, PCC) show biased PEs is consistent with the "external model" of cortical signals biasing learning processes in the striatum. However, this model makes the crucial prediction that these biased learning signals should be present first in cortical areas and only later in the striatum. Next, we used trial-by-trial BOLD signal from those regions encoding biased PE to predict midfrontal EEG power. By determining the time points at which different regions correlated with EEG power, we were able to infer the relative order of biased PE processing across cortical and subcortical regions, revealing whether cortical processing preceded striatal processing. We used trial-by-trial BOLD signal from the seven regions encoding biased PEs, i.e., striatum, dACC, pgACC, PCC, left motor cortex, left ITG, and primary visual cortex (see masks in Supplementary Note 11 with Supplementary Figs. 11 and 12) as regressors on average EEG power over midfrontal electrodes (Fz/ FCz/ Cz; see Supplementary Note 15 with Supplementary Fig. 17 for a graphical illustration of this approach). We performed analyses with and without PEs included in the model, which yielded identical results and suggested that EEG-fMRI correlations did not merely result from PE processing as a "common cause" driving signals in both modalities. Instead, EEG-fMRI correlations reflected incremental variance explained in EEG power by the BOLD signal in selected regions (even beyond variance explained by the model-based PE estimates), providing the strongest test for the hypothesis that BOLD and EEG signal

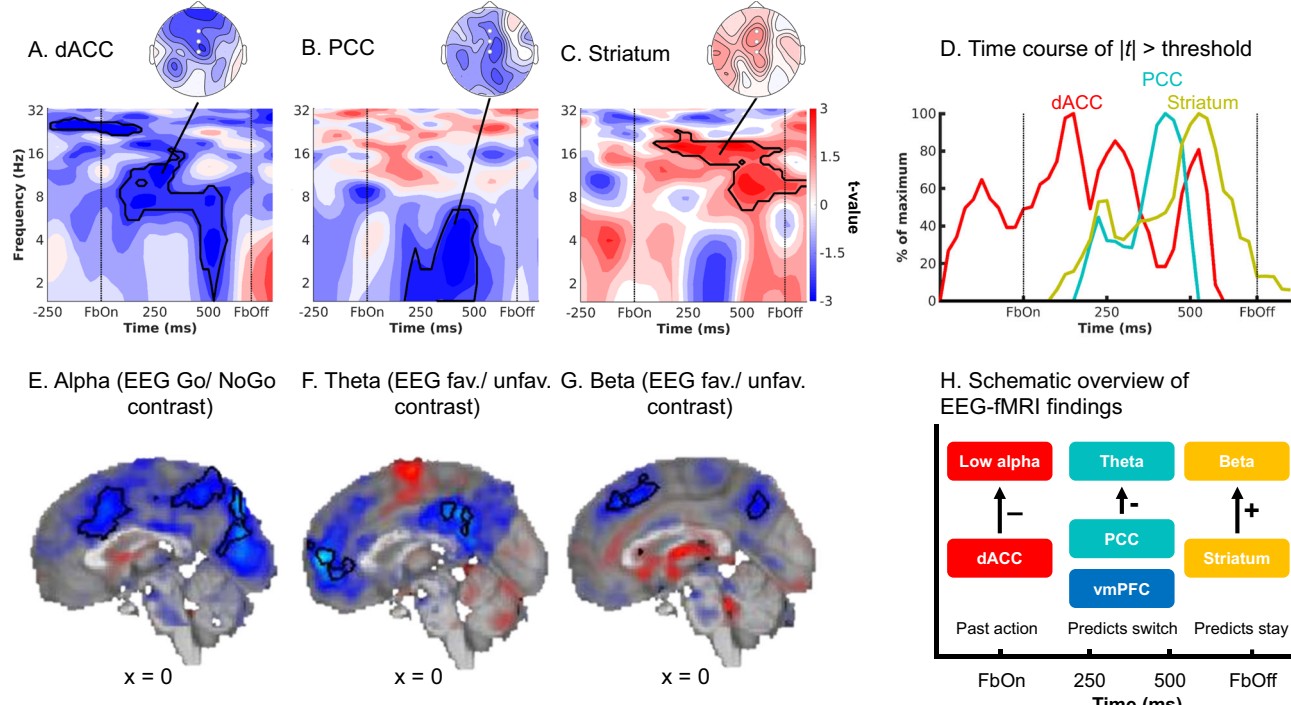

**Fig. 5 | fMRI-informed EEG analyses.** Unique temporal contributions of BOLD signal in (**A**) dACC, (**B**) PCC, and (**C**) striatum to average EEG power over midfrontal electrodes (Fz/FCz/Cz). Group-level $t$-maps display the modulation of the EEG power by trial-by-trial BOLD signal in the selected ROIs. dACC BOLD correlated negatively with early alpha/ theta power 100–575 ms, 2–17 Hz, $p = 0.016$, two-tailed, PCC BOLD negatively with theta/ delta power (175–500 ms, 1–6 Hz, $p = 0.014$, two-tailed), and striatal BOLD positively with beta/ alpha power (100–800 ms, 7–23 Hz, $p = 0.010$, two-tailed). Areas surrounded by a black edge indicate clusters of $|t| > 2$ with $p < 0.05$ (cluster-corrected). Topoplots indicate the topography of the respective cluster. Source data are provided as a Source Data file. **D** Time course of dACC, PCC, and striatal BOLD correlations, normalized to the peak of the time course of each region. dACC-lower alpha band correlations emerged first, followed by (negative) PCC-theta correlations and finally positive striatum-beta correlations. The reverse approach using lower alpha (**E**), theta (**F**) and beta (**G**) power as trial-by-trial regressors in fMRI GLMs corroborated the fMRI-informed EEG analyses: Lower alpha band power correlated negatively with the dACC BOLD, theta power negatively with vmPFC and PCC BOLD, and beta power positively with striatal BOLD. **H** Schematic overview of the main EEG-fMRI results: dACC encoded the previously performed response and correlated with early midfrontal lower alpha band power. vmPFC/ PCC (correlated with theta power) and striatum (correlated with beta power) both encoded outcome valence, but had opposite effects on subsequent behavior. Note that activity in these regions temporally overlaps; boxes are ordered in temporal precedence of peak activity.

reflect the same neural phenomenon. As the timeseries of all seven regions were included in one single regression, their regression weights reflected each region's unique contribution, controlling for any shared variance. In line with the "external model", BOLD signal from prefrontal cortical regions correlated with midfrontal EEG power earlier after outcome onset than did striatal BOLD signal:

First, dACC BOLD was significantly negatively correlated with alpha/ theta power early after outcome onset (100–575 ms, 2–17 Hz, $p = 0.016$, two-tailed; Fig. 5A). This cluster started in the alpha/ theta range and then spread into the theta/delta range (henceforth called "lower alpha band power"). It was not observed in the EEG-only analyses reported above.

Second, while pgACC BOLD did not correlate significantly with midfrontal EEG power ($p = 0.184$, two-tailed), BOLD in PCC was negatively correlated with theta/ delta power (Fig. 5B; 175–500 ms, 1–6 Hz, $p = 0.014$, two-tailed). This finding bore resemblance in terms of time-frequency space to the cluster of (negative) PE valence encoding in the theta band and (positive) PE magnitude encoding in the delta band identified in the EEG-only analyses (Fig. 4A). Complementary to the fMRI-informed EEG analyses, we also performed independent EEG-informed fMRI analyses, which showed the robustness of this EEG-fMRI correlation. We used the trial-by-trial EEG signal in the cluster identified in the EEG-only analyses (see Fig. 4A, B) to predict BOLD signal across the brain (see Supplementary Note 15 with Supplementary Fig. 18 for a graphical illustration of this approach). The EEG time-frequency-mask used to create the EEG regressor was defined based on

the EEG-only analyses (Fig. 4A, B) and thus blind to the result of the fMRI-informed EEG analysis. We observed significant clusters of negative EEG-BOLD correlation in vmPFC and PCC (Fig. 5F; Supplementary Table 7). We thus discuss vmPFC and PCC together in the following.

Third, there was a significant positive correlation between striatal BOLD and midfrontal beta/ alpha power (driven by a cluster at 100–800 ms, 7–23 Hz, $p = 0.010$, two-tailed; Fig. 5C). This finding bore resemblance in time-frequency space to the cluster of positive PE valence encoding in beta power identified in the EEG-only analyses (Fig. 4A, C), but extended into the alpha range. Again, to support the robustness of this finding, we used trial-by-trial midfrontal beta power in the cluster identified in the EEG-only analyses (see Fig. 4A, C) to predict BOLD signal across the brain. Clusters of positive EEG-BOLD correlations in right dorsal caudate (and left parahippocampal gyrus) as well as clusters of negative correlations in bilateral dorsolateral PFC (dlPFC) and supramarginal gyrus (SMG; Fig. 5G; Supplementary Table 7) confirmed the positive striatal BOLD-beta power association. Given that the striatum is far away from the scalp and thus unlikely to be the source of midfrontal beta power over the scalp, and given the assumption that trial-by-trial variation in an oscillatory signal should correlate with BOLD signal in its source[39,40], we speculate that dlPFC and SMG (identified in the EEG-informed fMRI analyses) are the sources of beta power over the scalp and act as an "antenna" for striatal signals. In line with this idea, previous studies have localized feedback-related beta power

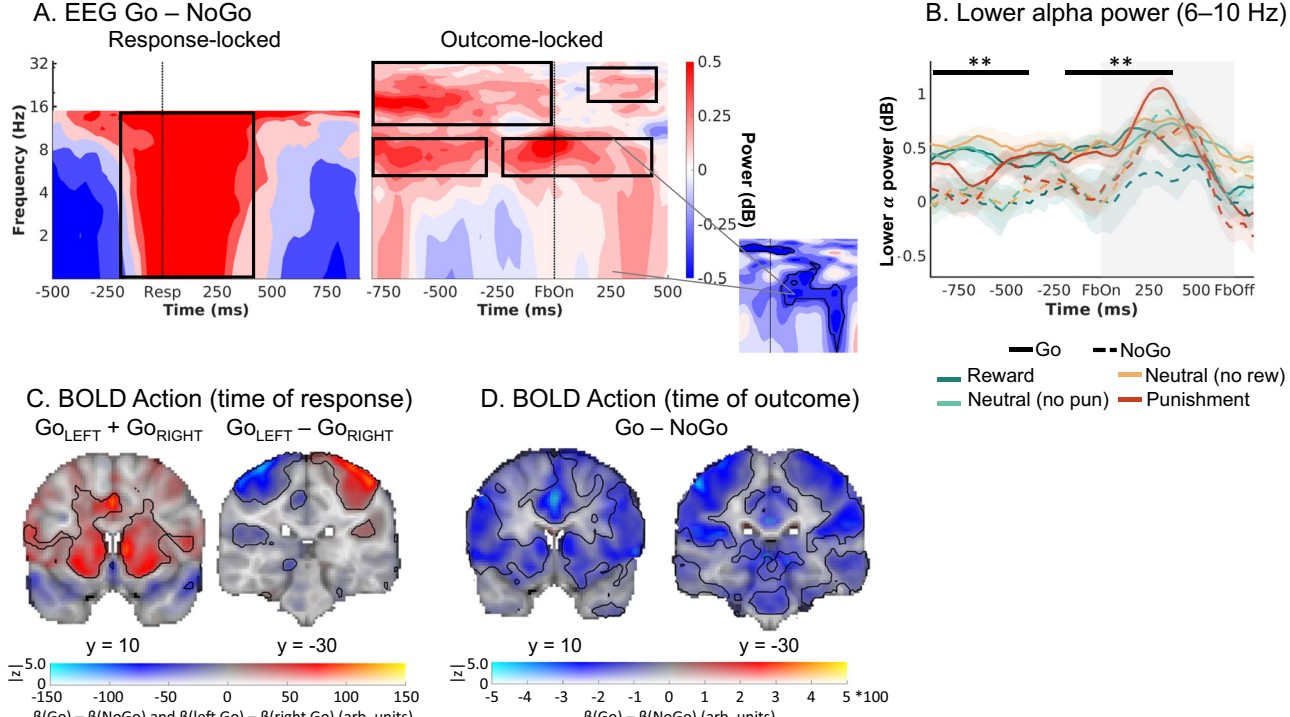

**Fig. 6 | Exploratory follow-up analyses on dACC BOLD signal and midfrontal lower alpha band power. A** Midfrontal time-frequency response-locked (left panel) and outcome-locked (right panel). Before and shortly after outcome onset, power in the lower alpha band was higher on trials with Go actions than on trials with NoGo actions. The shape of this difference resembles the shape of dACC BOLD-EEG TF correlations (small plot; note that this plot depicts BOLD-EEG correlations, which were negative). Note that differences between Go and NoGo trials occurred already before outcome onset in the alpha and beta range, reminiscent of delay activity, but were not fully sustained throughout the delay between response and outcome. **B** Midfrontal power in the lower alpha band per action x outcome condition (error bands are ±SEM across participants, *n* = 32). Lower alpha band power was consistently higher on trials with Go actions than on trials with NoGo actions, starting already before outcome onset. Source data are provided as a Source Data file. **C** BOLD signal differences between Go and NoGo actions (activation by either left or right Go actions compared to the implicit baseline in the GLM, which contains the NoGo actions; left panel) and left vs. right hand responses (right panel) at the time or responses. Response-locked dACC BOLD signal was significantly higher for Go than NoGo actions. **D** BOLD signal differences between Go and NoGo actions at the time of outcomes. Outcome-locked dACC BOLD signal (and BOLD signal in other parts of cortex) was significantly lower on trials with Go than on trials with NoGo actions. **p* < 0.01, cluster-based permutation test.

in lateral frontal and parietal regions, both using simultaneous EEG-fMRI[41–43] and source-localization[44,45].

Finally, regarding the other three regions that showed a significant BOLD signature of biased PEs, BOLD in left motor cortex was significantly negatively correlated with midfrontal beta power (*p* = 0.002, two-tailed; around 0–625 ms; Supplementary Note 16 with Supplementary Fig. 19). There were no significant correlations between midfrontal EEG power and left inferior temporal gyrus or primary visual cortex BOLD (Supplementary Fig. 19). All results were robust to different analysis approaches including shorter trial windows, different GLM specifications, inclusion of task-condition and fMRI motion realignment regressors, and individual modeling of each region. TF results were not reducible to phenomena in the time domain (Supplementary Note 17 with Supplementary Fig. 20).

In sum, there were negative correlations between dACC BOLD and midfrontal lower alpha band power early after outcome onset, negative correlations between PCC BOLD and midfrontal theta/ delta power at intermediate time points, and positive correlations between striatal BOLD and midfrontal beta power at late time points. This temporal dissociation was especially clear in the time courses of the test statistics for each region (thresholded at |t| > 2 and summed across frequencies), for which the peaks of the cortical regions preceded the peak of the striatum (Fig. 5D, H). Note that time-frequency power is estimated over temporally extended windows (400 ms in our case), which renders any interpretation of the "onset" or "offset" of such correlations more difficult. In sum, these results are consistent with an "external model" of motivational biases arising from early cortical processes biasing later learning processes in the striatum.

### dACC BOLD and midfrontal lower alpha band power encode the previously performed action during outcome presentation
While the clusters of EEG-fMRI correlation in the theta/delta and beta range matched the clusters identified in EEG-only analyses, the cluster of negative correlations between dACC BOLD and early midfrontal lower alpha band power was novel and did not match our expectations. Given that these correlations arose very soon after outcome onset, we hypothesized that dACC BOLD and midfrontal lower alpha band power might reflect a process occurring even before outcome onset, such as the maintenance ("memory trace") of the previously performed response to which credit may later be assigned. We therefore assessed whether information of the previous response was present in dACC BOLD and in the lower alpha band around the time of outcome onset.

First, we tested for BOLD correlates of the previous response at the time of *outcomes* (eight outcome-locked regressors for every Go/NoGo x reward/no reward/no punishment/punishment combination) while controlling for motor-related signals at the time of the *response* (response-locked regressors for left-hand and right-hand button presses). As previously mentioned, at the time of responses, BOLD was higher for Go compared to NoGo responses in several regions including dACC and striatum (Fig. 6C left panel), and BOLD in contralateral motor cortex and operculum was higher for Go responses of

one hand compared to Go responses of the other hand (Fig. 6C right panel). In contrast, at the time of outcomes, there was higher BOLD signal for NoGo than Go responses across several cortical and sub-cortical regions, peaking in both the dACC and striatum (Fig. 6D). This inversion of effects−higher BOLD for Go than NoGo responses at the time of response (see quality checks), but the reverse at the time of outcome−was also observed in the upsampled raw BOLD and was independent of the response of the next trial (Supplementary Note 18 with Supplementary Fig. 21). In sum, large parts of cortex, including the dACC, encoded the previously performed response at the moment outcomes were presented, in line with the idea that the dACC maintains a "memory trace" of the previously performed response.

Second, we tested for differences between Go and NoGo responses at the time of outcomes in midfrontal broadband EEG power. Power was significantly higher on trials with Go than on trials with NoGo responses, driven by clusters in the lower alpha band (spreading into the theta band; around 0.000–0.425 s, 1–11 Hz, $p = 0.012$, two-tailed) and in the beta band (around 0.200–0.450 s, 18–27 Hz, $p = 0.022$, two-tailed; Fig. 6A, B). The first cluster matched the time-frequency pattern of dACC BOLD-alpha power correlations (Fig. 5A).

If this activity cluster contained a signature of the previously performed response, it might have been present throughout the delay between cue offset and outcome onset. When repeating the above permutation test including the last second before outcome onset, there were significant differences again, driven by a sustained cluster in the beta band ($-1$–$0$ s, 13–33 Hz, $p = 0.002$, two-tailed) and two clusters in the alpha/ theta band (Cluster 1: $-1.000$– $-0.275$ s, 1–10 Hz, $p = 0.014$, two-tailed; Cluster 2: $-0.225$–0.425 s, 1–11 Hz, $p = 0.022$, two-tailed; Fig. 6B). These findings suggest that lower alpha band power might reflect a sustained memory of the previously performed response. Additional analyses (Supplementary Note 18 with Supplementary Fig. 21) yielded that this Go-NoGo trace during outcome processing did not change over the time course of the experiment, suggesting that it did not reflect typical fatigue/ time-on task effects often observed in the alpha band.

Again, we performed the reverse EEG-fMRI analysis using trial-by-trial power in the identified lower alpha band cluster (Fig. 6B) as an additional regressor in the quality-check fMRI GLM. Clusters of negative EEG-BOLD occurred correlation in a range of cortical regions, including dACC and precuneus (Fig. 5E; Supplementary Table 7). In sum, both dACC BOLD signal and midfrontal lower alpha band power contained information about the previously performed response, consistent with the idea that both signals reflect a "memory trace" of the response to which credit is assigned once an outcome is obtained.

### Striatal and vmPFC/PCC BOLD differentially relate to action policy updating

EEG correlates of PCC BOLD and striatal BOLD occurred later than for the dACC BOLD and overlapped with classical feedback-related midfrontal theta and beta power responses. We hypothesized that those neural signals might be more closely related to the updating of action policies (i.e., which action to perform for each cue) and predict the next response to the same cue[30,46]. We thus used the trial-by-trial BOLD responses in dACC, PCC/vmPFC, and striatum to predict whether participants would repeat the same response on the next trial with the same cue ("stay") or switch to another response ("shift"). Mixed-effects logistic regression yielded that dACC BOLD did not significantly predict response repetition ($\chi^2(1) = 1.294$, $p = 0.255$, $b = -0.019$, *95%-CI* [−0.050, 0.012], two-tailed; for all reported logistic regression models, assumptions of independence of residuals, low regressor collinearity, and linearity of log odds in regressors were not violated). In contrast, BOLD in PCC/vmPFC and striatum did predict response repetition, though in opposite directions: Participants were significantly more likely to repeat the same response when striatal BOLD was high

($\chi^2(1) = 9.051$, $p = 0.003$, $b = 0.067$, *95%-CI* [0.020, 0.114], two-tailed), but more likely to switch to another response when vmPFC BOLD ($\chi^2(1) = 8.765$, $p = 0.003$, $b = -0.065$, *95%-CI* [−0.104, −0.026], two-tailed) or PCC BOLD ($\chi^2(1) = 3.691$, $p = 0.030$, b = −0.036, *95%-CI* [−0.067, −0.005], two-tailed; Fig. 5H) was high (Supplementary Note 19 with Supplementary Fig. 22). Similarly, high pgACC BOLD predicted a higher likelihood of switching, likening it with the circuits formed by vmPFC and PCC ($\chi^2(1) = 15.559$, $p < 0.001$, $b = -0.076$, *95%-CI* [−0.109, −0.043], two-tailed). We also inspected the raw upsampled HRF shapes per region per condition, confirming that differential relationships were not driven by differences in HRF shapes across regions.

We also tested whether trial-by-trial midfrontal lower alpha band, theta, or beta power (within the clusters identified in the EEG-only analyses) predicted action policy updating. Participants were significantly more likely to repeat the same response when beta power was high ($\chi^2(1) = 11.886$, $p < 0.001$, $b = 0.145$, *95%-CI* [0.065, 0.225], two-tailed), but more likely to switch when theta power was high ($\chi^2(1) = 4.179$, $p = 0.041$, $b = -0.099$, *95%-CI* [−0.191, −0.007], two-tailed). Notably, unlike its BOLD correlate in ACC, lower alpha band power did predict response repetition, with more repetition when alpha power was high ($\chi^2(1) = 10.711$, $p = 0.001$, $b = 0.179$, *95%-CI* [0.077, 0.281], two-tailed; Supplementary Fig. 22).

In sum, high striatal BOLD and midfrontal beta power predicted that the same response would be repeated on the next encounter of a cue, while high vmPFC and PCC BOLD and high theta power predicted that participants would switch to another response. Thus, although both striatal and vmPFC/PCC BOLD positively encoded biased prediction errors, these two sets of regions had opposite roles in learning: while the striatum reinforced previous responses, vmPFC/PCC triggered the shift to another response strategy (Fig. 5H).

## Discussion

We investigated neural correlates of biased learning for Go and NoGo responses. In line with previous research[3,9], participants' behavior was best described by a computational model featuring faster learning from rewarded Go responses and slower learning from punished NoGo responses. Neural correlates of biased PEs were present in BOLD signals in several regions, including ACC, PCC, and striatum. These regions exhibited distinct midfrontal EEG power correlates. Most importantly, correlates of prefrontal cortical BOLD preceded correlates of striatal BOLD: Trial-by-trial dACC BOLD correlated with lower alpha band power immediately after outcome onset, followed by PCC (and vmPFC) BOLD correlated with theta power, and finally, striatal BOLD correlated with beta power. These results suggest that the architecture of the asymmetric striatal pathways might not be the only neural structure that gives rise to motivational learning biases; instead, the PFC might critically contribute to these biases.

The observation that both PFC and striatal BOLD signal reflected biased PEs might be explained by three different models. One model assumes that both PFC and striatal processes arrive at biased learning independently of each other, which is highly unlikely given strong recurrent connections between both regions[18,19,47]. Another model incorporates such interconnections, but assumes that striatum leads the PFC. While such a model is in line with past animal studies[48] and modeling work[49], it would predict EEG correlates of the PFC to trail after EEG correlates of the striatum−or at least to occur with considerable delay after outcome onset. This model is not supported by our findings, which showed EEG correlates of PFC regions soon after outcome onset, preceding striatal EEG correlates. These early EEG correlates of PFC BOLD are in line with single cell recordings in PFC which show responses confined to the first 500 ms following outcome onset[50,51], corroborating that PFC outcome processing occurs before the time of EEG correlates of striatal BOLD. The only model consistent with our data assumes recurrent connections between PFC and striatum, but with the PFC leading the striatum. Hence, these results are in

line with a model of PFC biasing striatal outcome processing, giving rise to motivational learning biases in behavior.

The dominant idea about the origin of motivational biases has been that these biases are an emergent feature of the asymmetric direct/ indirect pathway architecture in the basal ganglia[2,19]. We find that these biases are present first in prefrontal cortical areas, notably dACC and PCC, which argues against biases being purely driven by subcortical circuits. Rather, motivational learning biases might be an instance of sophisticated, even "model-based" learning processes in the striatum instructed by the PFC[52,53]. An influence of PFC on striatal RL has prominently been observed in the case of model-based vs. model-free learning[23,24] and has been stipulated as a mechanism of how instructions can impact RL[20,21]. Although there are reports of striatal processes preceding prefrontal processes within learning tasks[48,54], the opposite pattern of PFC preceding striatum has been observed as well[55] and a causal impact of PFC on striatal learning is well established[56,57]. In particular, we have previously observed that motivational biases in action selection might arise from early prefrontal inputs to the striatum, as well[17]. Prefrontal influences on striatal processes might thus be a common signature of both motivational response and learning biases.

The particular subregion of PFC showing the earliest EEG correlates was the dACC. This observation is in line with an earlier EEG-fMRI study reporting dACC to be part of an early valuation system preceding a later system comprising vmPFC and striatum[58]. The dACC has been suggested to encode models of agents' environment[59,60] that are relevant for interpreting outcomes, with BOLD in this region scaling with the size of PEs[25,26] and indexing how much should be learned from new outcomes. We hypothesize that, at the moment of outcome, dACC maintains a "memory trace" of the previously performed response[61] which might modulate the processing of outcomes as soon as they become available[62,63]. Notably, dACC exhibited stronger BOLD signal for Go than NoGo responses at the time of participants' response, but this pattern reversed at the time of outcomes. This reversal rules out the possibility that response-locked BOLD signal simply spilled over into the time of outcomes. Future research will be necessary to corroborate such a motor "memory trace" in dACC. In sum, the dACC might be in a designated position to inform subsequent outcome processing in downstream regions by modulating the learning rate as a function of the previously performed response and the obtained outcome. Rather than striatal circuits being sufficient for the emergence of motivational biases, the more "flexible" PFC seems to play an important role in instructing downstream striatal learning processes.

Striatal, dACC and PCC BOLD encoded biased PEs. In line with previous research, striatal BOLD positively linked to midfrontal beta power[41,42], which positively encoded PE valence[28,34,64], with correlations extending into alpha power. PCC and vmPFC BOLD negatively linked to midfrontal theta/delta power[17,65,66], which encoded PE valence negatively, but PE magnitude positively. Notably, theta/ delta power correlates of vmPFC/PCC BOLD preceded beta power correlates of striatal BOLD in time, which aligns with previous findings of motivational response biases being first visible in the vmPFC BOLD before they impact striatal action selection[17]. Notably, EEG correlates of striatal BOLD during outcome processing were in the beta band−in contrast to previously observed correlates of striatal BOLD during action selection in the theta band[17]. This dissociation suggests important differences in the role of the striatum in these two processes. The frequency-specific nature of these EEG-fMRI correlations further suggests that they are signatures of task-induced events that are specific to the trial phase and unlikely to reflect general anatomical connectivity. In sum, while these EEG-fMRI findings on outcome processing resemble our previous EEG-fMRI findings on action selection in that prefrontal signals precede striatal signals, they are dissociated in terms of the frequency specificity, highlighting the distinct roles of the striatum in these processes.

Positive encoding of prediction errors in striatal BOLD signal is a well-established phenomenon[38,67]. Striatal BOLD was better described by biased PEs than by standard PEs, corroborating the presence of motivational learning biases also in striatal learning processes. Notably, EEG correlates of striatal BOLD peaked rather late, suggesting that these processes are informed by early sources in PFC which are connected to the striatum via recurrent feedback loops[18,47]. Positive prediction errors increase the value of a performed action and thus strengthen action policies. Hence, it is not surprising that high striatal BOLD signal and midfrontal beta power predicted action repetition[68,69].

In contrast to striatal learning signals, the PCC and vmPFC BOLD as well as midfrontal theta and delta power signals were more complicated: Theta encoded PE valence, delta encoded PE magnitude. Both correlates showed opposite polarities. This observation is in line with previous literature suggesting that midfrontal theta and delta power might reflect the "saliency" or "surprise" aspect of PEs[31,32,70]. Surprises have the potential to disrupt an ongoing action policy[71] and motivate a shift to another policy, which might explain why these signals predicted switching to another response[72,73]. Notably, this EEG surprise signal was only significantly correlated with the biased (but not the standard) PE term, corroborating that the surprise attributed to outcomes depends on the previously performed response in line with motivational learning biases. In sum, both vmPFC and striatum encode biased PEs, though with different consequences for future action policies.

Taken together, distinct brain regions processed outcomes in a biased fashion at distinct time points with distinct EEG power correlates. Simultaneous EEG-fMRI recordings allowed us to infer when those regions reached their peak activity[74]. However, the correlational nature of BOLD-EEG links precludes strong statements about these regions actually generating the respective power phenomena. Alternatively, activity in those regions might merely modulate the amplitude of time-frequency responses originating from other sources. Furthermore, while the observed associations align with previous literature[17,41,42,65,66], the considerable distance of the striatum to the scalp raises the question whether scalp EEG could in principle reflect striatal activity, at all[75,76]. Intracranial recordings have observed beta oscillations during outcome processing in the striatum before[69,77–79]. Also, our analysis controlled for BOLD signal in motor cortex, an alternative candidate source for beta power, suggesting that late midfrontal beta power did not merely reflect motor cortex beta. Even if the striatum is not the generator of the beta oscillations over the scalp, their true (cortical) generator might be tightly coupled to the striatum and thus act as a "transmitter" of striatal beta oscillations. In fact, the analyses using trial-by-trial beta power to predict BOLD yielded significant clusters in dlPFC and SMG, two candidate regions for such a "transmitter".

We observed EEG correlates of striatal BOLD at a rather late time point after outcome onset. While we conclude that biased outcome processing occurs much earlier in cortical regions than the striatum, it is possible that the modulating influence of the striatum on cortical sources of beta synchronization over the scalp (possibly dlPFC and SMG, corroborating previous EEG-fMRI[41–43] and source-reconstruction findings[44,45]) takes time to surface. However, speaking against any delay, some single studies have reported maximal correlations between striatal LFPs and scalp EEG at a time lag of 0[80]. Regardless, even in the presence of a non-zero lag, our main conclusion would hold: Biased learning is present in cortical regions early after outcome onset, which cannot be a consequence of striatal input, but must constitute an independent origin of motivational learning biases.

In order to make inferences about the relative order of PE processing in different brain regions, we must assume that the regressor in our EEG-fMRI analysis approach−the trial-by-trial BOLD amplitude in selected regions−mostly reflects the PE signal rather than learning-

unrelated processes occurring in parallel. In support of this assumption, animal recordings have indeed found that neural activity in ACC, PCC, and striatum is dominated by reward processing during outcome receipt[81–85] and meta-analyses on human BOLD signal have found strong effect sizes for PE processing in these regions[38,67]. Importantly, we observe transient EEG-fMRI correlations that are likely event-related rather than reflecting resting-state like correlations. We thus favor the conclusion that the observed EEG-fMRI correlations reflect differences in the timing of PE processing in these regions, although we cannot fully exclude the possibility that parallel processes unrelated to (biased) learning contribute to these correlations. Note that, while outcome processing in these regions is better described by biased than by standard PEs, each region might encode PEs in an idiosyncratic way (potentially reflecting noise in the value representations[86]) and these residual idiosyncrasies drive the EEG-fMRI correlations even when controlling for biased PEs predicted by the winning computational model.

The correlational nature of the study prevents strong statements over any causal interactions between the observed regions. We assume here that a region showing an earlier midfrontal EEG correlate influences other regions showing later midfrontal EEG correlates, and such an influence is plausible given findings of feedback loops between prefrontal regions and the striatum[47]. Future studies targeting those regions via selective causal manipulations will be necessary to test for the causal role of PFC in informing striatal learning. Furthermore, while parameter recovery for most parameters in the winning computational model (including the effective learning rates incorporating the learning bias) was excellent, parameter recovery for the learning bias term itself was positive, but weaker (see Supplementary Note 6). Supplementary models tested incorporating a perseveration parameter (see Supplementary Note 8) yielded higher model recovery, but failed to capture crucial aspects of the biased learning under investigation. Future studies comprising larger samples of participants should explore alternative implementations to reliably quantify individual differences in these learning biases.

In conclusion, biased learning—increased credit assignment to rewarded action, decreased credit assignment to punished inaction—was visible both in behavior and in BOLD signal in a range of regions. EEG correlates of prefrontal cortical regions, notably dACC and PCC, *preceded* correlates of the striatum, consistent with a model of the PFC biasing RL in the striatum. The dACC appeared to hold a "motor memory trace" of the past response, biasing early outcome processing. Subsequently, biased learning was also present in vmPFC/PCC and striatum, with opposite roles in adjusting vs. maintaining action policies. These results refine previous views on the neural origin of these learning biases, suggesting they might not only rely on subcortical parts of the brain typically associated with rigid, habit-like responding, but rather incorporate frontal inputs that are associated with counterfactual reasoning and increased behavioral flexibility[87,88]. The PFC is typically believed to facilitate goal-directed over instinctive processes. Hence, PFC involvement into biased learning suggests that these biases are not necessarily agents' inescapable "fate", but rather likely act as global "priors" that facilitate learning of more local relationships. They allow for combining "the best of both worlds"—long-term experience with consequences of actions and inactions together with flexible learning from rewards and punishments.

## Methods

### Participants

Thirty-six participants ($M_{age} = 23.6$, $SD_{age} = 3.4$, range 19–32; 25 women, 11 men, gender self-reported and not of relevance in our analyses; all right-handed; all normal or corrected-to-normal vision) took part in a single 3-h data collection session, for which they received €30 flat fee plus a performance-dependent bonus (range €0–5, $M_{bonus} = €1.28$, $SD_{bonus} = 1.54$). The study was approved by the local

ethics committee (CMO2014/288; Commissie Mensengeboden Onderzoek Arnhem-Nijmegen) and all participants provided written informed consent. Exclusion criteria comprised claustrophobia, allergy to gels used for EEG electrode application, hearing aids, impaired vision, colorblindness, history of neurological or psychiatric diseases (including heavy concussions and brain surgery), epilepsy and metal parts in the body, or heart problems. Sample size was based on previous EEG studies with a comparable paradigm[9,89].

Behavioral and modeling results include all 36 participants. The following participants were excluded from analyses of neural data: For two participants, fMRI functional-to-standard image registration failed; hence, all fMRI-only results are based on 34 participants ($M_{age} = 23.47$, 25 women). Four participants exhibited excessive residual noise in their EEG data (>33% rejected trials) and were thus excluded from all EEG analyses; hence, all EEG-only analyses are based on 32 participants ($M_{age} = 23.09$, 23 women). For combined EEG-fMRI analyses, we excluded the above-mentioned six participants plus one more participant whose regression weights for every regressor were about ten times larger than for other participants, leaving 29 participants ($M_{age} = 23.00$, 22 women). Exclusions were in line with a previous analysis of this data set[17]. fMRI- and EEG-only results held when analyzing only those 29 participants (see Supplementary Notes 1–5 with Supplementary Figs. 1–4).

### Task

Participants performed a motivational Go/ NoGo learning task[3,9] administered via MATLAB 2014b (MathWorks, Natick, MA, United States) and Psychtoolbox-3.0.13. On each trial, participants saw a gem-shaped cue for 1300 ms which signaled whether they could potentially win a reward (Win cues) or avoid a punishment (Avoid cues) and whether they had to perform a Go (Go cue) or NoGo response (NoGo cue). They could press a left ($Go_{LEFT}$), right ($Go_{RIGHT}$), or no (NoGo) button while the cue was presented. Only one response option was correct per cue. Participants had to learn both cue valence and required action from trial-and-error. After a variable inter-stimulus-interval of 1400–1600 ms, the outcome was presented for 750 ms. Potential outcomes were a reward (symbolized by coins falling into a can) or neutral outcome (can without money) for Win cues, and a neutral outcome or punishment (symbolized by money falling out of a can) for Avoid cues. Feedback validity was 80%, i.e., correct responses were followed by positive outcomes (rewards/ no punishments) on only 80% of trials, while incorrect responses were still followed by positive outcomes on 20% of trials. Trials ended with a jittered inter-trial interval of 1250–2000 ms, yielding total trial lengths of 4700–6650 ms.

Participants gave left and right Go responses via two button boxes positioned lateral to their body. Each box featured four buttons, but only one button per box was required in this task. When participants accidentally pressed a non-instructed button, they received the message "Please press one of the correct keys" instead of an outcome. In the analyses, these responses were recoded into the instructed button on the respective button box. In the fMRI GLMs, such trials were modeled with a separate regressor.

Before the task, participants were instructed that each cue could be followed by either reward or punishment, that each cue had one optimal response, that feedback was probabilistic, and that the rewards and punishments were converted into a monetary bonus upon completion of the study. They performed an elaborate practice session in which they got familiarized first with each condition separately (using practice stimuli) and finally practiced all conditions together. They then performed 640 trials of the main task, separated into two sessions of 320 trials with separate cue sets. Introducing a new set of cues allowed us to prevent ceiling effects in performance and investigate continuous learning throughout the task. Each session featured eight cues that were presented 40 times. After every 100–110 trials

(~6 min), participants could take a self-paced break. The assignment of the gems to cue conditions was counterbalanced across participants, and trial order was pseudo-random (preventing that the same cue occurred on more than two consecutive trials).

## Behavior analyses

We used mixed-effects logistic regression (as implemented in the package *lme4* version 1.1.26 in R version 3.3.2) to analyze behavioral responses (Go vs. NoGo) as a function of required action (Go/ NoGo), cue valence (Win/ Avoid), and their interaction. We included a random intercept and all possible random slopes and correlations per participant to achieve a maximal random-effects structure[90]. Sum-to-zero coding was employed for the factors. Type 3 *p*-values were based on likelihood ratio tests (implemented in the R package *afex* version 0.28.1). We used a significance criterion of α = 0.05 for all the analyses.

Furthermore, we used mixed-effects logistic regression to analyze "stay behavior", i.e., whether participants repeated an action on the next encounter of the same cue, as a function of outcome valence (positive: reward or no punishment/negative: no reward or punishment), outcome salience (salient: reward or punishment/neutral: no reward or no punishment), and performed action (Go/NoGo). We again included all possible random intercepts, slopes, and correlations.

## Computational modeling

We fit a series of increasingly complex RL models to participants' choices to decide between different algorithmic explanations for the emergence of motivational biases in behavior. We employed the same set of nested models as in previous studies using this task[3,9]. For tests of alternative biases specifications, see Supplementary Notes 7–9 with Supplementary Figs. 6–8.

To determine whether a Pavlovian response bias, a learning bias, or both biases jointly predicted behavior best, we fitted a series of increasing complex computational models. In each trial (t), choice probabilities for all three response options (a) given the displayed cue (s) were computed from their action weights (modified Q-values) using a softmax function:

$$p(a_t|s_t) = \frac{\exp(w(a_t, s_t))}{\sum_a \exp(w(a', s_t))} \qquad (1)$$

After each response, action values were updated with the prediction error based on the obtained outcome $r \in \{-1; 0; 1\}$. As the starting model (M1), we fitted an standard delta-learning model[91] in which action values were updated with prediction errors, i.e., the deviation between the experienced outcome and expected outcome. This model contained two free parameters: the learning rate (ε) scaling the updating term and the feedback sensitivity (ρ) scaling the received outcome (i.e., higher feedback sensitivity led to choices more strongly guided by value difference, akin to the role of the inverse temperature parameter frequency used in RL models):

$$Q_t(a_t, s_t) = Q_{t-1}(a_t, s_t) + \varepsilon(\rho r - Q_{t-1}(a_t, s_t)) \qquad (2)$$

In this model, choice probabilities were fully determined by action values, without any bias. We initialized action values $Q_0$ such that they reflected a "neutral" expected value for each action. Win cues could lead to reward (+1) or neutral (0) outcomes and Avoid cues to neutral (0) or punishment (−1) outcomes. A neutral expected value would assign equal probability to either possible outcome, leading to expectations of +1/2 and −1/2, respectively. In addition, because participants' feedback sensitivity parameter ρ reflected how participants weighed the outcomes they received, also the initial values had to be multiplied with the feedback sensitivity to stay neutral between 0 and participants' re-weighted positive/ negative outcome of +/−1*ρ. Thus,

initial action values $Q_0$ were set to 1/2*ρ (Win cues) and −1/2*ρ (Avoid cues).

Unlike previous versions of the task[3], cue valences were not instructed, but had to be learned from outcomes, as well[9]. Thus, until experiencing the first non-neutral outcome (reward or punishment) for a cue, participants could not know its valence and thus not learn from neutral feedback. Hence, for these early trials, action values were multiplied with zero when computing choice probabilities[9]. After the first encounter of a valenced outcome, action values were "unmuted" and started to influence choices probabilities, retrospectively considering all previous outcomes[9].

In M2, we added the Go bias parameter *b*, which accounted for individual differences in participants' overall propensity to make Go responses, to the action values Q, resulting in action weights w-

$$w(a_t, s_t) = \begin{cases} Q_t(a_t, s_t) + b & if\, a = Go \\ Q_t(a_t, s_t) & else \end{cases} \qquad (3)$$

In M3, we added a Pavlovian response bias π, scaling how positive/ negative cue valence (Pavlovian values) increased/ decreased the weights of Go responses:

$$w(a_t, s_t) = \begin{cases} Q_t(a_t, s_t) + b + \pi V(s) & if\, a = Go \\ Q_t(a_t, s_t) & else \end{cases} \qquad (4)$$

Participants were instructed that a cue was either a Win cue (affording rewards or neutral outcomes) or an Avoid cue (affording neutral outcomes or punishments). Hence, cue valence (Win/ Avoid) did not have to be learned instrumentally; instead, it could be inferred as soon participants experienced a non-neutral outcome. Until that moment, cue valence $V(s)$ was set to zero. Afterwards, $V(s)$ was set to +0.5 for Win cues and −0.5 for Avoid cues. Note that choosing different values than 0.5 would merely rescale the bias parameter π (e.g., halving π with cue valences of +1 and −1) without any changes in the model's predictions. The Pavlovian response bias affected left-hand and right-hand Go responses similarly and thus reflected generalized activation/ inactivation by the cue valence.

In M4, we added a learning bias κ, increasing the learning rate for rewards after Go responses and decreasing it for punishments after NoGo responses. The learning bias was specific to the response shown, thus reflecting a specific enhancement in action learning/ impairment in unlearning for that particular response. Conceptually, learning rates differed between response-outcome conditions in the following way:

$$\varepsilon = \begin{cases} \varepsilon_0 + \kappa & if\, r_t = 1\, and\, a = go \\ \varepsilon_0 - \kappa & if\, r_t = -1\, and\, a = nogo \\ \varepsilon_0 & else \end{cases} \qquad (5)$$

In the technical implementation of this model, learning rates were sampled in continuous space and then inverse-logit transformed to constrain them to the range [0 1][3,9]. However, after this transformation, the impact of adding vs. subtracting the learning bias κ would no longer be symmetric. Hence, for baseline learning rates $\varepsilon_0 < 0.5$, we first computed the difference between the baseline learning rate and the learning rates for punished NoGo responses and used this difference to compute the learning rate for rewarded Go responses:

$$\varepsilon = \begin{cases} \varepsilon_0 = inv.logit(\varepsilon) \\ \varepsilon_{punished\,NoGo} = inv.logit(\varepsilon - \kappa) & if\, \varepsilon_0 < 0.5 \\ \varepsilon_{rewarded\,Go} = \varepsilon_0 + (\varepsilon_0 - \varepsilon_{punished\,NoGo}) & if\, \varepsilon_0 < 0.5 \end{cases} \qquad (6)$$

Notably, this procedure is only guaranteed to work when $\varepsilon_0 < 0.5$. For $\varepsilon_0 > 0.5$, the difference term could become >0.5 and the learning rate for rewarded Go responses would become >1, which is impractical.

Hence, for $\varepsilon_0 > 0.5$, we first computed the learning rate for rewarded Go responses and used the difference to the baseline learning rate $\varepsilon_0$ to compute the learning rate for punished NoGo responses:

$$\varepsilon = \begin{cases} \varepsilon_0 = inv.logit(\varepsilon) \\ \varepsilon_{rewarded\,Go} = inv.logit(\varepsilon + \kappa) & if\ \varepsilon_0 > 0.5 \\ \varepsilon_{punished\,NoGo} = \varepsilon_0 - (\varepsilon_{rewarded\,Go} - \varepsilon_0) & if\ \varepsilon_0 > 0.5 \end{cases} \quad (7)$$

In the model M5, we included both the Pavlovian response bias and the learning bias.

The weakly informative hyperpriors were set to $X_\rho \sim \mathcal{N}(2, 3)$, $X_\varepsilon \sim \mathcal{N}(0, 2)$, $X_{b,\pi,\kappa} \sim \mathcal{N}(0, 3)$, in line with previous implementations of this model[3,9]. The same priors (for the same parameters) were used across different model implementations to not bias model comparison. Alternative hyperpriors did not change the results. For computing the participant-level parameters, $\rho$ was exponentiated to constrain it to positive values, and the inverse-logit transformation was applied to $\varepsilon$.

For model fitting and comparison, we used hierarchical Bayesian inference as implemented in the CBM toolbox in MATLAB[92]. This approach combines hierarchical Bayesian parameter estimation with random-effects model comparison[93]. The fitting procedure involves two steps, starting with the Laplace approximation of the model evidence to compute the group evidence, which quantifies how well each model fits the data while penalizing for model complexity. Both group-level and individual-level parameters are estimated using an iterative algorithm. We used wide Gaussian priors (see hyperpriors above) and exponential and sigmoid transforms to constrain parameter spaces. Subsequent random-effects model selection allows for the possibility that different models generated the data for different participants. Participants contribute to the group-level parameter estimation in proportion to how well a given model fits their data, quantified via a responsibility measure (i.e., the probability that the model at hand is responsible for generating data of the respective participant). This model-comparison approach has been shown to be less susceptible to the influence of outliers[92]. We selected the "winning" model based on the protected exceedance probability.

We assured that the winning model was able to reproduce the data, using the sampled combinations of participant-level parameter estimates to create 3600 agents that "played" the task. We employed two approaches to simulate the task: *posterior predictive model simulations* and *one-step-ahead model predictions*. In the posterior predictive model simulations, agents' choices were sampled probabilistically based on their action values, and outcomes probabilistically sampled based on their choices. This method ignores participant-specific choice histories and can thus yield choice/outcome sequences that diverge considerably from participants' actual experiences. In contrast, one-step-ahead predictions use participants' actual choices and experienced outcomes in each trial to update action values. We simulated choices for each participant using both methods, which confirmed that the winning model M5 ("asymmetric pathways model") was able to qualitatively reproduce the data, while an alternative implementation of biased learning ("action priming model") failed to do so (see Supplementary Note 7 with Supplementary Fig. 6).

## fMRI data acquisition
fMRI data were collected on a 3T Siemens Magnetom Prisma fit MRI scanner with a 64-channel head coil. During scanning, participants' heads were restricted using foam pillows and strips of adhesive tape were applied to participants' forehead to provide active motion feedback and minimize head movement[94]. After two localizer scans to position slices, we collected functional scans with a whole-brain T2*-weighted sequence (68 axial-oblique slices, TR = 1400 ms, TE = 32 ms, voxel size 2.0 mm isotropic, interslice gap 0 mm, interleaved multi-band slice acquisition with acceleration factor 4, FOV 210 mm, flip

angle 75°, A/P phase encoding direction). The first seven volumes of each run were automatically discarded. This sequence was chosen because of its balance between a short TR and relatively high spatial resolution, which was required to disentangle cue and outcome-related neural activity. Pilots using different sequences yielded that this sequence performed best in reducing signal loss in striatum.

Furthermore, after task completion, we removed the EEG cap and collected a high-resolution anatomical image using a T1-weighted MP-RAGE sequence (192 sagittal slices per slab, GRAPPA acceleration factor = 2, TI = 1100 ms, TR = 2300 ms, TE = 3.03 ms, FOV 256 mm, voxel size 1.0 mm isotropic, flip angle 8°) which was used to aid image registration, and a gradient fieldmap (GRE; TR = 614 ms, TE1 = 4.92 ms, voxel size 2.4 mm isotropic, flip angle 60°) for distortion correction. For one participant, no fieldmap was collected due to time constraints. At the end of each session, an additional DTI data collection took place; results will be reported elsewhere.

## fMRI preprocessing
All fMRI pre-processing was performed in FSL 6.0.0. After cleaning images from non-brain tissue (brain-extraction with BET), we performed motion correction (MC-FLIRT), spatial smoothing (FWHM 3 mm), and used fieldmaps for B0 unwarping and distortion correction in orbitofrontal areas. We used ICA-AROMA[95] to automatically detect and reject independent components associated with head motion. Finally, images were high-pass filtered at 100 s and pre-whitened. After the first-level GLM analyses, we computed and applied co-registration of EPI images to high-resolution images (linearly with FLIRT using boundary-based registration) and to MNI152 2 mm isotropic standard space (non-linearly with FNIRT using 12 DOF and 10 mm warp resolution).

## ROI selection
For fMRI-informed EEG analyses, we first created a functional mask as the conjunction of the $PE_{STD}$ and $PE_{DIF}$ contrasts by thresholding both z-maps at $z > 3.1$, binarizing, and multiplying them (see Supplementary Note 10 with Supplementary Figs. 9 and 10). After visual inspection of the respective clusters, we created seven anatomical masks based on the probabilistic Harvard-Oxford Atlas (included in FSL; thresholded at 10%): striatum and ACC (see above), vmPFC (combined frontal pole, frontal medial cortex, and paracingulate gyrus), motor cortex (combined precentral and postcentral gyrus), PCC (Cingulate Gyrus, posterior division), ITG (Inferior Temporal Gyrus, posterior division, and Inferior Temporal Gyrus, temporooccipital part) and primary visual cortex (Lingual Gyrus, Occipital Fusiform Gyrus, Occipital Pole). We then multiplied this functional mask with each of the seven anatomical masks, returning seven masks focused on the respective significant clusters, which were then used for signal extraction. For the dACC mask, we manually excluded voxels in pgACC belonging to a distinct cluster. Masks were back-transformed to each participant's native space.

For bar plots in Fig. 3A, we multiplied the anatomical masks of vmPFC and striatum specified above with the binarized outcome valence contrast.

## fMRI analyses
For each participant, data were modeled using two event-related GLMs. First, we performed a model-based GLM in which we used trial-by-trial estimates of biased PEs as regressors. Second, we used another model-free GLM in which we modeled all possible action x outcome combinations via outcome-locked categorical regressors while at the same time modeling response-locked left- and right-hand response regressors. This model free GLM also contained the outcome valence contrast reported as an initial manipulation check.

In the model-based GLM, we used two model-based regressors that reflected the trial-by-trial prediction error (PE) update term. The

update term was computed by multiplying the prediction-error with the condition-specific learning rate. As described above, in the winning model M5, the learning bias term κ leads to altered learning from "congruent" action-outcome pairs, with faster learning of Go actions followed by rewards, but slower unlearning of NoGo actions followed by punishments. To compute trial-by-trial updates, we extracted the group-level parameters of the best fitting computational model M5 (asymmetric pathways model) and used those parameters to compute the prediction error on every trial for every participant. Using the same parameter for each participant is warranted when testing for the same qualitative learning pattern across participants[96]. Given that both standard (base model M1) and biased (winning model M5) PEs were highly correlated (mean correlation of 0.921 across participants, range 0.884–0.952), it appeared difficult to distinguish standard learning from biased learning. As a remedy, we decomposed the biased PE into the standard PE plus a difference term as $PE_{BIAS} = PE_{STD} + PE_{DIF}$[22,36]. Any region displaying truly biased learning should significantly encode *both* the standard PE term and the difference term. The standard PE and difference term were much less correlated (mean correlation of −0.020, range −0.326–0.237). To control for cue-related activation, we furthermore added four regressors spanned by crossing cue valence and performed action (Go response to Win cue, Go response to Avoid cue, NoGo response to Win cue, NoGo response to Avoid cue).

The model-free GLM included a separate regressor for each of the eight conditions obtained when crossing performed action (Go/NoGo) and obtained outcome (reward/no reward/no punishment/punishment). We fitted four contrasts: (1) one contrast comparing conditions with positive (reward/no punishment) and negative (no reward/punishment) outcomes, used as a quality check to identify regions that encoded outcome valence; (2) one contrast comparing Go vs. NoGo responses at the time of the outcome; (3) one contrast summing of left- and right-hand responses, reflecting Go vs. NoGo responses at the time of the response; and (4) one contrast subtracting right- from left-handed responses, reflecting lateralized motor activation. As this GLM resulted in empty regressors for several participants when fitted on a block level, making it impossible to use the data of the respective blocks on a higher level, we instead concatenated blocks and performed a single GLM per participant. We therefore registered the data from all blocks to the middle image of the first block (default reference volume in FSL) using MCFLIRT. The first and last 20 s of each block did not feature any task-related events, such that carry-over effects of task events in the design matrix from one block to another were not possible.

In both GLMs, we added four regressors of no interest: one for the motor response (left = +1, right = −1, NoGo = 0), one for error trials, one for outcome onset, and one for trials with invalid motor response (and no outcome respectively). We also added nine or more nuisance regressors: the six realignment parameters from motion correction, mean cerebrospinal fluid (CSF) signal, mean out-of-brain (OBO) signal, and a separate spike regressor for each volume with a relative displacement of more than 2 mm (occurred in 10 participants; in those participants: M = 7.40, range 1–29). For the model-free GLM, nuisance regressors were added separately for each block as well as an overall intercept per block. We convolved task regressors with double-gamma haemodynamic response function (HRF) and high-pass filtered the design matrix at 100 s.

First-level contrasts were fit in native space. Afterwards, co-registration and reslicing was applied to participants' contrast maps, which were then combined on a (participant and) group level using FSL's mixed effects models tool FLAME with a cluster-forming threshold of z > 3.1 and cluster-level error control at α < 0.05 (i.e., two one-sided tests with α < 0.025).

## EEG data acquisition
We recorded EEG data with 64 channels (BrainCap-MR-3-0 64Ch-Standard; Easycap GmbH; Herrsching, Germany; international 10–20

layout, reference electrode at FCz) plus channels for electro-cardiogram, heart rate, and respiration (used for MR artifact correction) at a sampling rate of 1000 Hz. We placed MRI-compatible EEG amplifiers (BrainAmp MR plus; Brain Products GmbH, Gilching, Germany) behind the MR scanner and attached cables to the participants once they were located in final position in the scanner. Furthermore, we fixated cables using sand-filled pillows to reduce artifacts induced through cable movement in the magnetic field. During functional scans, the MR helium pump was switched off to reduce EEG artifacts. After the scanning, we recorded the exact EEG electrode locations on participants' heads relative to three fiducial points using a Polhemus FASTRAK device. For four participants, no such data were available due to time constraints/ technical errors, in which case we used the average electrode locations of the remaining 32 participants.

## EEG pre-processing
First, raw EEG data were cleaned from MR scanner and cardioballistic artifacts using BrainVisionAnalyzer[97]. The rest of the pre-processing was performed in Fieldtrip in MATLAB 2018b[98]. After rejecting channels with high residual MR noise (mean 4.8 channels per participant, range 1–13), we epoched trials into time windows of −1400–2000 ms relative to the onset of outcomes. Timing of this epochs was determined by the minimal inter-stimulus interval beforehand until the minimal inter-trial interval afterwards. Data was re-referenced to the grand average, which allowed us to recover the reference as channel FCz, and then band-pass filtered using a two-pass 4th order Butterworth IIR filter (Fieldtrip default) in the range of 0.5–35 Hz. These filter settings allowed us to distinguish the delta, theta, alpha, and beta band, while filtering out residual high-frequency MR noise. This low-pass filter cut-off was different from a previous analysis of this data in which we set it at 15 Hz[17] because, in this analysis, we had a hypothesis on outcome valence encoding in the beta range. We then applied linear baseline correction based on the 200 ms prior to cue onset and used ICA to detect and reject independent components related to eye-blinks, saccades, head motion, and residual MR artifacts (mean number of rejected components per participant: 32.694, range 24–45). Afterwards, we manually rejected trials with residual motion (for all 36 participants: M = 117.722, range 11–499). Based on trial rejection, four participants for which more than 211 (33%) of trials were rejected were excluded from any further analyses (rejected trials after excluding those participants: M = 81.875, range 11–194). Finally, we computed a Laplacian filter with the spherical spline method to remove global noise (using the exact electrode positions recorded with Polhemus FASTRAK), which we also used to interpolate previously rejected channels. This filter attenuates more global signals (e.g., signal from deep sources or global noise) and noise (heart-beat and muscle artifacts) while accentuating more local effects (e.g., superficial sources).

## EEG TF decomposition
We decomposed the trial-by-trial EEG time series into their time-frequency representations using 33 Hanning tapers between 1 and 33 Hz in steps of 1 Hz, every 25 ms from −1000 until 1300 ms relative to outcome onset. We first zero-padded trials to a length of 8 s and then performed time-frequency decomposition in steps of 1 Hz by multiplying the Fourier transform of the trial with the Fourier transform of a Hanning taper of 400 ms width, centered around the time point of interest. This procedure results in an effective resolution of 2.5 Hz (Rayleigh frequency), interpolated in 1 Hz steps, which was more robust to the choice of exact frequency bins. To exclude the possibility of slow drifts in power over the time course of the experiment, we performed baseline correction across participants and trials by fitting a linear model for each channel/ frequency combination with trial number as predictor and the average power 250–50 ms before outcome onset as outcome, and subtracting the power predicted by this model from the data. This procedure is able to remove slow linear

drifts in power over time from the data. In absence of such drifts, it is equivalent to correcting all trials by the grand mean across trials per frequency in the selected baseline time window. Afterwards, we averaged power over trials within each condition spanned by performed action (Go/NoGo) and outcome (reward/no reward/no punishment/ punishment). We finally converted the average time-frequency data per condition to decibel to ensure that data across frequencies, time points, electrodes, and participants were on same scale.

### EEG analyses
All analyses were performed on the average signal of a-priori selected channels Fz, FCz, and Cz based on previous literature[9,17]. We again performed model-free and model-based analyses. For the model-free analyses, we sorted trials based on the performed action (Go/NoGo) and obtained outcome (reward/no reward/no punishment/punishment) and computed the mean TF power across trials for each of the resultant eight conditions for each participant. We tested whether theta power (average power 4–8 Hz) and beta power (average power 13–30 Hz) encoded outcome valence by contrasting positive (reward/ no punishment) and negative (no reward/punishment) conditions (irrespective of the performed action). We also tested for differences between Go and NoGo responses in the lower alpha band (6–10 Hz). For all contrasts, we employed two-sided cluster-based permutation tests in a window from 0 to 1000 ms relative to outcome onset. For beta power, results were driven by a cluster that was at the edge of 1000 ms; to more accurately report the time span during which this cluster exceeded the threshold, we extended the time window to 1300 ms in this particular analysis. Such tests are able to reject the null hypothesis of exchangeability of two experimental conditions, but they are not suited to precisely locate clusters in time-frequency space. Hence, interpretations were mostly based on the visual inspection of plots of the signal time courses.

For model-based analyses, similar to fMRI analyses, we used the group-level parameters from the best fitting computational model M5 to compute the trial-by-trial biased PE term and decomposed it into the standard PE term and the difference to the biased PE term. We used both terms as predictors in a multiple linear regression for each channel-time-frequency bin for each participant, and then performed one-sample cluster-based permutation-tests across the resultant *b*-maps of all participants[99]. For further details on this procedure, see fMRI-informed EEG analyses.

### fMRI-informed EEG analyses
The BOLD signal is sluggish. It is thus hard to determine when different brain regions become active. In contrast, EEG provides much higher temporal resolution. A fruitful approach can be to identify distinct EEG correlates of the BOLD signal in different regions, allowing to test hypotheses about the temporal order in which regions might become active and modulated EEG power[17,74]. Furthermore, by using the BOLD signal from different regions in a multiple linear regression, one can control for variance shared among regions (e.g., changes in global signal; variance due to task regressors) and test which region is the best unique predictor of a certain EEG signal. In such an analysis, any correlation between EEG and BOLD signal from a certain region reflects an association above and beyond those induced by task conditions.

We used the trial-by-trial BOLD signal in selected regions in a multiple linear regression to predict EEG signal over the scalp[17,74] (building on existing code from https://github.com/tuhauser/TAfT; see Supplementary Note 15 with Supplementary Fig. 17 for a graphical illustration). As a first step, we extracted the volume-by-volume signal (first eigenvariate) from each of the seven regions identified to encode biased PEs (conjunction of $PE_{STD}$ and $PE_{DIF}$: striatum, dACC, pgACC, left motor cortex, PCC, left ITG, and primary visual cortex). We applied a highpass-filter at 128 s and regressed out nuisance regressors (6

realignment parameters, CSF, OOB, single volumes with strong motion, same as in the fMRI GLM). We then upsampled the signal by a factor 10, epoched it into trials of 8 s duration, and fitted a separate HRF (based on the SPM template) to each trial (58 upsampled data points), resulting in trial-by-trial regression weights reflecting the respective BOLD response. We then combined the regression weights of all trials and regions of a certain participant into a design matrix with trials as rows and the seven ROIs as columns, which we then used to predict power at each time-frequency-channel bin. As further control variables, we added the behavioral $PE_{STD}$ and $PE_{DIF}$ regressors to the design matrix. All results were identical with and without the inclusion of PEs as covariates in the regression, suggesting that EEG-fMRI correlations did not merely arise from both modalities encoded PEs as a "common cause" that induced correlations. Instead, these correlations reflected the incremental variance explained in EEG power that was afforded by the BOLD signal even beyond the PEs. All predictors and outcomes were demeaned such that the intercept became zero. Such a multiple linear regression was performed for each participant, resulting in a time-frequency-channel-ROI *b*-map reflecting the association between trial-by-trial BOLD signal and TF power at each time-frequency-channel bin. *B*-maps were Fisher-*z* transformed, which makes the sampling distribution of correlation coefficients approximately normal and allows for combining them across participants. Finally, we tested for fMRI-EEG associations with a cluster-based one-sample permutation *t*-test[99] on the mean regression weights over channels Fz, FCz, and Cz across participants in the range of 0–1000 ms, 1–33 Hz. We first obtained a null distribution of maximal cluster mass statistics from 10000 permutations. For each permutation, we flipped the sign of the *b*-map of a random subset of participants, computed a separate *t*-test at each time-frequency bin (bins of 25 ms, 1 Hz) across participants (results in *t*-map), thresholded these maps at |t| > 2, and finally computed the maximal cluster mask statistic (sum of all *t*-values) for any cluster (adjacent voxels above threshold). Afterwards, we computed the same *t*-map for the real data, identified the cluster with the biggest cluster-mass statistic, and computed the corresponding *p*-value as number of permutations in the null distribution that were larger than the maximal cluster mass statistic in the real data.

### EEG-informed fMRI analyses
For the EEG-informed fMRI analyses, we fit three additional GLMs for which we entered the trial-by-trial theta/delta power (1–8 Hz), beta power (13–30 Hz), and lower alpha band power (6–10 Hz) as parametric regressors on top of the task regressors of the model-free GLM. These measures were created by using the 3-D (time-frequency-channel) *t*-map obtained when contrasting positive vs. negative outcomes (theta/ delta and beta; Fig. 4A, B) and Go vs. NoGo conditions (lower alpha band) as a linear filter (Fig. 4; see Supplementary Note 15 with Supplementary Fig. 18 for a graphical illustration of this approach). Note that these signals were selected based on the EEG-only results and not informed by the fMRI-informed EEG analyses. We enforced strict frequency cut-offs. For lower alpha band and beta, we used midfrontal channels (Fz/FCz/Cz). For theta/ delta power, given the topography that reached far beyond midfrontal channels and over the entire frontal scalp, we used a much wider ROI (AF3/ AF4/ AF7/ AF8/ F1/ F2/ F3/ F4/ F5/F6/F7/F8/FC1/FC2/FC3/FC4/FC5/FC6/FCz/Fp1/Fp2/Fpz/Fz). We extracted those maps and retained all voxels with *t* > 2. These masks were applied to the trial-by-trial time-frequency data to create weighted summary measures of the average power in the identified clusters in each trial. For trials for which EEG data was rejected, we imputed the participant mean value of the respective action (Go/ NoGo) x outcome (reward/no reward/no punishment/punishment) condition. Note that this approach accentuates differences between conditions, which were already captured by the task regressors in the GLM, but decreases trial-by-trial variability within each condition,

which is of interest in this analysis. This imputation approach is thus conservative. While trial-by-trial beta and theta power were largely uncorrelated, mean $r = 0.104$, range $-0.118$–$0.283$ across participants, and so were beta and alpha, mean $r = 0.097$, range $-0.162$–$0.284$ across participants, theta and alpha power were moderately correlated, mean $r = 0.412$, range $0.121$–$0.836$ across participants, warranting the use of a separate channel ROI for theta and using separate GLMs for each frequency band.

**Analyses of behavior as a function of BOLD signal and EEG power**
We used mixed-effects logistic regression to analyze "stay behavior", i.e., whether participants repeated an action on the next encounter of the same cue, as a function of BOLD signal and EEG power in selected regions. For analyses featuring BOLD signal, we used the trial-by-trial HRF amplitude also used for fMRI-informed EEG analyses. For analyses featuring EEG, we used the trial-by-trial EEG power also used in the EEG-informed fMRI analyses.

**Reporting summary**
Further information on research design is available in the Nature Portfolio Reporting Summary linked to this article.

## Data availability
This is a re-analysis of previously published data[17]. This raw data has previously been published on the Radboud Data Repository under: https://doi.org/10.34973/pezs-pw62. Preprocessed data and fMRI results presented in this paper have been deposited on the Radboud Data Repository under: https://doi.org/10.34973/peg8-xy67. In line with requirements of the Ethics Committee and the Radboud University security officer, potentially identifying data (such as imaging data) can only be shared to identifiable researchers. Hence, researchers requesting access to these repositories have to register and accept a data user agreement; access will then automatically be granted via a "click-through" procedure (without involvement of authors or data stewards). Group-level unthresholded fMRI z-maps are available on Neurovault (https://neurovault.org/collections/11184/). Source data are provided with this paper.

## Code availability
All code required to achieve the reported results is available under: https://doi.org/10.34973/peg8-xy67. Code will be maintained under https://github.com/johalgermissen/Algermissen2024NatComms, with a permanent copy at the time of publication under https://github.com/denoudenlab/Algermissen2024NatComms and on Zenodo (https://doi.org/10.5281/zenodo.10352241)[100].

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

## Acknowledgements

We thank Emma van Dijk for assistance with data collection, Michael J. Frank for helpful discussions, and the weekly Donders M/EEG meeting for discussions of these results and many helpful suggestions. J.C.S. was funded by Netherlands Organization for Scientific Research (NWO) research talent grant 406-14-028. R.S. was funded by Netherlands Organization for Scientific Research (NWO) VENI grant 451-12-021. R.C. was funded by Netherlands Organization for Scientific Research (NWO) VICI grant 453-14-005, Netherlands Organization for Scientific Research (NWO) Ammodo KNAW Award 2017, and James S. McDonnell Foundation James McDonnell Scholar Award. HEMDO was funded by Netherlands Organization for Scientific Research (NWO) VIDI grant 452-17-016.

## Author contributions

Conceptualization: J.A., J.C.S., R.C., H.E.M.D.O. Data curation: J.A. Formal analysis: J.A. Funding acquisition: J.C.S., R.C., H.E.M.D.O. Investigation: J.A., J.C.S. Methodology: J.A., H.E.M.D.O. Project administration: J.A., J.C.S., H.E.M.D.O. Resources: R.C., H.E.M.D.O. Software: J.A., J.C.S., H.E.M.D.O. Supervision: J.C.S., R.S., R.C., H.E.M.D.O. Validation: J.A., J.C.S., R.S., R.C., H.E.M.D.O. Visualization: J.A. Writing – original draft: J.A., H.E.M.D.O. Writing – review & editing: J.A., J.C.S., R.S., R.C., H.E.M.D.O.

## Competing interests

The authors declare no competing interests.
