## [Peer Review File · Nature Communications]

Prefrontal signals precede striatal signals for biased credit assignment to (in)actionsREVIEWER COMMENTS

Reviewer #1 (Remarks to the Author):

The manuscript by Algermissen and colleagues presents a study combining simultaneous EEG-fMRI during a learning task with computational modelling of the learning/choice behaviour. They address the question of where biases in (a) action selection (b) instrumental learning arise. Here, (a) refers to the observation that agents are more adept at performing vs withholding a response to obtain a reward and, conversely, better at suppressing vs performing an action to avoid punishment. (b) refers to agents more readily assigning credit for a reward to action vs inaction, and more readily assigning credit for avoiding punishment to inaction vs action. The central question of this manuscript is whether such biased learning arises in the striatum, or instead in the prefrontal cortex which then biases learning in the striatum.

The authors find fMRI correlates of biased prediction errors (PE) in (amongst others) ventromedial prefrontal cortex and striatum. In the EEG, there is a representation of biased PE sign in mid-frontal theta and beta power. Using fMRI-informed EEG, they show that fluctuations of the trial-by-trial BOLD signal in prefrontal cortex are represented in (mostly mid-frontal) EEG before the emergence of striatal contributions. Based on this, they argue that biased learning arises in the frontal cortex, not in the striatum.

Overall, this is clearly an impressive study that must have been a tour-de-force to conduct and analyze. I am not entirely convinced yet that the approaches taken by the authors can really answer the central question of whether the bias originates in the striatum or prefrontal cortex (see my first major point below). However, even if the answer to this question should be negative, I think one should ask whether this study does not still provide enough important results that would be important in their own right. I am inclined to think this would be the case. This however rests on whether or not a few methodological questions that I have (particularly about the "biased PE") can be addressed/answered.

1) Regarding the fMRI-EEG analyses presented in Fig. 5. I am not convinced this informs us about whether the relevant computations occur first in striatum or cortex. My understanding is that the authors extracted each trial's BOLD signal from the relevant ROI and then, for each trial, fit an HRF to this signal, yielding a coefficient for each trial and ROI. This coefficient represents the BOLD signal on each trial, however, does not contain specific information about the PE. These coefficients are then regressed against the EEG signal. Thus, to me it appears that the results merely indicate when and at what frequency the EEG correlates with striatal BOLD following outcome presentation, independent of any particular PE representation. It also seems to be the case that the correlations between BOLD and EEG (Fig 5) emerge at very short latency (almost immediately after outcome presentation) compared to the representations of PE sign in (Fig 4), which seem to occur considerably later. I also think it might be more sensible (Fig 5D) to use the latency to first emergence of a significant correlation. Thus, currently I would say that all we can say with confidence from this analysis is that ACC and PCC correlate with cortical EEG at an earlier time point and at different frequency band compared to striatum, which may simply reflect the general anatomical connectivity. To resolve this, maybe it would be possible to incorporate the PE as a covariate in these analyses? (However, see my concern below about the "biased PE").

2) I am skeptical about the PE_bias. As the authors highlight, the PE from this (the winning) model is correlated at $r = 0.92$ with the PE from a standard Rescorla-Wagner model (PE_std).

To remedy this issue, they compute a PE_diff as

$$PE_diff = PE_bias - PE_std$$

This is also used as reasoning for the analysis presented in Fig. 3 (line 195): "We tested for biased PEs PE_BIAS by computing which regions significantly encoded the conjunction of both its components, i.e., standard prediction errors PE_STD and the difference to biased PEs PE_DIF."

I am not convinced this solves the issue. The PE in the model arise from different learning mechanism, can one say that the PE_diff is simply the PE_std plus something else? It does not sound convincing to me to use the PE_diff as a proxy for the PE_bias, but maybe I am getting things wrong here. I am wondering whether it would not be more appropriate to orthogonalize for (regress out the contributions of) the PE_std, although I appreciate that, given the high correlation, it may be difficult to still detect anything.

3) Representation of biased PE in the EEG signal (Fig. 4). I would argue that what is shown here is not a representation of PE_bias (magnitude), but more likely of valence, as the regressor that is shown here is the sign regressor? Then, in 4D the correlation of PE_bias with delta is shown - however the authors have argued (rightly so I think) above that this regressor is hard to interpret given the high correlation with PE_std. Accordingly, I also think the statements about delta power and PE magnitude do not hold (line 232/233: "In sum, ...delta power encoded PE magnitude (positively)."; line 420: "...and delta encoded PE magnitude").

4) I have a few questions about the modelling:

4a) The authors have performed a model validation and show that the model quite accurately (but see next point) captures the behavioural pattern, but I think what is missing is a parameter recovery analysis.

4b) The model validation recapitulates the behaviour, barring the exception that the model shows significantly reduced $p(\text{stay})$ for the go response in no reward vs no punishment. This is not in the behaviour (if anything, there is the opposite pattern, Fig 2C vs 2G).

4c) Did the softmax choice rule not contain a temperature parameter? If not, why so?

4d) Line 556: "We assigned cue valence V_s to 0.5 for Win cues and -0.5 for Avoid cues and used cue valence scaled by participants' individual feedback sensitivity as initial action values Q_0 . Unlike previous versions of the task, cue valences were not instructed, but had to be learned from outcomes, as well".

I do not understand this. If cue valence was not instructed, but had to be learnt from experience, why then are they set to fixed values? And the initialization of the Q-values is set to feedback sensitivity ρ ? But then this parameter cannot be independently fit?

4e) Similarly with value initialization and "unmuting" of action values after the first trial? If values are initialized with zero, softmax $p = 0.5$, prior to any experience, no?

4f) line 187 "...to compute trial-by-trial belief updates ($PE \cdot \text{learning rate}$)..." It was not clear to me whether the PE was used as regressor of interest or the product $PE \cdot \text{learning rate}$. If the latter, why? (Effectively, it would merely scale each trial's PE with a subject-specific constant?)

5) Reversal of Go/Nogo responses between time of response to outcome onset - it would be reassuring to see the average BOLD timecourse for Go vs NoGo trials for the relevant interval

Minor

1) Please tone down the title a little. The authors are quite careful in discussing limitations of this study and explicitly point out the need for caution about strong causal statements

2) Figure 2 uses red and green colours - it may be worth considering that quite a fraction of people might have difficulty distinguishing these.

3) In line 307, Fig 6E is referenced <-- 6E does not exist, I think this should read 6D

4) In line 59, it reads:

"On the other hand, biases might be responsible for phenomena of "animal superstition" like [...], where rats and pigeons repeat behavioral patterns that co-occurred with the attainment of [...] rewards and keep showing such behavior even if it delays or decreases rewards"

I would argue that this does not necessarily relate to the motivation bias the authors are focussing on, but rather a simple spread of credit for a reward to (temporally) proximal actions.

Reviewer #2 (Remarks to the Author):

Algermissen et al. present a technically impressive study combining EEG and fMRI to examine the relationship between subcortical and cortical regions in mediating motivational biases. This is the second part of a larger research project: the authors have used the same data to answer different questions. The first part has already been published in another journal (Algermissen et al., *Cerebral Cortex*, 2021). While the data and analyses of the neural data is the same (the authors collected behavioral and multimodal neural data on the same participants and used trial-by-trial BOLD or EEG signal to informed EEG or BOLD respectively), some aspects of the methodology are new (like the use of RL model to fit behaviors) and the questions are also novel (they look at outcome phase instead of the decision phase).

Most analyses are sophisticated and the results are carefully described. However I think there remains substantial work to do to 1) properly situate the current study in light of their previous work and previous literature and provide clarity for the reader on the novel aspect of this work and the key question that the paper is seeking to answer and 2) clarify that the EEG oscillation and the brain network identified for PE_bias are meaningful.

Major points:

1-Background: Previous work have shown that the network encompassing the ACC, ant

insula etc rises before the reward network at time of outcome. Previous findings also showed that motivational response biases are first visible in the vmPFC before they modulate striatal action selection (including their own work!). Perhaps it would be better to clarify the link with previous papers and especially the ones from the same group and explain how this one differ and also, importantly, how they fit together. Can we also lines between decision and outcomes phases as similar vmPFC-striatal interactions are at play? This is lacking. Similarly, it might be good practice to build on previous work to emphasize their complementarity and support to similar theories.

2-Narrative: While I appreciate that talking about "old" and "younger" parts of the brain is useful for a simple story, this narrative is quite problematic. Though widely shared in introductory neuroscience textbooks, the ideas that, as animals evolved, "newer" brain structures were added over existing "older" brain structures, and that these newer, more complex structures endowed animals with newer and more complex cognitive functions, behavioural flexibility, and so on, have long been discredited among neurobiologists. This stands in stark contrast to the clear and unanimous agreement on these issues among those specialising in neuroscience (see the work of Deacon, Northcutt or Eisthen). In short, the idea that layers were added to existing structures over evolutionary time as species became more "complex" is simply incorrect. This would have to be changed throughout the whole manuscript.

3-EEG oscillation: As the authors know, power at a frequency in a time-frequency transform does not mean that an oscillation was present at that frequency. Any waveform will have a time-frequency equivalent, whether or not the neural data contains any oscillation. As a rule of thumb, a broad band of power is not a true oscillation. I am therefor a bit doubtful that the one present in fig 4 for the PE_bias reflects a true oscillation for example. Is it possible to see the y axis starting at 0? Similarly for the alpha and theta power in figure 5 for the ACC and PCC? The authors would have to show that the neural data do contain neural oscillation (maybe in the time domain) before making claims about their neural generators. Related to the previous comments, is it possible to see classic ERP at outcome as a sanity check to show that the EEG is of sufficient quality after GA and BCG artefact corrections?

4-Neural model comparison: the authors claim that “Striatal BOLD was better described by biased PEs than by standard PEs “ but they do not test for it. One way to do this is to run some model comparison at the neural level. Ideally, they want to show that, within the network they find in their first GLM, the one having positive versus negative outcomes (so that the ROIs are selected to favour the hypothesis that they want to reject), a GLM containing the PE_bias (in its full form, it doesn’t need to be split) provides a significantly better account of the network’s neural activity than the one with the pos/neg outcomes and the one with a simple PR. Bayesian statistics are a great way to do this. This would convince the reader that the last model is indeed the one to consider. A good example of this is from this paper: Palminteri et al. 2015, Nature Comm.

Other points:

1-Title: the title will have to be changed in accordance with point 1&2.

2-RL Model 6: The authors show an important model in supplementary materials (M6) but do not present it at all in the text. This seems important! In fact, the winning model fit to behavior is not convincing (Figure 2g the bright green and pink colours) and indicate a poor generative performance. An alternative model accounts for this specific behavioral effect is of interest but is not presented in the methods neither in the results. I would suggest to show it to the results parts, and explain clearly why this model has not be chosen.

3-Relation between the vmPFC and striatum: I can see that using both the EEG and fMRI to carefully tease apart temporal and spatial neural signatures of motivational biases opens up lots of interesting questions. But for the current focus on the vmPFC and its relationship with the striatum (e.g. the vmPFC “influences” the striatum), I don’t see how the authors, from their results can make the claim that the vmPFC mediates what is happening in the striatum, aside showing that it arises earlier. Since they do not use causal manipulation or directional methods, the data cannot support any conclusion on the direction between the vmPFC and Striatum. While the authors acknowledge it in the discussion, the title would have to be rephrase.

Typos and other minor details:

- line 55: One the one hand

- Caption Figure 2: orange and red bits (green and light green) on panel B are not distinguishable in the legend

- line 179: what is the reference to S04 presenting here? Must be a typo

Palminteri, S., Khamassi, M., Joffily, M. et al. Contextual modulation of value signals in reward and punishment learning. *Nat Commun* 6, 8096 (2015).

<https://doi.org/10.1038/ncomms9096>

Reviewer #3 (Remarks to the Author):

The authors present an empirical study investigating the neural mechanism of motivational bias in reinforcement learning. The authors use simultaneous EEG and fMRI to track the order of the encoding of biased prediction error, and use reinforcement learning models to characterize the motivational bias in learning. They provide evidence of biased prediction error in a set of striatal and cortical regions using fMRI, as well as in midfrontal power using EEG. They link both fMRI and EEG data and identify unique contributions of fMRI BOLD in anterior and posterior cingulate, striatum to EEG power in midfrontal electrodes, and argue that cortical influences precede striatal activity in driving the motivational learning bias.

Overall, the methods in this paper are sound, building on a well-validated task and associated models, and the authors have performed an extensive number of careful analyses. Generally speaking, the use of model-based analyses to examine simultaneous EEG-fMRI data is rare and exciting, but in the case of this paper that novelty is diminished somewhat by the fact that the authors have already published another paper doing something very similar within the exact same dataset (albeit examining largely orthogonal elements of the task). Having said that, I think the findings presented in this paper could be of interest to researchers studying motivation, learning, and decision making, including the demonstration that a model accounting for a Pavlovian bias provides a good account of both fMRI and EEG signatures of RL updating; and the careful unraveling of the time-course of motivationally biased stimulus evaluation, response preparation and execution, feedback

processing, and policy updating. Questions of novelty aside, my main concerns, unpacked below, pertain to the overarching interpretation of these findings as reflecting the prefrontal biasing of striatal learning, and to their localization and interpretation of specific prefrontal signals putatively related to policy updating.

Major comments:

1. Overall, the evidence supporting the conclusion that the biased prediction error occurs in cortical regions before subcortical regions seems rather tenuous. In Figure 5, the authors tried to convey that the EEG correlates of PFC precede the correlates of striatum, so that cortical regions encode biased prediction error earlier than subcortical regions. However, the EEG correlates of ACC are mostly in the lower alpha band, which does not correlate with the magnitude of PE_biased. Moreover, based on Figure 5D, it seems that correlates of PCC increase at about the same time with striatum. All of these findings might indicate that ACC and PCC are involved in the outcome process earlier than striatum, not necessarily that these two areas encode the bias signal earlier than striatum.
2. Related to the previous point, how much of the difference in latency of EEG signals between striatal and cortical regions can be accounted for by their relative proximities to the recordings at the scalp? I'm wondering in particular as pertains to the regressions that regress EEG activity on activity in cortical and subcortical ROIs. It would be helpful if the authors can provide some information to rule this out either a priori or with relevant analyses (or otherwise adjust their interpretation accordingly).
3. The claim that ACC maintains the eligibility trace of the previously performed response (p. 16), would benefit from a formal test of this hypothesis, showing that the eligibility trace from the fitted RL model provides an adequate account of activity in this region. The same is true for the proposed link between vmPFC/PCC and action policy updating – a stronger test would be to explicitly link the neural activity to individual-specific changes in the likelihood of one action or another.
4. Related to the previous point, to what extent can outcome-related responses to positive or negative feedback reflect changes in policy certainty? For instance, the authors found that midfrontal theta was associated with negative feedback and increased likelihood of switching policies. Could this reflect an association between theta and policy uncertainty (cf. Cavanagh & Frank, 2015, Trends Cog Sci) rather than policy updating per se.

5. The authors initially refer to the vmPFC region as vmPFC/perigenual ACC (32d), and to the ACC region as ventral ACC (23/24), but transition to referring to these as vmPFC and ACC. To avoid confusion with other regions of PFC, I would recommend continuing to refer to these in ways that reference their common connections to regions of cingulate. This is relevant to their finding that vmPFC appeared to track theta activity, which past literature has linked to regions of the cingulate. It is also relevant context for discussion of evolutionary age (e.g., p. 19), which is still germane for cingulate and striatum, but with a different connotation than if vmPFC referred to a region like BA10.

6. The EEG-fMRI analyses are overall quite dense and difficult to follow, and require additional clarification. Perhaps a figure could be useful for visualizing the rationale and/or predictions for the EEG-informed fMRI and fMRI-informed EEG analyses. A few specific questions about these analyses:

a. p. 12. “biased PEs were controlled to capture additional variance in EEG explained by BOLD signal beyond task regressors”. Can the authors clarify why the unique variance explained by brain activity *_after_* controlling for biased PE is still informative regarding the order of biased PE processing?

b. P13-14. Why are vmPFC and PCC included in the ‘reverse check of the link’ analyses but the same is not being done for all of the EEG signatures in the opposite analysis (for example, ACC)? Additionally, it not clear that simply reversing the regression model would necessarily provide evidence to combine PCC and vmPFC in the follow-up analysis (e.g., fMRI ~ EEG does not necessarily provide the same interpretation as EEG ~ fMRI). The authors should provide a more thorough rationale for combining vmPFC and PCC based on this reverse check. Specifically, it is unclear why it is reasonable and necessary to perform the reverse check to include new areas.

c. The authors stated that ‘this analysis suggests dlPFC and SMG as likely candidate sources’. What is the rationale for this? The authors should provide additional evidence to support this assignment of candidate sources (either from the current dataset or existing literature), or simply remove this speculation from the text.

Minor comments:

1. P. 11, “Only the sum of both terms...was significantly encoded...” - The authors should be cautious when interpreting this and related findings. The fact that PEstd ($p=0.074$) and PE_{dif}

($p=0.185$) were not significantly associated with delta power but PEbias was ($p=0.017$) does not on its own imply that PEbias provided a reliably better account of delta power than the other two. To infer otherwise would require a direct contrast of these (e.g., R-sq or BIC change).

2. P. 36. “these measures were created by using the 3-D (time-frequency-channel) t-map obtained when contrasting positive vs. negative outcomes (theta/ delta and beta) and Go vs. NoGo conditions (lower alpha band) as a linear filter.”-- please include details about this analysis (e.g., in the supplement).

3. Please clarify how the hyperpriors in Model M5 were chosen.

4. It would be helpful to provide an intuition for why the presence of two epsilon values in Equation 6 ensured that the effect of kappa on epsilon was symmetrical.

Legend of reviewer comments, responses, and textual changes to the manuscript:

Bold	= reviewer comment
Normal	= our response
“normal”	= citations from manuscript (if entire paragraphs of text)
Blue	= changes in manuscript
Highlight	= header of entirely new sections in the supplemental materials

%%%

Reviewer #1 (Remarks to the Author):

The manuscript by Algermissen and colleagues presents a study combining simultaneous EEG-fMRI during a learning task with computational modelling of the learning/choice behaviour. They address the question of where biases in (a) action selection (b) instrumental learning arise. Here, (a) refers to the observation that agents are more adept at performing vs withholding a response to obtain a reward and, conversely, better at suppressing vs performing an action to avoid punishment. (b) refers to agents more readily assigning credit for a reward to action vs inaction, and more readily assigning credit for avoiding punishment to inaction vs action. The central question of this manuscript is whether such biased learning arises in the striatum, or instead in the prefrontal cortex which then biases learning in the striatum.

The authors find fMRI correlates of biased prediction errors (PE) in (amongst others) ventromedial prefrontal cortex and striatum. In the EEG, there is a representation of biased PE sign in mid-frontal theta and beta power. Using fMRI-informed EEG, they show that fluctuations of the trial-by-trial BOLD signal in prefrontal cortex are represented in (mostly mid-frontal) EEG before the emergence of striatal contributions. Based on this, they argue that biased learning arises in the frontal cortex, not in the striatum.

Overall, this is clearly an impressive study that must have been a tour-de-force to conduct and analyze. I am not entirely convinced yet that the approaches taken by the authors can really answer the central question of whether the bias originates in the striatum or prefrontal cortex (see my first major point below). However, even if the answer to this question should be negative, I think one should ask whether this study does not still provide enough important results that would be important in their own right. I am inclined to think this would be the case. This however rests on whether or not a few methodological questions that I have (particularly about the "biased PE") can be addressed/answered.

We thank the reviewer for their positive evaluation and thoughtful comments, which we believe we all address in the specific points below.

1) Regarding the fMRI-EEG analyses presented in Fig. 5. I am not convinced this informs us about whether the relevant computations occur first in striatum or cortex. My understanding is that the authors extracted each trial's BOLD signal from the relevant ROI and then, for each trial, fit an HRF to this signal, yielding a coefficient for each trial and ROI. This coefficient represents the BOLD signal on each trial, however, does not contain specific information about the PE. These coefficients are then regressed against the EEG signal. Thus, to me it appears that the results merely indicate when and at what frequency the EEG correlates with striatal BOLD following outcome presentation, independent of any particular PE representation. It also seems to be the case that the correlations between BOLD and EEG (Fig 5) emerge at very short latency (almost immediately after outcome presentation) compared to the representations of PE sign in (Fig 4), which seem to occur considerably later.

I also think it might be more sensible (Fig 5D) to use the latency to first emergence of a significant correlation. Thus, currently I would say that all we can say with confidence from this analysis is that ACC and PCC correlate with cortical EEG at an earlier time point and at different frequency band compared to striatum, which may simply reflect the general anatomical connectivity. To resolve

this, maybe it would be possible to incorporate the PE as a covariate in these analyses? (However, see my concern below about the "biased PE").

First, we would like to reassure the reviewer that we indeed had already included PEs as covariates in our fMRI-informed EEG analyses, as we now also explicitly highlight now:

In the Results section in lines 291-297:

“We performed analyses with and without PEs included in the model, which yielded identical results and suggested that EEG-fMRI correlations did not merely result from PE processing as a “common cause” driving signals in both modalities. Instead, EEG-fMRI correlations reflected incremental variance explained in EEG power by the BOLD signal in selected regions (even beyond variance explained by the model-based PE estimates) providing the strongest test for the hypothesis that BOLD and EEG signal reflect the same neural phenomenon.”

In the Methods section in lines 901-905:

“All results were identical with and without the inclusion of PEs as covariate in the regression [...].”

Given that the regressors in these analyses (trial-by-trial BOLD from various regions) were selected *based on the fact* that they encoded biased PEs, it is not surprising that these regressors already capture much of the variance to be explained and the addition of the PEs as covariates does not change the results.

We fully agree with the reviewer that our primary conclusion is that “ACC and PCC correlate with cortical EEG at an earlier time point and at different frequency band compared to striatum”. Together with our finding that ACC, PCC and striatum (unlike other regions) all show specific signatures of biased learning, we conclude that biased learning occurs earlier in cortical regions (ACC/ PCC) than in the striatum. If we understand the reviewer correctly, their primary concern is that EEG-BOLD correlations in cortical regions might be induced by some other (“unspecific”) neural event(s) that occur in these regions. If this was true and these unspecific events would occur in parallel or even later than biased PE processing (i.e., aiding or resulting from biased PE processing), our conclusion that biased PE processing occurs earlier in PFC than in the striatum would still hold. However, if they occurred (much) earlier than the biased PE processing (i.e. reflecting a yet “incomplete” processing step), our conclusion might have to be changed. Under such circumstances, biased PE processing in cortex would occur relatively late and most likely play an “epiphenomenal” role, which, as we argue below, is highly unlikely given the single cell literature on cortical responses to outcomes. We have now modified the discussion section to further elucidate and support our conclusions. Also, we discuss why peak timing is a more suitable measure to relate the relative timing of regions to each other and why the observed correlation are likely transient and task-induced, rather than resting-state-like/ reflecting general anatomical connectivity.

First, the reviewer suggests that the trial-by-trial estimates for the fMRI informed EEG analysis do not reflect a particular PE representation. We disagree since we selected the regions for the fMRI-informed EEG analyses (Fig. 5A-C) based on the specific signature of biased learning as predicted by our winning computational model (Fig. 3C). In contrast, we did not select regions that only encode positive vs. negative outcomes (Fig. 3A) or “standard” prediction errors (Fig. 3B). This selection criterion therefore ensures that these regions indeed encode specifically biased prediction errors. We agree with the reviewer that, beyond this specific biased-learning signature, the trial-by-trial BOLD signal likely reflects additional neural processes that occur during outcome presentation. The reviewer’s concern seems to be that a) such additional processes might dominate neural activity in the first 500 ms after outcome onset and be visible in the EEG signal, and that b) only later, PE

processing comes to dominate neural activity, which then drives the major trial-by-trial variation in the BOLD signal. In this scenario, the timing of EEG correlates of PFC BOLD is dissociated from the timing of biased PE processing, rendering our conclusion invalid. However, as a consequence of this scenario, biased learning processes would occur so late that they might at best play an epiphenomenal role. This scenario is incompatible with a rich literature of single cell recordings from cortical cells (e.g. OFC), which has shown that responses to outcomes occur largely in the first 500 ms after outcome onset (e.g. Rolls et al., 1996, Fig. 2; Morrison & Salzman, 2009, Fig. 6). Hence, also (biased) PE processing will most likely occur in this time range. Furthermore, we know from lesion studies that the PFC is causally involved in learning/ credit assignment and unlikely to play a late, “epiphenomenal” role (Takahashi et al., 2009; Walton et al., 2010; Noonan et al., 2010). In sum, as the most likely scenario, the majority of prefrontal activity reflecting outcoming processing happens in the first 500 ms after outcome onset, as corroborated by previous single cell literature and by our own findings. This (rather unitary) outcome processing step is already “biased” in the sense of biased PEs predicted by our winning computational model. The alternative that prefrontal outcome processing falls apart into several, temporally distinct processing stages, of which one drives EEG correlates and another is biased, seems very unlikely.

We also now discuss this previous literature supporting an early, causal role of PFC in outcome processing in the Discussion section in lines 446-448:

“These early EEG correlates of PFC BOLD are in line with single cell recordings in PFC which show responses confined to the first 500 ms following outcome onset (Rolls et al., 1996; Morrison & Salzman, 2009), corroborating that PFC outcome processing occurs before the time of EEG correlates of striatal BOLD.”

Second, the reviewer remarks the potential discrepancy of the “*very short latency*” of ACC-alpha correlations (Fig. 5A) and the EEG theta representations of PE sign/ outcome valence (Fig. 4 A; changed to “PE valence” for clarity throughout the manuscript). Here, it is important to note that the ACC-alpha correlations (100–575 ms) temporally “envelop” the PE valence-theta correlations (225–475 ms), which we believe is consistent with the idea that ACC processes have (at some time point) access to information about the valence.

Furthermore, it is also important to keep in mind that power cannot be estimated instantaneously, but is estimated over a temporally extended time window (here 400 ms, which gives 2.5 Hz frequency resolution; the use of Hanning tapers, which have a Gaussian shape, ensures that the power estimate is dominated by the center of the time window). Given this estimation over a temporally extended time window, the “latency to first emergence of a significant correlation” is likely a suboptimal measure for inferring timing relations in time-frequency data (unlike time-domain or spike data). In contrast, the timing of the peak of such correlations can be estimated more robustly (i.e., is less influenced by potential outliers) and is likely more meaningful.

We now also clarify why we rely on the timings of the peaks in the Results section in lines 343–347:

“This temporal dissociation was especially clear in the time courses of the test statistics for each region (thresholded at $|t| > 2$ and summed across frequencies), for which the peaks of the cortical regions preceded the peak of the striatum (Fig. 5D, H). Note that time-frequency power is estimated over temporally extended windows (400 ms in our case), which renders any interpretation of the “onset” or “offset” of such correlations more difficult.”

Third, these EEG-fMRI correlations are unlikely to reflect “general anatomical connectivity” (as would be reflected in e.g. “resting state” data), because they occur in different frequency bands depending on the current neural process and they fluctuate over time. Here, we report that striatal BOLD signal during outcome processing correlates with midfrontal *beta* power, while we previously found that striatal BOLD signal during action selection correlate with midfrontal *theta* power (Algermissen et al.,

2021). Also, correlations between ACC/ PCC/ striatal BOLD signal and alpha/ theta/ beta power were relatively transient compared to the correlation between motor cortex BOLD and beta power (Supplementary Fig. 17), which seems a more temporally continuous relationship.

We now discuss this process-specificity in the Discussion section in lines 482–490:

“Notably, EEG correlates of striatal BOLD during outcome processing were in the beta band—in contrast to previously observed correlates of striatal BOLD during action selection in the theta band (Algermissen et al., 2021). This dissociation suggests important differences in the roles of the striatum during these two processes. The frequency-specific nature of these EEG-fMRI correlations further suggests that they are signatures of task-induced events that are specific to the trial phase and unlikely to reflect general anatomical connectivity. In sum, while the EEG-fMRI findings on outcome processing resemble our previous EEG-fMRI findings on action selection in that prefrontal signals precede striatal signals, they are dissociated in terms of the frequency specificity, highlighting the distinct roles of the striatum during these two processes.”

2) I am skeptical about the PE_bias. As the authors highlight, the PE from this (the winning) model is correlated at $r = 0.92$ with the PE from a standard Rescorla-Wagner model (PE_std).

To remedy this issue, they compute a PE_diff as

$$\text{PE_diff} = \text{PE_bias} - \text{PE_std}$$

This is also used as reasoning for the analysis presented in Fig. 3 (line 195): "We tested for biased PEs PE_BIAS by computing which regions significantly encoded the conjunction of both its components, i.e., standard prediction errors PE_STD and the difference to biased PEs PE_DIF."

I am not convinced this solves the issue. The PE in the model arise from different learning mechanism, can one say that the PE_diff is simply the PE_std plus something else? It does not sound convincing to me to use the PE_diff as a proxy for the PE_bias, but maybe I am getting things wrong here. I am wondering whether it would not be more appropriate to orthogonalize for (regress out the contributions of) the PE_std, although I appreciate that, given the high correlation, it may be difficult to still detect anything.

We believe there might be a misunderstanding here and would like to reassure the reviewer that our analysis strategy does exactly as the reviewer suggests: we decompose PE_{BIAS} into PE_{STD} and PE_{DIF} , which are (almost) orthogonal to each other (mean correlation of -0.02 across participants). In a GLM containing both terms, PE_{STD} will capture the variance explained by standard PE updating (this variance is “regressed out”), and the question is whether, in a given region, PE_{DIF} still explains additional variance on top of PE_{STD} . Then, rather than using PE_{DIF} as a proxy for PE_{BIAS} , we use the **conjunction** of PE_{STD} and PE_{DIF} (which sum up to PE_{BIAS}) as a test of which regions show significantly better encoding of PE_{BIAS} than PE_{STD} . Dissociating two alternative learning signals by decomposing one into the other plus a difference term is a long-established procedure (Wittmann et al. 2008; Daw et al. 2011; Eldar and Niv 2015) which has for example been used to dissociate correlates of model-based and model-free learning updates in the two-step task, which also tend to be highly correlated (Daw et al. 2011).

We have now clarified this further in the Results section, lines 223–228:

“We tested for biased prediction errors PE_{BIAS} by testing which regions significantly encoded the conjunction of both its components, i.e. significant encoding of both PE_{STD} and PE_{DIF} . Dissociating two alternative learning signals by decomposing one into the other plus a difference term is an established procedure to disentangle the contributions of two highly correlated signals (Wittmann et al. 2008; Daw et al. 2011). Significant encoding of both components (with the same sign) provides strong evidence for encoding of biased prediction errors PE_{BIAS} .”

3) Representation of biased PE in the EEG signal (Fig. 4). I would argue that what is shown here is not a representation of PE_bias (magnitude), but more likely of valence, as the regressor that is

shown here is the sign regressor? Then, in 4D the correlation of PE_bias with delta is shown - however the authors have argued (rightly so I think) above that this regressor is hard to interpret given the high correlation with PE_std. Accordingly, I also think the statements about delta power and PE magnitude do not hold (line 232/233: "In sum, ...delta power encoded PE magnitude (positively)."; line 420: "...and delta encoded PE magnitude").

We apologize for any confusion caused: Fig. 4A shows PE valence (positive vs. negative PEs, i.e., better or worse than expected), which is the "signed" component of the prediction error. We now term this variable "PE valence" throughout the manuscript, to avoid further confusion. *PE valence* is negatively encoded in the theta range and positively encoded in the beta range (Fig. 4A). In contrast, Fig. 4D-F display the *PE magnitude* regressor, i.e. the *absolute* prediction error, which is a measure of surprise. Note that the PE valence regressor is included in these regressions, and thus controlled for. PE magnitude as computed under biased learning is positively encoded in the delta range, which is illustrated in Fig. 4D. We hope this clarifies Figure 4 and associated conclusions.

We clarify this now in the Results section in lines 243–250:

"Note that differently from BOLD signal, EEG signatures of learning typically do not encode the full prediction error. Instead, PE valence (better vs. worse than expected) and PE magnitude (saliency, surprise) have been found encoded in the theta and delta band, respectively, but with opposite signs (Cavanagh, 2015; Talmi et al., 2013; Hauser et al., 2014). When testing for parametric correlates of PE magnitude, we therefore controlled for PE valence, thereby effectively testing for correlations with the absolute PE magnitude (i.e. degree of surprise). Note that PE valence was identical for standard and biased PEs. Thus, only PE magnitude could distinguish both learning models."

and in lines 260–265:

"When testing for correlates of PE magnitude, we controlled for PE valence given that previous studies have reported TF correlates of both PE valence and PE magnitude in a similar time and frequency range, but with opposite signs (Cavanagh, 2015; Talmi et al., 2013; Hauser et al., 2014). Midfrontal delta power was indeed positively correlated with the (summed) PE_{BIAS} term (225–475 ms; $p = .017$; Fig. 4D). Decomposition of the PE_{BIAS} into the constituent terms showed that this correlation was not significant for the PE_{STD} term ($p = 0.074$, Fig. 4E) nor for the PE_{DIF} term ($p = 0.185$; Fig. 4F)."

We also now further clarify in the caption of Fig. 4:

A. Time-frequency plot (logarithmic y-axis) displaying higher theta (4–8 Hz) power for negative (non-reward for Win cues and punishment for Avoid cues) outcomes and higher beta power (16–32 Hz) for positive (reward and non-punishment) outcomes. This contrast reflects EEG correlates of PE valence (better vs. worse than expected).

D-F. Correlations between midfrontal EEG power and model-based trial-by-trial PE magnitudes, controlling for PE valence (thus effectively testing for correlates of "absolute" PEs). Panel D displays the correlates of biased prediction errors PE_{BIAS} , which are decomposed into (E) PE_{STD} based on the non-biased learning model M1, and (F) their difference PE_{DIF} ."

4) I have a few questions about the modelling:

4a) The authors have performed a model validation and show that the model quite accurately (but see next point) captures the behavioural pattern, but I think what is missing is a parameter recovery analysis.

We now present a parameter recovery analysis in the new Supplementary Note 6 and Supplementary Fig. 5. Summarizing this section, parameter recovery for ρ , ϵ_0 , b , and π was excellent (correlation between true and recovered parameters $> .90$). Parameter recovery for κ was not quite as good (partly due to an outlier), yet still strongly positive ($\rho > .50$). More importantly, however, recovery for the

biased learning rates (combining ϵ_0 and κ to $\epsilon_{\text{rewarded Go}}$ and $\epsilon_{\text{punished NoGo}}$, which determines the effective learning rate on each trial) was very good ($> .85$ when excluding one outlier for $\epsilon_{\text{punished NoGo}}$, see below). Recovery became even better when adding other parameters to the model (e.g., the perseveration parameters discussed in Supplementary Note 8 in response to point 4b).

Supplementary Figure 5. Parameter recovery results for the asymmetric pathways (M5) model. The feedback sensitivity parameter ρ (A), the baseline learning rate ϵ_0 (B), the Go bias b (C), and the Pavlovian response bias π (D) all showed excellent parameter recovery, i.e., between-participants correlations of ground-truth and fitted parameters all exceeded $r > 0.90$. Parameters ρ and ϵ_0 are still in sampling space and thus untransformed (which means they can be negative). Dashed lines represent the identity line; red solid lines represent a linear regression line of fitted parameters regressed onto true parameters. Only recovery of the learning bias parameter κ (E) was not quite as good, though the correlation between ground-truth and fitted parameters was still strongly positive ($r > 0.50$). Note an outlier at the bottom left of κ values; the regression line was fitted without this data point. When combining the baseline learning rate ϵ_0 with the learning bias κ to compute the biased learning rates for rewarded Go actions $\epsilon_{\text{rewarded Go}}$ (F) and punished NoGo actions $\epsilon_{\text{punished NoGo}}$ (G), correlations between ground-truth and fitted parameter values were considerably higher (r 's > 0.86). Note again an outlier at the top right of for $\epsilon_{\text{punished NoGo}}$ values; the regression line was fitted without this data point.

We also add the following conclusion to this Supplemental Note 6 lines 429–441:

“In sum, parameter recovery was excellent for all parameters but the learning bias κ . More relevant than recovery of κ , however, was that we could recover the effective learning rate well (combining baseline learning rate ϵ_0 and the learning bias κ). Note that the ability to accurately capture individual differences in biased learning is not of interest in this study, nor relevant to the imaging analyses. In fact, we used a single set of parameters (the group-level parameters) to compute trial-by-trial regressors for the EEG and fMRI analyses. This is a standard approach in model-based fMRI for two main reasons. First, it has been shown that the exact parameter values for relatively simple RL models like the ones used here have little impact on results of fMRI analyses (Wilson and Niv 2015). For the current study, of most relevance is the qualitatively differential pattern of learning updates after Go and NoGo responses (Nassar and Frank 2016; Palminteri et al. 2017; Wilson and Collins 2019), as embodied by the algorithmic specification of the model. This pattern drives the EEG and fMRI results and indeed, using a different set of parameter values, we obtain essentially identical fMRI results (see Supplementary Note 9 and Supplementary Fig. 8).

4b) The model validation recapitulates the behaviour, barring the exception that the model shows significantly reduced $p(\text{stay})$ for the go response in no reward vs no punishment. This is not in the behaviour (if anything, there is the opposite pattern, Fig 2C vs 2G).

Fitting a limited set of parameters (< 6 in our models) to many data points ($n = 640$ trials in our case) implies a strong data reduction. Hence, the limited representations of such models can never capture all aspects of the data perfectly (Wilson & Collins, 2019). We agree with the reviewer that the winning model M5 predictions do not exactly replicate the $p(\text{stay})$ pattern observed in the data, but fails to capture two features of the data:

- 1) M5 predictions underestimate the overall propensity to repeat a response to the same cue
- 2) as the reviewer observes, M5 predicted a higher probability to repeat a Go response after non-punished Avoid cues, relative to non-rewarded Win cues. In contrast, in the data, there was (perhaps surprisingly) no such significant difference, and numerically, participants were even more likely to repeat a Go response for a non-rewarded Win cue than for a non-punished Avoid cue (however, this difference was not statistically significant).

We hypothesized that two potential mechanisms could account for these data features, and present three new models in Supplementary Note 8. Below, we summarize these additional analyses:

Mechanism 1: overall response stickiness (i.e., a tendency to repeat action irrespective of the obtained outcome), potentially with differential strength for Win and Avoid cues. We fitted two new models to capture these features.

M7: *single perseveration*: like M5, but with a single extra perseveration parameter ϕ

M8: *dual perseveration*: like M5, but with two separate perseveration parameters, one for Win cues ϕ_{WIN} and one for Avoid cues ϕ_{AVOID}

Mechanism 2: We reasoned that, perhaps, the “Win” or “Avoid” frame of a cue itself could reframe the interpretation of the (neutral) outcome in a positive / negative light. To capture such an effect, we added a parameter to the outcome term for the neutral outcomes (represented as 0), which added/subtracted a bonus/ malus from 0 based on the cue valence. Note that it would be superfluous to add this term for the valenced outcomes (rewards and punishments represented as +1 and -1), as such an effect would already be captured by the reward sensitivity parameter p .

M9: *neutral outcomes reinterpretation*: like M5, but neutral outcomes are modulated by the cue valence V scaled by parameter η that determines the degree of reframing. In addition, given the strong evidence for perseveration (from model M7), we also added a single extra perseveration parameter ϕ to capture overall response stickiness.

Note that models M7-9 reduce to M5 when the values of the additional parameters are zero.

The key findings were:

- Quantitative model comparison showed that all three models, and particularly M8, provided a better fit to the data.
- Model simulations with the optimized parameter estimates showed that all models captured the overall pattern of “stay” behavior better.
- However, all of these models strongly underestimated the specific effect of interest that we aimed to capture with our models: the degree to which people showed biased responses, i.e., made incorrect “Go” responses to Win cues. In other words, while their quantitative fit was better, their qualitative fit (ability to capture relevant aspects of the data) was worse.

- When re-analyzing the fMRI and EEG data with regressors based on these new models M8 and M9, results remained essentially unchanged and did not change the conclusions.

We now provide the following Supplementary Fig. 7, which displays the (mis)matches between the data and the model predictions:

*Supplementary Figure 7. Model comparison and validation of the single perseveration (M7), dual perseveration (M8) and cue valence-based outcome reinterpretation models. **First row.** Trial-by-trial proportion of Go responses (\pm SEM across participants) for Go cues (solid lines) and NoGo cues (dashed lines). **Second row.** Mean (\pm SEM across participants) proportion Go responses per cue condition (points are individual participants' means). **Third row.** Probability to repeat a response ("stay") on the next encounter of the same cue as a function of action and outcome. **Fourth row.** Log-model evidence, model frequency, and protected exceedance probability all favored the dual perseveration model (M8) over the other models. In sum, the additional models M7-9 provided a better quantitative fit to the data compared to the asymmetric pathways model M5 reported in the main text. They also predicted the propensity of staying overall more accurately than M5. However, these additional models all overestimated the proportion of incorrect Go responses. Furthermore, although the predicted patterns of the propensity of staying mimicked the data more closely than M5, these predicted patterns still mismatched some aspects of the empirical data. Taken together, these models could capture certain qualitative patterns in the data, but not others, which was expectable given the data reduction that comes with fitting a learning model with few parameters only.*

We conclude in Supplementary Note 8 in lines 558–569:

“In sum, the three additional models provided a better quantitative fit to the data compared to the winning model M5 reported in the main text. Also, these additional models predicted the propensity more accurately than the base models did. However, their qualitative fit (i.e. the ability to capture relevant aspects of the data) was worse: These additional models systematically underestimated the proportion of incorrect Go responses. Furthermore, although the predicted patterns of the propensity to stay matched the data more closely than M5, these predicted patterns still mis-matched some aspects of the data, particularly now over-estimating the tendency to stay following a punishment. Taken together, these models could capture certain qualitative patterns in the data, but not others, which of course is a core feature of computational modelling, as this by definition constitutes a data reduction procedure that necessarily loses some details of the data. In terms of qualitative model validation/ falsification (Nassar and Frank 2016; Palminteri et al. 2017), M5 and M8/M9 capture different qualitative features of the data, but no model captured all features well.”

We summarize these findings in the Results section of the main text in lines 186–192:

“We explored three additional models featuring supplementary mechanisms to account for this behavioral pattern (Supplementary Note 8 and Supplementary Fig. 7). All these models fitted the data well and captured the propensity of staying better than M5; however, these models overestimated the proportion of incorrect Go responses. Model-based fMRI analyses based on these models led to results largely identical to those obtained with M5 (Supplementary Note 9 and Supplementary Fig. 8). We thus focused on M5, which relied on only a single mechanism (i.e., biased learning from rewarded Go and punishment NoGo actions).”

Given that M5 best captured the feature that is the process under study in this paper, and that we have used M5 in previous publications (Swart et al. 2017, 2018), we propose to keep results from M5 in the main text, but report additional fMRI results from M8 and M9 below.

In brief, fMRI results using parameter estimates from M8/ M9 to compute biased PEs fully replicated the regions identified with M5 and displayed in Fig. 3 of the main text. In addition, M9 (the neutral outcomes reinterpretation model) yielded additional correlates of biased PEs in larger regions of vmPFC and PCC. Although not the process under study in this paper, these results tentatively suggest that vmPFC and PCC keep traces of the cue valence up until the time of the outcome, putatively biasing the processing of neutral outcomes.

We added the following Supplementary Figure 8:

Supplementary Figure 8. BOLD correlates of biased prediction errors as predicted by the asymmetric pathways model (M5), the cue valence-dependent perseveration model (M8) and the neutral outcomes reinterpretation model (M9). (A) Regions encoding both the standard PE term and the difference term to biased PEs (conjunction) as predicted from the asymmetric pathways model (M5) at different cluster-forming thresholds ($1 < z < 5$, color coding; opacity constant; replotted from Fig. 3C main text). Clusters significant at a threshold of $z > 3.1$ are surrounded by black edges. This is a version of Fig. 3C reprinted with a color scheme consistent with the other two panels. (B) Regions encoding both the standard PE term and the difference term to biased PEs (conjunction) as predicted from the cue valence-dependent perseveration model (M8) at different cluster-forming thresholds ($1 < z < 5$, color coding; opacity constant). Clusters significant at a threshold of $z > 3.1$ are surrounded by black edges. In line with correlates of biased PEs as predicted by M5, BOLD signal in bilateral striatum, dACC (small-volume corrected), pgACC, PCC, left motor cortex, left inferior temporal gyrus, and primary visual cortex was significantly better explained by biased learning than by standard learning. This finding was not surprising given that adding perseveration to the model did not change the learning mechanism, but only led to slightly different best fitting parameter values. (C) Regions encoding both the standard PE term and the difference term to biased PEs (conjunction) as predicted from the neutral outcomes reinterpretation model (M9). In addition to the regions in which BOLD signal was significantly better explained by biased than standard PEs as derived from M5 and M8, biased PEs derived from M9 also explained BOLD signal in vmPFC (larger cluster than M5), PCC (larger cluster than

M5), left inferior frontal gyrus and multiple clusters in superior and inferior lateral occipital cortex significantly better than standard PEs. These results tentatively suggested that vmPFC, PCC, and these other occipital regions might implement an additional mechanism besides biased learning which encodes the cue valence also at the time of the outcome, biasing the processing of neutral outcomes.

4c) Did the softmax choice rule not contain a temperature parameter? If not, why so?

In line with previous implementations of variants of these models (Guitart-Masip, Huys, et al. 2012; Swart et al. 2017, 2018; de Boer et al. 2019; Perosa et al. 2020; van Nuland et al. 2020), we implemented a feedback sensitivity parameter ρ instead of an inverse temperature parameter. The feedback sensitivity parameter is directly multiplied with outcomes and has an effect highly similar to the inverse temperature parameter: High feedback sensitivity leads to outcomes being processed as more distinct (e.g., values of +10 and -10 with $\rho = 10$ instead of +1 and -1 with $\rho = 1$), leading to more deterministic choices. The advantage of this implementation is that it allows for multiple feedback sensitivity parameters to assess asymmetric processing of positive and negative outcomes (Guitart-Masip et al. 2014; Perosa et al. 2020). Here, we only use a single feedback sensitivity parameter, but we prefer to keep this implementation for consistency and comparability with previous work.

We now further clarify the role of the rho parameter and its relation to the softmax inverse temperature in the Methods section in lines 636–639:

“This model contained two free parameters: the learning rate (ϵ) scaling the updating term and the feedback sensitivity (ρ) scaling the received outcome (i.e., higher feedback sensitivity led to choices more strongly guided by value difference, akin to the role of the inverse temperature parameter frequency used in reinforcement learning models).”

4d) Line 556: "We assigned cue valence V_s to 0.5 for Win cues and -0.5 for Avoid cues and used cue valence scaled by participants' individual feedback sensitivity as initial action values Q_0 . Unlike previous versions of the task, cue valences were not instructed, but had to be learned from outcomes, as well".

I do not understand this. If cue valence was not instructed, but had to be learnt from experience, why then are they set to fixed values? And the initialization of the Q-values is set to feedback sensitivity rho? But then this parameter cannot be independently fit?

Apologies for the confusion, we would like to further clarify the logic behind the initialization values.

Cue valence V : We now clarified this in the Methods section in lines 663–669:

“Participants were instructed that a cue was either a Win cue (affording rewards or neutral outcomes) or an Avoid cue (affording neutral outcomes or punishments). Hence, cue valence (Win/ Avoid) did not have to be learned instrumentally; instead, it could be inferred as soon as participants experienced a non-neutral outcome. Until that moment, cue valence $V(s)$ was set to zero. Afterwards, $V(s)$ was set to +0.5 for Win cues and -0.5 for Avoid cues. Note that choosing different values than 0.5 would merely rescale the bias parameter π (e.g., halving π with cue valences of +1 and -1) without any changes in the model predictions.”

We hope that this clarifies the procedure. Indeed, in the data of this as well as previous studies (Swart et al. 2018), the bias is present immediately following the first non-neutral outcome (and relatively constant afterwards). This pattern mimics data from studies in which the cue valence was explicitly cued (Swart et al. 2017, 2018). Thus, in our models, we set cue valence to “0” for each cue until the first non-neutral outcome was observed, and then set it to a constant value.

Action value Q :

First, we indeed did not aim to independently fit the starting values for Q , as estimates of such starting points are usually highly unstable (as they are based on only a few datapoints).

We now clarified the Q value initialization in the Methods section in lines 641–649:

“We initialized action values Q_0 so that they reflect a “neutral” expected value of each action. Win cues could lead to reward (+1) and neutral (0) outcomes and Avoid cues to neutral (0) and punishment (-1) outcomes. A neutral expected value would assign equal probability to either possible outcome, leading to expectations of $+1/2$ and $-1/2$, respectively. In addition, because participants’ feedback sensitivity parameter ρ reflected how participants weighed the outcomes they received, also the initial values had to be multiplied with the feedback sensitivity to stay neutral between 0 and participants’ positive/ negative outcome of $+/-1*\rho$. Thus, initial action values Q_0 were set to $1/2*\rho$ (Win cues) and $-1/2*\rho$ (Avoid cues).”

4e) Similarly with value initialization and "unmuting" of action values after the first trial? If values are initialized with zero, softmax $p = 0.5$, prior to any experience, no?

Indeed, if values are initialized to the mean, then the probability of each action is equal (though in our case with three response options, it is $1/3$). As discussed above, when a neutral outcome is received, it is unclear whether it should be interpreted as positive (i.e. for the Avoid cues), or negative (for the Win cues), and thus whether it should or should not reinforce the action that led to them. One solution to this model would be to simply not learn from neutral outcomes. However, once participants experience the first non-neutral outcome (reward or punishment), they can potentially “revalue” previous neutral outcomes (e.g., “For this cue, I made a NoGo response and got a neutral outcome; now I made a Go and got a punishment; hence this is an Avoid cue; hence the neutral outcome on the previous trial was in fact positive; hence I should do more NoGo responses in the future”). This revaluation process is implemented by allowing the model to update Q -values for neutral outcomes as if it is aware of the cue valence from trial 1 onwards. However, until participants experience a non-neutral outcome (for a given cue), the Q -values are “muted”, i.e. multiplied by zero and do not affect the choice (Swart et al. 2018). Note that this muting holds for only a couple of trials at most, and so does not affect behavior dramatically.

We explain this muting now in more detail in lines 650–655:

“Unlike previous versions of the task (Swart et al., 2017), cue valences were not instructed, but had to be learned from outcomes, as well (Swart et al., 2018). Thus, until experiencing the first non-neutral outcome (reward or punishment) for a cue, participants could not know its valence and thus not learn from neutral feedback. Hence, for these early trials, action values were multiplied with zero when computing choice probabilities (Swart et al., 2018). After the first encounter of a valenced outcome, action values were “unmuted” and started to influence choices probabilities, retrospectively considering all previous outcomes (Swart et al., 2018).”

4f) line 187 "...to compute trial-by-trial belief updates (PE*learning rate)..." It was not clear to me whether the PE was used as regressor of interest or the product PE*learning rate. If the latter, why? (Effectively, it would merely scale each trial's PE with a subject-specific constant?)

We indeed used the product of PEs and the learning rate computed the trial-by-trial belief updates. If the learning rate was a constant, this would indeed merely scale the PEs. However, for the winning model M5, learning rate is not constant, but varies across action-outcome conditions, to capture the biased learning we aim to study: The learning rate is higher for rewarded Go responses and a reduced for learning punished NoGo responses. This leads to faster learning of Go actions to win rewards, and slower unlearning of NoGo actions to avoid punishments. To study this differential updating as a function of action-outcome combination, we used the total update term (learning rate * PE), rather than ‘just’ the PE.

We now clarify this point in the Methods section in lines 763–767:

“[...] The update term was computed by multiplying the prediction-error with the condition-specific learning rate. As described above, in the winning model M5, the learning bias term κ leads to altered learning from “congruent” action-outcome pairs, with faster learning of Go actions followed by rewards, but slower unlearning of NoGo actions followed by punishments.”

5) Reversal of Go/Nogo responses between time of response to outcome onset - it would be reassuring to see the average BOLD timecourse for Go vs NoGo trials for the relevant interval

We did already provide the average BOLD time courses for Go and NoGo trials for the relevant interval; after restructuring the supplementary materials, this information can now be found in Supplementary Note 15 and Supplementary Fig. 19E/F. To direct attention to this more clearly, we changed the title of this supplementary material section to:

“Go/NoGo difference over time in BOLD signal, alpha power, and beta power”

We hope that this addresses the comment.

Minor

1) Please tone down the title a little. The authors are quite careful in discussing limitations of this study and explicitly point out the need for caution about strong causal statements

We agree and now changed the title to:

“Prefrontal signals precede striatal signals for biased credit assignment to (in)actions”

2) Figure 2 uses red and green colours - it may be worth considering that quite a fraction of people might have difficulty distinguishing these.

Thanks for pointing this out. We changed the green/ red color shades throughout the manuscript (Fig. 2, 3, 4, 5, 6, Supplementary Fig. 1 – 4, 6, 7, 11, 13, 19). We have ensured that the new colors are also distinguishable under various color blindness conditions using <https://www.color-blindness.com/coblis-color-blindness-simulator/>.

3) In line 307, Fig 6E is referenced <-- 6E does not exist, I think this should read 6D

Thanks for spotting this, this reference is to indeed Fig. 6D, we changed it accordingly (now line 366).

4) In line 59, it reads:

"On the other hand, biases might be responsible for phenomena of "animal superstition" like [...], where rats and pigeons repeat behavioral patterns that co-occurred with the attainment of [...] rewards and keep showing such behavior even if it delays or decreases rewards"

I would argue that this does not necessarily relate to the motivation bias the authors are focussing on, but rather a simple spread of credit for a reward to (temporally) proximal actions.

Apologies for the imprecise wording here. The reviewer is right that the classical “animal superstition” experiments are likely driven by a form of spread of credit to proximal actions. However, the two particular studies we refer to (Brown and Jenkins 1968; Williams and Williams 1969) are distinct from these classical experiments and our description was a bit too general to capture their peculiarities.

We now introduce a more detailed description in the Introduction section in lines 68–74:

“[...] On the other hand, biases might be responsible for phenomena of “animal superstition” like negative auto-maintenance. Studies of this phenomenon used strict omission schedules in which rewards were never delivered on trials on which animals showed a Go action (key peck, lever press), but only when animals inhibited responding over a given time period. Still, animals showed continued key picking in such paradigms, which might either reflect a strong “prior belief” that any situation in which rewards were available requires active work to obtain those rewards, or vice versa an inability to attribute rewards to having held back one’s actions (Dayan et al., 2006; Williams & Williams, 1969; Brown & Jenkins, 1968).”

We hope that this clarifies how these experiments relate to the motivational bias that we focus on in this study.

%%%

Reviewer #2 (Remarks to the Author):

Algermissen et al. present a technically impressive study combining EEG and fMRI to examine the relationship between subcortical and cortical regions in mediating motivational biases. This is the second part of a larger research project: the authors have used the same data to answer different questions. The first part has already been published in another journal (Algermissen et al., Cerebral Cortex, 2021). While the data and analyses of the neural data is the same (the authors collected behavioral and multimodal neural data on the same participants and used trial-by-trial BOLD or EEG signal to informed EEG or BOLD respectively), some aspects of the methodology are new (like the use of RL model to fit behaviors) and the questions are also novel (they look at outcome phase instead of the decision phase).

Most analyses are sophisticated and the results are carefully described. However I think there remains substantial work to do to 1) properly situate the current study in light of their previous work and previous literature and provide clarity for the reader on the novel aspect of this work and the key question that the paper is seeking to answer and 2) clarify that the EEG oscillation and the brain network identified for PE_bias are meaningful.

We thank the reviewer for their very positive evaluation of our work, and we believe we have addressed all points raised below.

Major points:

1-Background: Previous work have shown that the network encompassing the ACC, ant insula etc rises before the reward network at time of outcome. Previous findings also showed that motivational response biases are first visible in the vmPFC before they modulate striatal action selection (including their own work!). Perhaps it would be better to clarify the link with previous papers and especially the ones from the same group and explain how this one differ and also, importantly, how they fit together. Can we also lines between decision and outcomes phases as similar vmPFC-striatal interactions are at play? This is lacking. Similarly, it might be good practice to build on previous work to emphasize their complementarity and support to similar theories.

We have now extensively rewritten the Introduction and Discussion sections to further embed the current study with the extant literature, including our own previous studies:

In the Introduction section in lines 81–87, we add a reference to the previous work, highlighting the difference between biasing of action selection versus learning:

“Previous fMRI studies have studied neural correlates of motivational biases in action selection at the time of cue presentation, finding that the striatal BOLD signal is dominated by the action rather than the cue valence (Guitart-Masip et al. 2011; Guitart-Masip, Chowdhury, et al. 2012; Guitart-Masip, Huys, et al. 2012). More recently, we have reported evidence for cue valence signals in ventromedial

prefrontal cortex (vmPFC) and anterior cingulate cortex (ACC), which putatively bias action selection processes in the striatum (Algermissen et al. 2022). The same regions might be involved in motivational biases in learning during outcome processing, given the prominent role of the basal ganglia system not only in action selection, but also learning.”

In the Discussion section in lines 461–464, we discuss commonalities with our previous study:

“[...] In particular, we have previously observed that motivational biases in action selection also arise from early prefrontal inputs to the striatum, as well (Algermissen et al. 2022). Prefrontal influences on striatal processes might thus be a common signature of motivational response and learning biases.”

In contrast, we also now discuss how EEG correlates of striatal BOLD during action selection and outcome processing differ in the Discussion section in lines 487–495:

“Notably, EEG correlates of striatal BOLD during outcome processing were in the beta band—in contrast to previously observed correlates of striatal BOLD during action selection in the theta band (Algermissen et al. 2022). This dissociation suggests important differences in the role of the striatum in these two processes. The frequency-specific nature of these EEG-fMRI correlations further suggests that they are signatures of task-induced events that are specific to the trial phase and unlikely to reflect general anatomical connectivity. In sum, while these EEG-fMRI findings on outcome processing resemble our previous EEG-fMRI findings on action selection in that prefrontal signals precede striatal signals, they are dissociated in terms of the frequency specificity, highlighting the distinct role of the striatum in these two processes.”

We also discuss the previously literature mentioned by the reviewer regarding an “early” valuation network involving the ACC in the Discussion section in lines 465–470:

“The particular subregion of PFC showing the earliest EEG correlates was the dACC. This observation is in line with an earlier EEG-fMRI study reporting dACC to be part of an early valuation system preceding a later system comprising vmPFC and striatum (Fouragnan et al., 2015). The dACC has been suggested to encode models of agents’ environment (Alexander & Brown, 2011; 2018) that are relevant for interpreting outcomes, with BOLD in this region scaling with the size of PEs (Behrens et al., 2007; Meder et al., 2017) and indexing how much should be learned from new outcomes.”

2-Narrative: While I appreciate that talking about "old" and "younger" parts of the brain is useful for a simple story, this narrative is quite problematic. Though widely shared in introductory neuroscience textbooks, the ideas that, as animals evolved, "newer" brain structures were added over existing "older" brain structures, and that these newer, more complex structures endowed animals with newer and more complex cognitive functions, behavioural flexibility, and so on, have long been discredited among neurobiologists. This stands in stark contrast to the clear and unanimous agreement on these issues among those specialising in neuroscience (see the work of Deacon, Northcutt or Eisthen). In short, the idea that layers were added to existing structures over evolutionary time as species became more "complex" is simply incorrect. This would have to be changed throughout the whole manuscript.

We agree that the simple terms such as “older” vs. “younger” brain areas might evoke associations from introductory text books that we do not subscribe to. What we aimed to highlight is that prefrontal cortex is typically proposed as the crucial brain area for temporally extended behaviors, e.g. long-range foraging in primates, which involve forward planning, counterfactual reasoning, abstraction from immediacy (Boorman et al. 2009; Fouragnan et al. 2019). Such abilities should lead to an emancipation from the rigid, “inflexible” learning biases we describe (e.g., enable thoughts associated with counterfactual reasoning such as “Could an inaction also have resulted in a reward?”

or “Would an action have prevented the punishment?”). We now highlight this dissociation, staying away from ‘older’ versus ‘younger’ dichotomies:

Abstract in lines 27–29:

“The neural origin of these biases is unclear; in particular, it remains open whether motivational biases arise primarily from the architecture of subcortical regions or also reflect cortical influences, the latter being typically associated with increased behavioral flexibility and emancipation from stereotyped behaviors.”

In the Discussion section in lines 454–455:

“[...] which argues against biases being purely driven by subcortical circuits.”

In the Discussion section in lines 549–552:

“These results refine previous views on the neural origin of these learning biases, suggesting they might not only rely on subcortical parts of the brain typically associated with rigid, habit-like responding, but rather incorporate frontal inputs that are associated with counterfactual reasoning and increased behavioral flexibility (Boorman et al. 2009; Fouragnan et al. 2019).”

3-EEG oscillation: As the authors know, power at a frequency in a time-frequency transform does not mean that an oscillation was present at that frequency. Any waveform will have a time-frequency equivalent, whether or not the neural data contains any oscillation. As a rule of thumb, a broad band of power is not a true oscillation. I am therefore a bit doubtful that the one present in fig 4 for the PE_bias reflects a true oscillation for example. Is it possible to see the y axis starting at 0? Similarly for the alpha and theta power in figure 5 for the ACC and PCC? The authors would have to show that the neural data do contain neural oscillation (maybe in the time domain) before making claims about their neural generators. Related to the previous comments, is it possible to see classic ERP at outcome as a sanity check to show that the EEG is of sufficient quality after GA and BCG artefact corrections?

If we understand correctly, the reviewer is concerned that differences between reward and punishment processing as well as the modulation of midfrontal voltage by the PE_{BIAS} term might be better described by an ERP (waveform), i.e., changes that are phase-locked to the outcome onset, rather than a time-frequency representation.

We agree that it is meaningful to ask whether differences between positive and negative outcomes as well as encoding of PE magnitude occur in the phase-locked or in the non-phase locked part of the signal. For this reason, we report the non-phase-locked and phase-locked components of the results described in Figure 4 in the Supplementary Materials (Supplementary Notes 10 and 11 and Supplementary Figures 11–13). Briefly, in Supplementary Fig. 11, we present the results of Fig. 4 of the main text, but with the condition-wise average ERPs subtracted prior to the time-frequency decomposition (which eliminates the phase-locked component). We find essentially the same results as in the main text (but delta correlates of PE magnitude are slightly attenuated and thus not significant any more, although still clearly visible in Supplementary Fig. 11). We conclude that PE encoding occurs primarily in the non-phase-locked component of the signal. This finding is in line with a large range of previous literature finding separate correlates of outcome valence and magnitude in the theta and delta range, respectively (Bernat et al. 2011, 2015; Talmi et al. 2013; Cavanagh 2015; Cavanagh et al. 2021). The key power of the time-frequency results is that it reveals that representations of PE valence and magnitude are encoded *at the same time*, but *in different frequency bands*, which likely reflects different neural generators. ERP analyses are *not* able to distinguish these two sub-signals.

We hope that this decomposition into phase and non-phase locked components of the signal convinced the reviewer that the time-frequency results are not purely driven by phase-locked responses. We would further like to reassure the reviewer that that we have been very careful to not refer to “oscillations” in our paper, but rather refer to “power”, as a more neutral description of the signal we assess, as we do not suggest to claim the presence of sinusoidal “oscillations” that continue for several cycles.

Furthermore, the reviewer suggests to present the power for ultra-low frequencies (0–1 Hz). Power for such low frequencies can only be fitted with sine waves of wave lengths spanning > 1 sec.; hence, such power estimates would integrate over multiple seconds and are unlikely to be informative about short-lived events happening in the narrow time after outcome onset. For this reason, starting time-frequency plots at 1 Hz or 2 Hz is common practice (Cavanagh and Frank 2014).

Finally, we would like to note that the correlate of midfrontal power with the PE_{BIAS} term is constrained to the delta band (1–4 Hz), in line with previous literature (Bernat et al. 2015; Cavanagh 2015), and thus not really “broadband”, a term used for phenomena spanning multiple frequency bands.

4-Neural model comparison: the authors claim that “Striatal BOLD was better described by biased PEs than by standard PEs “ but they do not test for it. One way to do this is to run some model comparison at the neural level. Ideally, they want to show that, within the network they find in their first GLM, the one having positive versus negative outcomes (so that the ROIs are selected to favour the hypothesis that they want to reject), a GLM containing the PE_{bias} (in its full form, it doesn't need to be split) provides a significantly better account of the network's neural activity than the one with the pos/neg outcomes and the one with a simple PR. Bayesian statistics are a great way to do this. This would convince the reader that the last model is indeed the one to consider. A good example of this is from this paper: Palminteri et al. 2015, Nature Comm.

We respectfully disagree with the reviewer that we do not test for this claim. In fact our approach to dissociate two alternative learning signals by decomposing one into the other plus a difference term is a long-established procedure (Wittmann et al. 2008; Daw et al. 2011; Eldar and Niv 2015), which has for example been used to dissociate correlates of model-based and model-free learning updates, which are also highly correlated (Daw et al. 2011). When BOLD signal at a certain voxel significantly correlates with the PE_{STD} term, but in addition also significantly correlates with the (orthogonal) PE_{DIF} term (which together with the PE_{STD} adds up to the PE_{BIAS} term), the p -value of the PE_{DIF} term indicates that assuming biased instead of standard learning leads to significantly more explained variance and thus to a better fit. Our whole-brain GLM (equivalent to multiple linear regression) is already set up in a way that it provides an implicit model comparison: For voxels that significantly encode the PE_{STD} term, the PE_{DIF} term is the test of the model comparison.

Other points:

1-Title: the title will have to be changed in accordance with point 1&2.

We agree and have changed the title to:

“Prefrontal signals precede striatal signals for biased credit assignment to (in)actions”

2-RL Model 6: The authors show an important model in supplementary materials (M6) but do not present it at all in the text. This seems important! In fact, the winning model fit to behavior is not convincing (Figure 2g the bright green and pink colours) and indicate a poor generative performance. An alternative model accounts for this specific behavioral effect is of interest but is not presented in the methods neither in the results. I would suggest to show it to the results parts, and explain clearly why this model has not been chosen.

With regard to the alternative model M6, this model is theoretically plausible based on past work (Cockburn et al. 2014), but shows considerably worse fit and predictive performance than model M5 (see Supplementary Note 7 and Supplementary Fig. 6), strongly underestimating biases in the learning curves as well as failing to yield differential updating after Go/NoGo responses in the p(Stay) plots, thus failing to capture two key aspects of the data. Hence, we discuss this model, but believe that presenting such a badly fitting model in the main text would be rather confusing to readers. If the reviewer insists, we will move this section to the main text, but we believe that M6 rather distracts from the main story of this paper.

Having said that, we did take this reviewer's (and reviewer #1's) comments seriously that, while the winning model M5 does capture the overall motivational bias (Fig. 2F), it captures less well the tendency to "stay" (i.e. repeat a response) following a neutral outcome (the originally bright green and pink colors, Fig. 2G).

We therefore evaluated three additional models to explore putative mechanisms that could explain the tendency to stay following neutral outcomes. We hypothesized that two potential mechanisms could account for these data features, and present three new models in Supplementary Note 8. Below, we summarize these additional analyses:

Mechanism 1: overall response stickiness (i.e., a tendency to repeat action irrespective of the obtained outcome), potentially with differential strength for Win and Avoid cues. We fitted two new models to capture these features.

M7: *single perseveration*: like M5, but with a single extra perseveration parameter ϕ
 M8: *dual perseveration*: like M5, but with two separate perseveration parameters, one for Win cues ϕ_{WIN} and one for Avoid cues ϕ_{AVOID}

Mechanism 2: We reasoned that, perhaps, the "Win" or "Avoid" frame of a cue itself could reframe the interpretation of the outcome in a positive / negative light. To capture such an effect, we added a parameter to the outcome term for the neutral outcomes (represented as 0), which added/ subtracted a bonus/ malus from 0 based on the cue valence. Note that it would be superfluous to add this term for the valenced outcomes (rewards and punishments represented as +1 and -1), as such an effect would already be captured by the reward sensitivity parameter ρ .

M9: *neutral outcomes reinterpretation*: like M5, but neutral outcomes are modulated by the cue valence V scaled by parameter η that determines the degree of reframing. In addition, given the strong evidence for perseveration (from model M7), we also added a single extra perseveration parameter ϕ to capture overall response stickiness.

Note that for models M7-9 reduce to M5 when the values of the additional parameters are zero.

The key findings were:

- Quantitative model comparison showed that all three additional models, and particularly M8, provided a better fit to the data.
- Model simulations with the optimized parameter estimates showed that all models captured the overall pattern of "stay" behavior better.
- However, all of these models *underestimated the crucial effect of interest that we aimed to capture with our models*: the degree to which people showed biased responses, i.e., made incorrect "Go" responses to Win cues.

- When re-analyzing the fMRI and EEG data with regressors based on these new models M8 and M9, results remained essentially unchanged and did not change the conclusions.

We conclude in Supplementary Note 8 in lines 558–569:

“In sum, the three additional models provided a better quantitative fit to the data compared to the winning model M5 reported in the main text. Also, these additional models predicted the propensity more accurately than the base models did. However, their qualitative fit (i.e. the ability to capture relevant aspects of the data) was worse: These additional models systematically underestimated the proportion of incorrect Go responses. Furthermore, although the predicted patterns of the propensity to stay matched the data more closely than M5, these predicted patterns still mis-matched some aspects of the data, particularly now over-estimating the tendency to stay following a punishment. Taken together, these models could capture certain qualitative patterns in the data, but not others, which is a core feature of computational modelling, which by definition constitutes a data reduction procedure that necessarily loses some details of the data. In terms of qualitative model validation/falsification (Nassar and Frank 2016; Palminteri et al. 2017), M5 and M8/M9 capture different qualitative features of the data, but no model captured all features well.”

We summarize these findings in the main text in lines 186–192:

“We explored three additional models featuring supplementary mechanisms to account for this behavioral pattern (Supplementary Note 8 and Supplementary Fig. 7). All these models fitted the data well and captured the propensity of staying better than M5; however, these models overestimated the proportion of incorrect Go responses. Model-based fMRI analyses based on these models led to results largely identical to those obtained with M5 (Supplementary Note 9 and Supplementary Fig. 8). We thus focused on M5, which relied on only a single mechanism (i.e., biased learning from rewarded Go and punishment NoGo actions.”

Given that M5 best captured the feature that is the process under study in this paper, and that we have used M5 in previous publications (Swart et al. 2017, 2018), we propose to keep results from M5 in the main text, but report additional fMRI results from M8 and M09 below.

In brief, fMRI results using parameter estimates from M8/ M9 to compute biased PEs fully replicated the regions identified with M5 and displayed in Fig. 3 of the main text. In addition, M9 (the neutral outcomes reinterpretation model) yielded additional correlates of biased PEs in larger regions of vmPFC and PCC. Although not the process under study in this paper, these results tentatively suggest that vmPFC and PCC keep traces of the cue valence up until the time of the outcome, putatively biasing the processing of neutral outcomes.

We added the following Supplementary Fig. 8:

Supplementary Figure 8. BOLD correlates of biased prediction errors as predicted by the asymmetric pathways model (M5), the cue valence-dependent perseveration model (M8) and the neutral outcomes reinterpretation model (M9). (A) Regions encoding both the standard PE term and the difference term to biased PEs (conjunction) as predicted from the asymmetric pathways model (M5) at different cluster-forming thresholds ($1 < z < 5$, color coding; opacity constant; replotted from Fig. 3C main text). Clusters significant at a threshold of $z > 3.1$ are surrounded by black edges. This is a version of Fig. 3C reprinted with a color scheme consistent with the other two panels. (B) Regions encoding both the standard PE term and the difference term to biased PEs (conjunction) as predicted from the cue valence-dependent perseveration model (M8) at different cluster-forming thresholds ($1 < z < 5$, color coding; opacity constant). Clusters significant at a threshold of $z > 3.1$ are surrounded by black edges. In line with correlates of biased PEs as predicted by M5, BOLD signal in bilateral striatum, dACC (small-volume corrected), pgACC, PCC, left motor cortex, left inferior temporal gyrus, and primary visual cortex was significantly better explained by biased learning than by standard learning. This finding was not surprising given that adding perseveration to the model did not change the learning mechanism, but only led to slightly different best fitting parameter values. (C) Regions encoding both the standard PE term and the difference term to biased PEs (conjunction) as predicted from the neutral outcomes reinterpretation model (M9). In addition to the regions in which BOLD signal was significantly better explained by biased than standard PEs as derived from M5 and M8, biased PEs derived from M9 also explained BOLD signal in vmPFC (larger cluster than M5), PCC (larger cluster than M5), left inferior frontal gyrus and multiple clusters in superior and inferior lateral occipital cortex significantly better than standard PEs. These results tentatively suggested that vmPFC, PCC, and these other occipital regions might implement an additional mechanism besides biased learning which encodes the cue valence also at the time of the outcome, biasing the processing of neutral outcomes.

3-Relation between the vmPFC and striatum: I can see that using both the EEG and fMRI to carefully tease apart temporal and spatial neural signatures of motivational biases opens up lots of interesting questions. But for the current focus on the vmPFC and its relationship with the striatum (e.g. the vmPFC “influences” the striatum), I don’t see how the authors, from their results can make the claim that the vmPFC mediates what is happening in the striatum, aside showing that it arises earlier. Since they do not use causal manipulation or directional methods, the data cannot support any conclusion on the direction between the vmPFC and Striatum. While the authors acknowledge it in the discussion, the title would have to be rephrase.

We agree with the reviewer, and have changed the title to:

“Prefrontal signals precede striatal signals for biased credit assignment to (in)actions”

We also went through the manuscript to carefully rephrase any implied causal/ directional claims into claims about temporal precedence of PE encoding.

Typos and other minor details:

- line 55: One the one hand

Thanks for spotting this typo, we corrected it.

- **Caption Figure 2: orange and red bits (green and light green) on panel B are not distinguishable in the legend**

We now changed the colors in the figure and legend to make them more distinguishable (and color-blind friendly; in response to a comment by reviewer #1).

- **line 179: what is the reference to S04 presenting here? Must be a typo**

Thank you for spotting this typo. This reference was to S07 instead of S04, we changed it (after restructuring this is now Supplementary Note 7 and Supplementary Fig. 6).

%%
Reviewer #3 (Remarks to the Author):

The authors present an empirical study investigating the neural mechanism of motivational bias in reinforcement learning. The authors use simultaneous EEG and fMRI to track the order of the encoding of biased prediction error, and use reinforcement learning models to characterize the motivational bias in learning. They provide evidence of biased prediction error in a set of striatal and cortical regions using fMRI, as well as in midfrontal power using EEG. They link both fMRI and EEG data and identify unique contributions of fMRI BOLD in anterior and posterior cingulate, striatum to EEG power in midfrontal electrodes, and argue that cortical influences precede striatal activity in driving the motivational learning bias.

Overall, the methods in this paper are sound, building on a well-validated task and associated models, and the authors have performed an extensive number of careful analyses. Generally speaking, the use of model-based analyses to examine simultaneous EEG-fMRI data is rare and exciting, but in the case of this paper that novelty is diminished somewhat by the fact that the authors have already published another paper doing something very similar within the exact same dataset (albeit examining largely orthogonal elements of the task). Having said that, I think the findings presented in this paper could be of interest to researchers studying motivation, learning, and decision making, including the demonstration that a model accounting for a Pavlovian bias provides a good account of both fMRI and EEG signatures of RL updating; and the careful unraveling of the time-course of motivationally biased stimulus evaluation, response preparation and execution, feedback processing, and policy updating. Questions of novelty aside, my main concerns, unpacked below, pertain to the overarching interpretation of these findings as reflecting the prefrontal biasing of striatal learning, and to their localization and interpretation of specific prefrontal signals putatively related to policy updating.

We thank the reviewer for their very positive evaluation of our work, and we believe we have addressed all points raised below.

Major comments:

1. Overall, the evidence supporting the conclusion that the biased prediction error occurs in cortical regions before subcortical regions seems rather tenuous. In Figure 5, the authors tried to convey that the EEG correlates of PFC precede the correlates of striatum, so that cortical regions encode biased prediction error earlier than subcortical regions. However, the EEG correlates of ACC are mostly in the lower alpha band, which does not correlate with the magnitude of PE_bias. Moreover, based on Figure 5D, it seems that correlates of PCC increase at about the same time with striatum. All of these findings might indicate that ACC and PCC are involved in the outcome process earlier than striatum, not necessarily that these two areas encode the bias signal earlier than striatum.

We agree with the reviewer's conclusion "ACC and PCC are involved in the outcome process earlier than striatum"—which we believe is in itself highly interesting, as it does not follow from established

models of reinforcement learning, and on top of this, is strongly suggestive of an important role for the prefrontal cortex in biased learning.

If we understand correctly, the reviewer's concern is that there might be changes over time in the type of outcome processing, with a certain region showing a different kind of processing early vs. late in the trial. In particular, our conclusions would be incorrect if a) cortical regions (ACC, PCC) would show a type of "unspecific" outcome processing early in the trial (visible in EEG over the scalp), b) the striatum were the first region to encode biased PEs, and c) biased PE would get propagated from the striatum to cortical regions late in the trial (shaping the BOLD signal and leading to biased PE correlates in the fMRI results).

We believe this possibility to be incompatible with what is known about feedback processing in the brain and with our data: Typically, neural outcome processing finishes within less than a second (Schultz, 2016). Responses to outcomes in single OFC cells occur largely in the first 500 ms after outcome onset (e.g. Rolls et al., 1996, Fig. 2; Morrison & Salzman, 2009, Fig. 6). Assuming that EEG correlates of the striatum truly reflect when the striatum encodes biased PEs (i.e., around 500–700 ms; in line with single cell recordings reviewed in Schultz, 2016), cortical processes that occur even later would appear "epiphenomenal" and unlikely to have any causal role in learning/ credit assignment. However, several lesion studies support a causal role of the PFC in learning/ credit assignment (Takahashi et al., 2009; Walton et al., 2010; Noonan et al., 2010). Hence, given previous literature, it is unlikely that early EEG correlates of ACC/ PCC reflect "unspecific" outcome processing that has no causal role in the particular kind of credit assignment and only becomes "biased" later (through striatal inputs). Instead, given previous literature, it is more plausible that prefrontal outcome processing is a fast and rather unitary process within the first 500 ms that is already "biased" in the sense of biased PEs predicted by our winning computational model (which explains why trial-by-trial BOLD in these regions is better explained by biased than by standard PEs) rather than falling apart into temporally distinct unbiased and biased processing stages.

We now reply to the separate points raised by the reviewer in more detail:

First, the reviewer is correct that PE_{BIAS} was only significantly encoded in delta power, but not in beta or lower alpha power. This finding is not surprising, but in line with a large body of previous literature finding absolute PE ("saliency", "surprise") signals in delta power (Bernat et al. 2011, 2015; Talmi et al. 2013; Cavanagh 2015; Cavanagh et al. 2021). In general, the literature on EEG (time-frequency) correlates of PEs is rather sparse compared to the fMRI literature showing a large number of regions encoding PEs. We obtained the conflicting finding that BOLD signal in the dACC reflects biased PEs (Fig. 3C), but its maximal EEG correlate (midfrontal alpha power) does not (which is not surprising given previous literature). The same holds for striatal BOLD and its correlate in beta power. In our view, the requirement that both a) the BOLD signal in a given region and b) the EEG correlate of this BOLD signal must encode a certain signal in order to conclude that the region encodes the signal (a type of cross-modal conjunction test) seems very conservative, especially compared to other papers that only use fMRI, and prone to false negative findings.

Our discussion mostly focuses on the *regions* likely involved in biased updating (based on the fMRI results) while we remain tacit on the putative roles of different frequency bands (apart from highlighting parallels to findings in previous literature). We have now carefully checked that our discussion does not make any claims about EEG correlates of biased PEs that are not substantiated by our results.

Second, regarding the reviewer's concern that PCC and striatal EEG correlations start being non-zero around the same time, it is important to keep in mind that power cannot be estimated instantaneously, but only over a temporally extended time windows (here 400 ms, which gives 2.5 Hz frequency resolution; the use of Hanning tapers, which have a Gaussian shape, ensures that the power

estimate is dominated by the center of the time window). Given this estimation over a temporally extended time window, any measure relying on a “*latency of first emergence*” or “*increase*” is likely suboptimal for inferring timing relations in time-frequency data (unlike time-domain or spike data). In contrast, the *timing of the peak* of such correlations can be estimated more robustly (i.e., is less influenced by any outlier) and is likely more meaningful.

We also explicitly discuss this choice now in the Results section in lines 343–347:

“This temporal dissociation was especially clear in the time courses of the test statistics for each region (thresholded at $|t| > 2$ and summed across frequencies), for which the peaks of the cortical regions preceded the peak of the striatum (Fig. 5D, H). Note that time-frequency power is estimated over temporally extended windows (400 ms in our case), which renders any interpretation of the “onset” or “offset” of such correlations more difficult.”

Third, the reviewer seems concerned that PCC and ACC might be involved in a form of “unspecific” outcome processing that does not necessarily include biased PE updating. We would like to highlight again that, based on our fMRI results (Fig. 3C), we have selected regions that show a very specific kind of outcome processing: We have identified biased learning in behavior, and then identified regions which encode the signal that would give rise to such biased learning—e.g., striatum, vmPFC, ACC, and PCC. In these regions, trial-by-trial BOLD signal is significantly better explained by biased (compared to standard) learning updates (shown in Fig. 3C for regions surviving a strict conjunction test). We have then identified the EEG correlates of these regions, suggestive of *when* they contribute to biased learning.

We now lay out our reasoning in terms the relative timing biased learning signals in cortical (vmPFC, ACC, PCC) and subcortical (striatum) regions in the Discussion section in lines 438–451:

“The observation that both PFC and striatal BOLD signal reflected biased PEs might be explained by three different models. One model assumes that both PFC and striatal processes arrive at biased learning independently of each other, which is highly unlikely given strong recurrent connections between both regions (Frank, 2005; Collins & Frank, 2014; Haber, 2003). Another model incorporates such interconnections, but assumes that striatum leads the PFC. While such a model is in line with past animal studies (Pasupathy & Miller, 2005) and modeling work (Wang et al., 2018), it would predict EEG correlates of the PFC to trail after EEG correlates of the striatum—or at least to occur with considerable delay after outcome onset. This model is not supported by our findings, which showed EEG correlates of PFC regions soon after outcome onset, preceding striatal EEG correlates. These early EEG correlates of PFC BOLD are in line with single cell recordings in PFC showing responses confined to the first 500 ms following outcome onset (Rolls et al., 1996; Morrison & Salzman, 2009), corroborating that PFC outcome processing occurs before the time of EEG correlates of striatal BOLD. The only model consistent with our data assumes recurrent connections between PFC and striatum, but with the PFC leading the striatum. Hence, these results are in line with a model of PFC biasing striatal outcome processing, giving rise to motivational learning biases in behavior.”

2. Related to the previous point, how much of the difference in latency of EEG signals between striatal and cortical regions can be accounted for by their relative proximities to the recordings at the scalp? I’m wondering in particular as pertains to the regressions that regress EEG activity on activity in cortical and subcortical ROIs. It would be helpful if the authors can provide some information to rule this out either a priori or with relevant analyses (or otherwise adjust their interpretation accordingly).

Electric fields originating from both cortical and subcortical regions are present at the scalp instantaneously. Hence, the relative proximities to different regions to the scalp are unlikely to lead to any lag in the scalp signal. This is backed up by empirical findings: Previous work that simultaneously

recorded intracranial striatal activity and scalp EEG (Cohen et al. 2009) found maximal correlations between both signals at a time lag of 0 ms. Similarly, studies correlating intracranial hippocampal signal with scalp EEG or MEG signal have found maximal correlations at a lag of zero (Attal and Schwartz 2013; Dalal et al. 2013; Pizzo et al. 2019; Alberto et al. 2021).

However, a different question is whether the beta power modulations measured in EEG are generated by the striatum (given that striatal signals might be too weak at the scalp) or another region that acts as an “antenna” for striatal signals. It is possible that the striatum drives signals in more superficial, cortical regions, whose activity becomes visible in beta power modulations over the scalp. Such an “entrainment” of a cortical region by the striatum might take time—and it is unclear how long. However, even a lag up to 100-200 ms would not change our conclusions.

We discuss the possibility of more beta power being generated by a “cortical antenna” closed linked to the striatum in the Results section in lines 325–331:

“Given that the striatum is far away from the scalp and thus unlikely to be the source of midfrontal beta power over the scalp, and given the assumption that trial-by-trial variation in an oscillatory signal should correlate with BOLD signal in its source (Debener et al. 2006; Huster et al. 2012), we speculate that dlPFC and SMG (identified in the EEG-informed fMRI analyses) are the sources of beta power over the scalp and act as an “antenna” for striatal signals. In line with this idea, previous studies have localized feedback-related beta power in lateral frontal and parietal regions, both using simultaneous EEG-fMRI (Sadaghiani et al. 2010; Andreou et al. 2017; Mas-Herrero et al., 2015) and source-localization (Sepe-Forrest et al., 2021; HajiHosseini & Holroyd, 2015).”

We now discuss the potential implications of a lag between EEG signals generated by cortical and subcortical regions in the Discussion section in lines 528–536:

“We observed EEG correlates of striatal BOLD at a rather late time point after outcome onset. While we conclude that biased outcome processing occurs much earlier in cortical regions than the striatum, it is possible that the modulating influence of the striatum on cortical sources of beta synchronization over the scalp (possibly dlPFC and SMG; corroborating previous EEG-fMRI (Sadaghiani et al., 2010; Andreou et al., 2017; Mas-Herrero et al., 2015) and source-reconstruction studies (Sepe-Forrest et al., 2021; HajiHosseini & Holroyd, 2015)) takes time to surface. However, speaking against any delay, some single studies have reported maximal correlations between striatal LFPs and scalp EEG at a time lag of 0 (Cohen et al. 2009). Regardless, even in the presence of a non-zero lag, our main conclusion would hold: Biased learning is present in cortical regions early after outcome onset, which cannot be a consequence of striatal input, but must constitute an independent origin of motivational learning biases.”

3. The claim that ACC maintains the eligibility trace of the previously performed response (p. 16), would benefit from a formal test of this hypothesis, showing that the eligibility trace from the fitted RL model provides an adequate account of activity in this region. The same is true for the proposed link between vmPFC/PCC and action policy updating – a stronger test would be to explicitly link the neural activity to individual-specific changes in the likelihood of one action or another.

Apology for the confusion about the term “eligibility trace”—we did not intend to refer to eligibility traces in multi-step reinforcement learning algorithms (e.g., the SARSA algorithm commonly applied to the two-step task). Given the nature of our task with only a single state, our Q-learning model does not need to employ such a computational eligibility trace. What we instead meant was simple memory maintenance retaining what response participants had just made and should attribute the outcome to. We have now updated the term “eligibility trace” to “memory trace” throughout the manuscript to avoid confusion.

Regarding the link between vmPFC/PCC and action policy updating, we are not sure what the reviewer means with “explicitly link the neural activity to individual-specific changes in the likelihood of one action or another”. We believe that we have exactly done that in our analyses: We first estimated the trial-by-trial HRF amplitude in selected ROIs, and then use these amplitudes to predict participants’ responses—stay or shift, which is the manifest index of latent changes in response likelihood—on the next trial with the same cue.

4. Related to the previous point, to what extent can outcome-related responses to positive or negative feedback reflect changes in policy certainty? For instance, the authors found that midfrontal theta was associated with negative feedback and increased likelihood of switching policies. Could this reflect an association between theta and policy uncertainty (cf. Cavanagh & Frank, 2015, Trends Cog Sci) rather than policy updating per se.

We agree with the reviewer that outcome-related theta might reflect changes in policy uncertainty, a view on the role of theta that unifies increases in theta in the context of response conflicts, errors, surprises, and negative feedback (Cavanagh, Figueroa, et al. 2012; Cavanagh, Zambrano-Vazquez, et al. 2012; Cavanagh and Frank 2014). However, in the context of our task, negative feedback necessarily increases uncertainty—assuming that participants usually act in accordance with the value beliefs, negative feedback indicates that these beliefs might be incorrect and thus increases uncertainty. Vice versa, positive feedback confirms their value beliefs and reduces uncertainty. Cases in which participants do not act in accordance with their value beliefs remain hidden to us and cannot be addressed separately. It is thus not possible within our task to disentangle outcome valence and increases/ decreases in policy certainty.

We now acknowledge this in the Discussion section in lines 506–511:

“[...] This observation is in line with previous literature suggesting that midfrontal theta and delta power might reflect the “saliency” or “surprise” aspect of PEs (Cavanagh, 2015; Talmi et al., 2013; Hauser et al., 2014). Surprises have the potential to disrupt an ongoing action policy (Wessel & Aron, 2017) and motivate a shift to another policy, which might explain why these signals predicted switching to another response (Trudel et al., 2020; Domenech et al., 2020). Notably, this EEG surprise signal was only significantly correlated with the biased (but not the standard) PE term, corroborating that the surprise attributed to outcomes depends on the previously performed response in line with motivational learning biases.”

5. The authors initially refer to the vmPFC region as vmPFC/perigenual ACC (32d), and to the ACC region as ventral ACC (23/24), but transition to referring to these as vmPFC and ACC. To avoid confusion with other regions of PFC, I would recommend continuing to refer to these in ways that reference their common connections to regions of cingulate. This is relevant to their finding that vmPFC appeared to track theta activity, which past literature has linked to regions of the cingulate. It is also relevant context for discussion of evolutionary age (e.g., p. 19), which is still germane for cingulate and striatum, but with a different connotation than if vmPFC referred to a region like BA10.

We agree with the reviewer that our nomenclature was not entirely consistent. We now updated the terminology for these clusters based on the Neubert Cingulate Atlas (Neubert et al., 2015). Regarding the two distinct subregions in ACC encoding biased prediction errors (Fig. 3C), we changed the labeling to perigenual ACC (pgACC) and dorsal ACC (dACC) throughout the manuscript and figures. Please note that trial-by-trial BOLD signal in both the *pgACC* cluster (Fig. 3C) and the *vmPFC* cluster correlating with trial-by-trial theta power (Fig. 5F) correlates negatively with the probability to stay with a given response, which implies that both *pgACC* and *vmPFC* might encode similar signals.

We have clarified this in lines 405–411:

“[...] Participants were significantly more likely to *repeat* the same response when striatal BOLD was high ($b = 0.067$, $SE = 0.024$, $\chi^2(1) = 9.051$, $p = .003$), but more likely to *switch* to another response when vmPFC BOLD ($b = -0.065$, $SE = 0.020$, $\chi^2(1) = 8.765$, $p = .003$) or PCC BOLD ($b = -0.036$, $SE = 0.016$, $\chi^2(1) = 3.691$, $p = .030$; Fig. 5H) was high (Supplementary Fig. 20). Similarly, high pgACC BOLD predicted a higher likelihood of switching, likening it with the circuits formed by vmPFC and PCC ($b = -0.076$, $SE = 0.017$, $\chi^2(1) = 15.559$, $p < .001$).”

When we refer to the vmPFC in the Discussion section, we mean the cluster identified with the correlation with trial-by-trial theta power through EEG-informed fMRI analyses as displayed in Fig. 5F, which should now be obvious following the relabeling.

6. The EEG-fMRI analyses are overall quite dense and difficult to follow, and require additional clarification. Perhaps a figure could be useful for visualizing the rationale and/or predictions for the EEG-informed fMRI and fMRI-informed EEG analyses. A few specific questions about these analyses:

This is a great suggestion. We now added Supplementary Figures 15 and 16 in which we visualize both the EEG-informed fMRI analyses and fMRI-informed EEG analyses.

a. p. 12. “biased PEs were controlled to capture additional variance in EEG explained by BOLD signal beyond task regressors”. Can the authors clarify why the unique variance explained by brain activity after controlling for biased PE is still informative regarding the order of biased PE processing?

If BOLD signal in a given region and a given EEG signal reflect the same behavioral variable, a spurious correlation can arise between them. For example, BOLD signal in the cerebellum and midfrontal theta both reflect Go/ NoGo responses; hence, using theta as a regressor to predict BOLD will yield correlations with cerebellar BOLD, even though it is very implausible that cerebellar activity would give rise to changes in EEG signal over the scalp (example chosen based on (Algermissen et al. 2022)). However, when controlling for the respective behavioral variable, such “spurious” correlations between EEG and BOLD signal should vanish. Hence, our strategy was to perform multiple linear regression controlling for a) all relevant behavioral variables and b) BOLD signal in other regions encoding the same behavioral variable. As a consequence, regression coefficients will reflect incremental unique variance that BOLD signal in a region explains beyond behavioral variables (such as biased PEs) and BOLD signal from any other candidate region. This is a much stronger test for the hypothesis that BOLD and EEG signal reflect the same neural phenomenon.

We now clarify this argument further in the manuscript:

In the Results section in lines 291–297:

“We performed analyses with and without PEs included in the model, which yielded identical results and suggested that EEG-fMRI correlations did not merely result from PE processing as a “common cause” driving signals in both modalities. Instead, EEG-fMRI correlations reflected incremental variance explained in EEG power that was afforded by the BOLD signal in selected regions (even beyond variance explained by the model-based PE estimates), providing the strongest test for the hypothesis that BOLD and EEG signal reflect the same neural phenomenon.”

In the Methods section in lines 901–905:

“All results were identical with and without including PEs into the model, suggesting that EEG-fMRI correlations did not merely arise from both modalities encoded PEs as a “common cause” that induced correlations. Instead, these correlations reflected the incremental variance explained in EEG power that was afforded by the BOLD signal even beyond the PEs.”

Given that the regressors in these analyses (trial-by-trial BOLD from various regions) were selected *based on the fact* that they encoded biased PEs, it is not surprising that these regressors already capture much of the variance to be explained and the addition of the PEs as covariates does not change the results.

b. P13-14. Why are vmPFC and PCC included in the ‘reverse check of the link’ analyses but the same is not being done for all of the EEG signatures in the opposite analysis (for example, ACC)? Additionally, it not clear that simply reversing the regression model would necessarily provide evidence to combine PCC and vmPFC in the follow-up analysis (e.g., fMRI ~ EEG does not necessarily provide the same interpretation as EEG ~ fMRI). The authors should provide a more thorough rationale for combining vmPFC and PCC based on this reverse check. Specifically, it is unclear why it is reasonable and necessary to perform the reverse check to include new areas.

Apologies for the confusion—the term “reverse check” might suggest a “backtranslation” of the analyses, when, in fact, these refer to *independent* and *complementary* analyses. Specifically, these analyses (reported in figures 5E-G) used trial-by-trial midfrontal EEG signal (Fig. 4A-C and Fig. 6A) to predict BOLD signal in a whole-brain GLM. We did not “include” vmPFC and PCC in these analyses, but *observed* them as results of these analyses, namely correlates of trial-by-trial midfrontal theta power (Fig. 5F). These analyses were not motivated by (and completely agnostic of) the EEG-fMRI findings presented in Figures 5A-C. Instead, they were motivated by the EEG-only findings presented in Fig. 4A-C (theta and beta) and Fig. 6A (alpha).

Furthermore, at no stage did we “combine” PCC and vmPFC. In the analyses relating trial-by-trial BOLD to action policy updating, we treat both as separate ROIs, but find that they relate to policy updating in the same way.

To further clarify (the independence of) our analysis procedures, we have added a new Supplementary Fig. 16, providing a visualization of the analysis process.

We now added in the Results section in lines 309–315:

“Complementary to the fMRI-informed EEG analyses, we also performed independent EEG-informed fMRI analyses, which showed the robustness of this EEG-BOLD correlation. We used the trial-by-trial EEG signal in the cluster identified in the EEG-only analyses (see Fig. 4 A, B) to predict BOLD signal across the brain (see Supplementary Fig. 16 for a graphical illustration of this approach). The EEG time-frequency-mask used to create the EEG regressor was defined based on the EEG-only analyses (Fig. 4A, B) and thus blind to the result of the fMRI-informed EEG analysis.”

and in lines 320–322:

“[...] Again, to support the robustness of this finding, we used trial-by-trial midfrontal beta power in the cluster identified in the EEG-only analyses (see Fig. 4A, C) to predict BOLD signal across the brain.”

c. The authors stated that ‘this analysis suggests dIPFC and SMG as likely candidate sources’. What is the rationale for this? The authors should provide additional evidence to support this assignment of candidate sources (either from the current dataset or existing literature), or simply remove this speculation from the text.

We added the following rationale for this speculation, and also make clearer that this is a speculation:

Results, lines 322–331:

“Clusters of positive EEG-BOLD correlations in right dorsal caudate (and left parahippocampal gyrus) as well as clusters of negative correlations in bilateral dorsolateral PFC (dIPFC) and supramarginal gyrus (SMG; Fig. 5G; Supplementary Table 7) confirmed the positive striatal BOLD-beta power

association. Given that the striatum is far away from the scalp and thus unlikely to be the source of midfrontal beta power over the scalp, and given the assumption that trial-by-trial variation in an oscillatory signal should correlate with BOLD signal in its source (Debener et al. 2006; Huster et al. 2012), we speculate that dlPFC and SMG (identified in the EEG-informed fMRI analyses) are the sources of beta power over the scalp and act as an “antenna” for striatal signals. In line with this idea, previous studies have localized feedback-related beta power in lateral frontal and parietal regions, both using simultaneous EEG-fMRI (Sadaghiani et al. 2010; Andreou et al. 2017; Mas-Herrero et al., 2015) and source-localization (Sepe-Forrest et al., 2021; HajiHosseini & Holroyd, 2015).”

Minor comments:

1. P. 11, “Only the sum of both terms...was significantly encoded...” - The authors should be cautious when interpreting this and related findings. The fact that PE_{STD} ($p=0.074$) and PE_{DIF} ($p=0.185$) were not significantly associated with delta power but PE_{BIAS} was ($p=0.017$) does not on its own imply that PE_{BIAS} provided a reliably better account of delta power than the other two. To infer otherwise would require a direct contrast of these (e.g., R-sq or BIC change).

We agree with the reviewer here. However, we do not believe that these effects have to be directly contrasted. Based on our behavioral model fitting and comparison, we have strong evidence that participants used the PE_{BIAS} term to update their belief. This term was significantly encoded in midfrontal (delta) power, which is the relevant finding for our subsequent fMRI-informed EEG analyses. The other two findings (PE_{STD} and PE_{DIF}) have no direct implications. However, given the p-values, it is clear that PE_{STD} did not provide a significantly better fit than PE_{BIAS} (but, based on comparing p-values, if anything, a worse fit). Given that these results are based on cluster-based permutation tests, it is difficult to compute an R^2 or BIC in this case to directly compare these findings.

We now reframed our results, de-emphasizing the findings of PE_{STD} and PE_{DIF} in lines 260–268:

“When testing for correlates of PE magnitude, we controlled for PE valence given that previous studies have reported TF correlates of both PE valence and PE magnitude in a similar time and frequency range, but with opposite signs (Talmi et al. 2013; Bernat et al. 2015; Cavanagh 2015). Midfrontal delta power was indeed positively correlated with the PE_{BIAS} term (225–475 ms; $p = .017$; Fig. 4D). Decomposition of the PE_{BIAS} term into its constituent terms showed that this correlation was not significant for the PE_{STD} term ($p = 0.074$, Fig. 4E) nor for the PE_{DIF} term ($p = 0.185$; Fig. 4F). This result does not imply that the PE_{BIAS} term explained delta power significantly better than the PE_{STD} term; it only implies significant encoding of the PE_{BIAS} term as suggested by the model that best fitted the behavioral data, with no significant evidence for a similar encoding of the conventional PE_{STD} term.”

2. P. 36. “these measures were created by using the 3-D (time-frequency-channel) t-map obtained when contrasting positive vs. negative outcomes (theta/ delta and beta) and Go vs. NoGo conditions (lower alpha band) as a linear filter.”-- please include details about this analysis (e.g., in the supplement).

We agree that more details of the analyses procedures would help clarify for the reader. For this reason, we created two new Supplementary Figures 15 and 16 to further illustrate this approach.

3. Please clarify how the hyperpriors in Model M5 were chosen.

We now clarify this in the Methods section in lines 677–680:

“The weakly informative hyperpriors were set to $X_\rho \sim \mathcal{N}(2,3)$, $X_\varepsilon \sim \mathcal{N}(0,2)$, $X_{b,\pi,\kappa} \sim \mathcal{N}(0,3)$, in line with previous implementations of this model (Swart et al., 2017; 2018). The same priors (for the same

parameters) were used across different model implementations to not bias model comparison. Alternative hyperprior specification did not change the results.”

4. It would be helpful to provide an intuition for why the presence of two epsilon values in Equation 6 ensured that the effect of kappa on epsilon was symmetrical.

In line with previous papers using this model (Swart et al. 2017, 2018), we aimed to ensure that the difference between a) $\epsilon_{rewarded\ Go}$ and the standard learning rate ϵ_0 and between b) ϵ_0 and $\epsilon_{punished\ NoGo}$ is of equal size *after* performing the sigmoid transform. Thua, we computed first $\epsilon_{rewarded\ Go}$ and ϵ_0 , computed the difference $\Delta\epsilon$ between them, and then subtracted $\Delta\epsilon$ from ϵ_0 to obtain $\epsilon_{punished\ NoGo}$. This procedure works as long as $\Delta\epsilon$ is numerically smaller than ϵ_0 . If, however, $\Delta\epsilon$ is numerically bigger than ϵ_0 , the value of $\epsilon_{punished\ NoGo}$ becomes negative, which is not meaningful for a learning rate parameter. For $\epsilon_0 > 0.5$, it is guaranteed that $\Delta\epsilon < \epsilon_0$ given that $\epsilon_{rewarded\ Go} < 1$ after the sigmoid transform. For $\epsilon_0 < 0.5$, it is however possible that $\Delta\epsilon > \epsilon_0$ and then $\epsilon_{punished\ NoGo} < 0$. Hence, for $\epsilon_0 < 0.5$, we used the opposite approach of first computing $\epsilon_{punished\ NoGo}$ and ϵ_0 , computing the difference $\Delta\epsilon$ between them, and then adding $\Delta\epsilon$ to ϵ_0 to obtain $\epsilon_{rewarded\ Go}$. For $\epsilon_0 < 0.5$, this procedure is guaranteed to yield $\Delta\epsilon < \epsilon_0$ and hence also $\epsilon_{rewarded\ Go} < 1$, which gives a meaningful learning rate parameter.

We now rewrote the equations to make their effects clearer in the Methods section, lines 679–680:

$$\epsilon = \begin{cases} \epsilon_0 = inv.logit(\epsilon) & \\ \epsilon_{punished\ NoGo} = inv.logit(\epsilon - \kappa) & \text{if } \epsilon_0 < .5 \\ \epsilon_{rewarded\ Go} = \epsilon_0 + (\epsilon_0 - \epsilon_{punished\ NoGo}) & \text{if } \epsilon_0 < .5 \end{cases} \quad (6)$$

$$\epsilon = \begin{cases} \epsilon_{rewarded\ Go} = inv.logit(\epsilon + \kappa) & \text{if } \epsilon_0 > .5 \\ \epsilon_{punished\ NoGo} = \epsilon_0 - (\epsilon_{rewarded\ Go} - \epsilon_0) & \text{if } \epsilon_0 > .5 \end{cases}$$

REVIEWER COMMENTS

Reviewer #1 (Remarks to the Author):

The authors have performed an extremely thorough, care- and thoughtful revision of their manuscript. Thank you for clarifying questions and misunderstandings I have had. Any remaining limitations are discussed with great care.

I strongly recommend publication of the manuscript in its present form.

Reviewer #2 (Remarks to the Author):

I think that the authors have adequately addressed the comments made by the reviewers in the revised version of the manuscript. The authors have performed convincing new analyses and substantially reviewed their narrative which has improved the manuscript a lot. I have no further comments.

Reviewer #3 (Remarks to the Author):

Algermissen and colleagues addressed many of my previous concerns about their neuroimaging findings, and now include more details about EEG-fMRI analysis as suggested. However, I still have some concerns and clarifications that I think bear addressing. Specifically, the justification for the decomposition of the PE bias term (as well as associated assumptions) are unclear and need further expansion. It is also unclear whether the results in Figure 5 are sufficient to support the author's claims about the striatum and beta frequency bands. These comments are detailed below.

Major comments

1. The explanation for why the decomposition of the PE_{bias} differentiates between the two learning signals is still not clear. Given the importance of this decomposition for all subsequent analyses, a detailed explanation of this procedure and key assumptions is essential. It would be helpful to elaborate more on how the standard PE (from Model 1) and biased PE (from Model 5) are substantively different and provide some understanding of the

range of the PE_diff (e.g., histogram or summary statistics) to aid in interpreting the conjunction analyses.

2. The authors provide a helpful justification in the response letter regarding why the difference PE leads to significantly more explained variance in the BOLD signal, which would be helpful to include in the main text. In addition to including this technical justification for how their approach captures the learning bias and key assumptions involved in this procedure, it would also help to include an intuitive explanation regarding how to interpret this decomposition in terms of its impact on neural correlates. The authors reference Daw et al. (2011), among others, but these papers used this decomposition (and specifically, partial derivatives) under the assumption that MB PEs were present over and above (i.e., in addition to) MF PEs. Here, the assumption seems to be that only biased PEs are present (in place of standard PEs), so there is at least a surface-level disconnect that may be worth noting and (if relevant) briefly unpacking when justifying the approach.

3. In addition to providing a more technical justification for how this approach captures the learning bias and key assumptions involved in this procedure, it would also help to include an intuitive explanation regarding how to interpret this decomposition in terms of its impact on neural correlates.

4. The authors have argued that the peaks in Figure 5D provide evidence for temporal differences between the dACC, PCC, and striatum correlations. However, these time courses appear to be agnostic to the frequency bands encompassed, which is problematic for interpretation of specific frequency bands. For example, in Figure 5C, the correlation with the striatum spans both beta and alpha waves – based on the time of the striatum peak in Figure 5D, it is likely that the peak represents the correlation in the alpha band rather than the beta band, which seems inconsistent with the description in the main text regarding midline beta power. Moreover, it is clear that there exists an early peak in the striatum correlation at 250 ms within the beta band. If this is an accurate read of the findings, the authors should modify their conclusion to indicate that the late striatum correlation is in the alpha band rather than the beta band, and update figures and discussions accordingly.

5. In their response to Q6a, the authors provided reasons for including PE in the model, but did not address why the remaining variance explained by fMRI data is still informative regarding PE. Since the purpose of this analysis is to infer the relative order of biased PE processing across cortical and subcortical regions, it would be more informative if the

authors could justify why the analysis controlling PE still provides evidence regarding the order of processing, e.g., why the observed earlier PE-controlled correlation between the cortical region and EEG signal reflect earlier biased PE processing, but not any other process that evolves in parallel. Some explanation in the main text would be helpful to clarify this.

Minor Comments

1. The authors provide a thorough explanation for the presence of two epsilon values in Equation 6 corresponding to M5 in the response letter, but this should be clarified in the manuscript as well. Given that M5 is the main model used throughout the model-based analysis, it would be helpful to explain this procedure for how the learning rate bias was updated. This will be especially helpful for readers as well as others who may want to adopt this model in future studies. Additionally, it is slightly confusing that the sigmoid is only included in the top but not bottom epsilon - it might be clearer to write out this updating procedure for epsilon in two separate steps.
2. For parameter recovery of M5, the authors report excellent parameter recovery for all free parameters except for learning bias k . The authors test additional models with a perseveration parameter, and still favor M5 because of its better fit to behavior. However, another possibility is that the learning rate bias may not be optimally implemented in the chosen model (e.g., the nonlinear transformations in M5 is significantly different than the additive function in M4, which may have added unnecessary complexity). While it is not necessary to go on a fishing expedition to test additional model variants, it would be helpful to include this limitation of the model fitting/parameter recovery in the discussion, and perhaps consider speculating about ways of improving the fit of the chosen computational model.
3. In Figure 6C, its not exactly clear that the left panel reflects Go vs NoGo actions, since the labels are "Left + Right."
4. Pg. 24 discussion, did the authors mean to say "notably dACC and PCC" precedes correlates of the striatum (rather than dACC and pgACC)?

Legend of reviewer comments, responses, and textual changes to the manuscript:

- Bold** = reviewer comment
- Normal = our response
- “normal” = citations from manuscript (if entire paragraphs of text)
- Blue = changes in manuscript
- Highlight = header of entirely new sections in the supplemental materials

%%%

Reviewer #3 (Remarks to the Author):

Algermissen and colleagues addressed many of my previous concerns about their neuroimaging findings, and now include more details about EEG-fMRI analysis as suggested. However, I still have some concerns and clarifications that I think bear addressing. Specifically, the justification for the decomposition of the PE bias term (as well as associated assumptions) are unclear and need further expansion. It is also unclear whether the results in Figure 5 are sufficient to support the author’s claims about the striatum and beta frequency bands. These comments are detailed below.

Major comments

1. The explanation for why the decomposition of the PE_bias differentiates between the two learning signals is still not clear. Given the importance of this decomposition for all subsequent analyses, a detailed explanation of this procedure and key assumptions is essential. It would be helpful to elaborate more on how the standard PE (from Model 1) and biased PE (from Model 5) are substantively different and provide some understanding of the range of the PE_diff (e.g., histogram or summary statistics) to aid in interpreting the conjunction analyses.

We now added two new figures (Supplementary Figures 9 and 10) to more fully explain this point. We reprint them below:

Supplementary Figure 9. Illustration of biased and standard learning for a representative example participant. (A) Prediction errors according to the standard Q-learning model M1 (PE_{STD} ; black dots) and according to the winning model M5 implementing biased learning (PE_{BIAS} ; colored dots). In M5, motivational biases partially arise through biased learning: Participants learn more readily that an action has caused a reward, and are reluctant to learn that inaction has led to a punishment. For each cue, the values of each of the three possible actions (GO_{LEFT} , GO_{RIGHT} , NoGo) are learnt independently, and prediction errors are calculated relative to the value of the chosen action. The learning bias acts such that the effective learning rate is increased when a reward follows any Go response, and decreased when a punishment follows a NoGo response (see equation 5 in the main manuscript). Hence, for Win cues, action values for Go responses (but not NoGo responses) are affected by the learning bias and approach the positive asymptote more quickly compared to standard learning, leading to faster decay of positive prediction errors. At the same time, negative outcomes remain surprising and elicit larger prediction

errors compared to standard learning. Hence, model predictions diverge for prediction errors after Go responses to Win cues, but not after NoGo responses to Win cues (colored dots are on top of black dots). Vice versa, for Avoid cues, action values for NoGo responses (but not Go responses) are affected by the learning bias and approach the negative asymptote more slowly compared to standard learning (with negative prediction errors remaining high) as participants are reluctant to take punishments after NoGo responses into account. At the same time, ignoring punishments leads to a faster approach of positive action values towards the positive asymptote (and a faster decay of positive prediction errors) compared to standard learning. Model predictions diverge for prediction errors after NoGo responses to Avoid cues, but not after Go responses to Avoid cues (colored dots are on top of black dots). (B) To assess evidence for biased learning despite this high multicollinearity, we decomposed PE_{BIAS} into PE_{STD} (black dots) plus a difference term $PE_{DIF} = PE_{BIAS} - PE_{STD}$ (colored dots). Note that PE_{DIF} is always zero after NoGo responses to Win cues and Go responses to Avoid cues as both M1 and M5 make identical predictions for these action values. In contrast, for Go responses to Win cues and NoGo responses to Avoid cues, the PE_{DIF} term is always negative because, in both cases, positive action values approach the positive asymptote more quickly (such that positive prediction errors decay more quickly) compared to standard learning, and negative action values approach the negatively asymptote more slowly (and thus negative prediction errors remain high) compared to standard learning.

Supplementary Figure 10. illustration of prediction error regressor decomposition for a representative example participant. (A) Prediction errors according to the standard Q-learning model M1 (PE_{STD} ; larger red dots) and according to the winning model M5 implementing biased learning with more learning from rewarded Go responses and less learning from punished NoGo responses (PE_{BIAS} ; smaller blue dots; blue dots with a red edge reflect trials on which both models make identical predictions). Both prediction error types have a highly similar profile. They key difference is an overall downward shift, with positive PE_{BIAS} approaching zero more quickly than positive PE_{STD} , while negative PE_{BIAS} remain more negative for NoGo compared to negative PE_{STD} . Note that, after trial 320, session 2 starts (vertical dashed line), featuring new cues. (B) The prediction errors from both models are highly correlated (mean across participants: $r = 0.99$, range 0.96–0.99), implicating that, when entered together into a multiple linear regression, both regressors would share most of their variance, which would be attributed to neither of them. (C) To assess evidence for biased learning despite this high multicollinearity, we decomposed PE_{BIAS} into PE_{STD} plus a difference term $PE_{DIF} = PE_{BIAS} - PE_{STD}$. PE_{STD} and PE_{DIF} show markedly different profiles, with PE_{DIF} being zero for trials on which both PE_{STD} and PE_{BIAS} make identical predictions, and being negative otherwise (reflecting the relatively faster decay of positive PE_{BIAS} and slower decay of negative PE_{BIAS}). (D) Both PE_{STD} and PE_{DIF} are much less correlated (mean across participants: $r = -0.02$, range -0.07–0.09), making it possible to enter them in the same multiple linear regression and test whether PE_{DIF} predicts variance in BOLD signal above and beyond PE_{STD} .

2. The authors provide a helpful justification in the response letter regarding why the difference PE leads to significantly more explained variance in the BOLD signal, which would be helpful to include in the main text. In addition to including this technical justification for how their approach captures the learning bias and key assumptions involved in this procedure, it would also help to include an intuitive explanation regarding how to interpret this decomposition in terms of its impact on neural correlates. The authors reference Daw et al. (2011), among others, but these papers used this

decomposition (and specifically, partial derivatives) under the assumption that MB PEs were present over and above (i.e., in addition to) MF PEs. Here, the assumption seems to be that only biased PEs are present (in place of standard PEs), so there is at least a surface-level disconnect that may be worth noting and (if relevant) briefly unpacking when justifying the approach.

We worry that the comparison to the Daw papers might confuse more than clarify for readers. Daw et al. rejected the null hypothesis of only MF PEs being present. Instead, they concluded that there would be a mixture of both MF and MB PEs (allowing for the possibility of 100% MB PEs). In our case, a significant correlation of the BOLD signal with the PE_{DIF} term implies that learning is not unbiased, but—at least to some degree, which varies across participants—biased, similar to the two other applications of this procedure (Wittmann et al, 2006; Eldar & Niv, 2015). Our winning model M5 does not make a point prediction of “100% bias” that would equate the “100% MB PEs” prediction by Daw et al.

We now further emphasize that the regressor decomposition is “just” a tool to orthogonalize the regressors (which is a very common procedure in fMRI analyses) while computing the “orthogonalized” regressor in a way that makes it easily interpretable.

We now write in the main text in the Results section, lines 224–235:

“We tested for biased prediction errors PE_{BIAS} by testing which regions significantly encoded the conjunction of both its components, i.e., the significant encoding of both PE_{STD} and PE_{DIF} . Dissociating two alternative learning signals by decomposing one into the other plus a difference term is an established procedure to disentangle the contributions of two highly correlated signals (Wittmann et al. 2008; Daw 2011). **It has an effect highly similar to orthogonalizing regressors (Mumford et al. 2015) while maintaining interpretability in that both regressors (PE_{STD} and PE_{DIF}) add up to the term of interest (PE_{BIAS}).** Significant encoding of both components (with the same sign) provides strong evidence for encoding of biased prediction errors PE_{BIAS} . **The PE_{DIF} term itself has no substantive neural interpretation; it is merely an implicit model comparison of a null model (PE_{STD}) against a full model (PE_{BIAS}).** Intuitively, for voxels for which both PE_{STD} and PE_{DIF} are significant, one can conclude that the BOLD signal correlates with the full biased prediction error term PE_{BIAS} , and that this correlation is significantly stronger than for the baseline prediction error term PE_{STD} .

3. In addition to providing a more technical justification for how this approach captures the learning bias and key assumptions involved in this procedure, it would also help to include an intuitive explanation regarding how to interpret this decomposition in terms of its impact on neural correlates.

Please see our responses to points #1 and #2; we hope these also cover an “intuitive” explanation.

4. The authors have argued that the peaks in Figure 5D provide evidence for temporal differences between the dACC, PCC, and striatum correlations. However, these time courses appear to be agnostic to the frequency bands encompassed, which is problematic for interpretation of specific frequency bands. For example, in Figure 5C, the correlation with the striatum spans both beta and alpha waves – based on the time of the striatum peak in Figure 5D, it is likely that the peak represents the correlation in the alpha band rather than the beta band, which seems inconsistent with the description in the main text regarding midline beta power. Moreover, it is clear that there exists an early peak in the striatum correlation at 250 ms within the beta band. If this is an accurate read of the findings, the authors should modify their conclusion to indicate that the late striatum correlation is in the alpha band rather than the beta band, and update figures and discussions accordingly.

When plotting Figure 5C while restricting the EEG data used to compute the striatal-BOLD-to-EEG correlation to the beta band (13–33 Hz), the late “bulk” of the correlation between striatal BOLD and EEG power still reaches its peak later than the correlations between PCC BOLD and EEG power:

Furthermore, please note that (a) our EEG-only results in Fig. 4 show only beta (but not alpha) power to reflect outcome valence, which is in line with a host of EEG literature which we cite (Marco-Pallarés et al. 2008, 2015; van de Vijver et al. 2011; Mas-Herrero et al. 2015); and (b) several previous studies with direct electrophysiological recordings from the striatum (Courtemanche et al. 2003; Feingold et al. 2015; Amemori et al. 2018, 2020); cited in the discussion) have found beta power, but not alpha power, modulated by outcomes. It seems appropriate to situate the correlations between striatal BOLD and beta power we observed in this literature, while there is no comparable literature for alpha power encoding outcomes.

Still, the fact that, in our data, these correlations extend into the alpha range seems noteworthy, and we now write the Results section in lines 325–327:

“This finding bore resemblance in time-frequency space to the cluster of positive PE valence encoding in beta power identified in the EEG-only analyses (Fig. 4A, C), **but extended into the alpha range**”

We also acknowledge in the Discussion section in lines 488–490:

“In line with previous research, striatal BOLD positively linked to midfrontal beta power (Sadaghiani et al. 2010; Andreou et al. 2017), which positively encoded PE valence (Marco-Pallarés et al. 2008, 2015; van de Vijver et al. 2011), **with correlations extending into alpha power.**”

5. In their response to Q6a, the authors provided reasons for including PE in the model, but did not address why the remaining variance explained by fMRI data is still informative regarding PE. Since the purpose of this analysis is to infer the relative order of biased PE processing across cortical and subcortical regions, it would be more informative if the authors could justify why the analysis controlling PE still provides evidence regarding the order of processing, e.g., why the observed earlier PE-controlled correlation between the cortical region and EEG signal reflect earlier biased PE processing, but not any other process that evolves in parallel. Some explanation in the main text would be helpful to clarify this.

We now write in the Discussion section in lines 544–557:

“**In order to make inferences about the relative order of PE processing in different brain regions, we must assume that the regressor in our EEG-fMRI analysis approach—the trial-by-trial BOLD amplitude in selected regions—mostly reflects the PE signal rather than learning-unrelated processes occurring in parallel. In support of this assumption, animal electrophysiological recordings have indeed found**

that neural activity in ACC, PCC, and striatum is dominated by reward processing during outcome receipt (Matsumoto et al. 2007; Seo and Lee 2007; Hayden et al. 2008; Mohebi et al. 2019; Hamid et al. 2021) and meta-analyses on human BOLD signal have found strong effect sizes for PE processing in these regions (Bartra et al. 2013; Fouragnan et al. 2018). Importantly, we observe transient EEG-fMRI correlations that are likely event-related rather than reflecting resting-state-like correlations. We thus favor the conclusion that the observed EEG-fMRI correlations reflect differences in the timing of PE processing in these regions, although we cannot fully exclude the possibility that parallel processes unrelated to (biased) learning contribute to these correlations. Note that, while outcome processing in these regions is better described by biased than by standard PEs, each region might encode PEs in an idiosyncratic way (potentially reflecting noise in the value representations (Findling et al. 2019)) and these residual idiosyncrasies drive the EEG-fMRI correlations even when controlling for biased PEs predicted by the winning computational model.”

Minor Comments

1. The authors provide a thorough explanation for the presence of two epsilon values in Equation 6 corresponding to M5 in the response letter, but this should be clarified in the manuscript as well. Given that M5 is the main model used throughout the model-based analysis, it would be helpful to explain this procedure for how the learning rate bias was updated. This will be especially helpful for readers as well as others who may want to adopt this model in future studies. Additionally, it is slightly confusing that the sigmoid is only included in the top but not bottom epsilon - it might be clearer to write out this updating procedure for epsilon in two separate steps.

Thanks for this suggestion; we now updated equation 6 and split it into two equations (6 and 7) which both include the sigmoid. We also moved the relevant section up to explain that equation 5 is merely “conceptual”, while equations 6 and 7 are the technical implementation of it.

We now write in the methods section in lines 705–719:

Conceptually, learning rates differed between response-outcome conditions in the following way:

$$\varepsilon = \begin{cases} \varepsilon_0 + \kappa & \text{if } r_t = 1 \text{ and } a = \text{go} \\ \varepsilon_0 - \kappa & \text{if } r_t = -1 \text{ and } a = \text{nogo} \\ \varepsilon_0 & \text{else} \end{cases} \quad (5)$$

In the technical implementation of this model, learning rates were sampled in continuous space and then inverse-logit transformed to constrain them to the range [0 1](Swart et al. 2017, 2018). However, after this transformation, the impact of adding vs. subtracting the learning bias κ would no longer be symmetric. Hence, for baseline learning rates $\varepsilon_0 < 0.5$, we first computed the difference between the baseline learning rate and the learning rates for punished NoGo responses and used this difference to compute the learning rate for rewarded Go responses:

$$\varepsilon = \begin{cases} \varepsilon_0 = \text{inv. logit}(\varepsilon) \\ \varepsilon_{\text{punished NoGo}} = \text{inv. logit}(\varepsilon - \kappa) & \text{if } \varepsilon_0 < 0.5 \\ \varepsilon_{\text{rewarded Go}} = \varepsilon_0 + (\varepsilon_0 - \varepsilon_{\text{punished NoGo}}) & \text{if } \varepsilon_0 < 0.5 \end{cases} \quad (6)$$

Notably, this procedure is only guaranteed to work when $\varepsilon_0 < 0.5$. For $\varepsilon_0 > 0.5$, the difference term could become > 0.5 and the learning rate for rewarded Go responses would become > 1 , which is impractical. Hence, for $\varepsilon_0 > 0.5$, we first computed the learning rate for rewarded Go responses and used the difference to the baseline learning rate ε_0 to compute the learning rate for punished NoGo responses:

$$\varepsilon = \begin{cases} \varepsilon_0 = \text{inv. logit}(\varepsilon) \\ \varepsilon_{\text{rewarded Go}} = \text{inv. logit}(\varepsilon + \kappa) & \text{if } \varepsilon_0 > 0.5 \\ \varepsilon_{\text{punished NoGo}} = \varepsilon_0 - (\varepsilon_{\text{rewarded Go}} - \varepsilon_0) & \text{if } \varepsilon_0 > 0.5 \end{cases} \quad (7)$$

2. For parameter recovery of M5, the authors report excellent parameter recovery for all free parameters except for learning bias k . The authors test additional models with a perseveration parameter, and still favor M5 because of its better fit to behavior. However, another possibility is that the learning rate bias may not be optimally implemented in the chosen model (e.g., the nonlinear transformations in M5 is significantly different than the additive function in M4, which may have added unnecessary complexity). While it is not necessary to go on a fishing expedition to test additional model variants, it would be helpful to include this limitation of the model fitting/parameter recovery in the discussion, and perhaps consider speculating about ways of improving the fit of the chosen computational model.

We would like to highlight that the implementation of the learning rate bias in M4 and M5 is identical (the learning bias is introduced for M4 on line 707) and consistent to past papers using this model (Swart et al. 2017, 2018); we have now clarified this (see response to last comment).

We now write in the discussion section in lines 563–569:

“Furthermore, while parameter recovery for most parameters in the winning computational model (including the effective learning rates incorporating the learning bias) was excellent, parameter recovery for the learning bias term itself was positive, but weaker (see Supplementary Note 6). Supplementary models tested incorporating a perseveration parameter (see Supplementary Note 8) yielded higher model recovery, but failed to capture crucial aspects of the biased learning under investigation. Future studies comprising larger samples of participants should explore alternative implementations to reliably quantify individual differences in these learning biases.”

3. In Figure 6C, its not exactly clear that the left panel reflects Go vs NoGo actions, since the labels are “Left + Right.”

We now changed the labels to “ $GO_{LEFT} + GO_{RIGHT}$ ” and “ $GO_{LEFT} - GO_{RIGHT}$ ”.

Also, in the figure caption, we now explain that the “ $GO_{LEFT} + GO_{RIGHT}$ ” contrast reflects “activation by either left or right Go actions compared to the implicit baseline in the GLM, which contains the NoGo actions”.

4. Pg. 24 discussion, did the authors mean to say “notably dACC and PCC” precedes correlates of the striatum (rather than dACC and pgACC)?

Thanks for spotting, yes, we meant PCC instead of pgACC, this is corrected now.

REVIEWERS' COMMENTS

Reviewer #3 (Remarks to the Author):

The authors have done an excellent addressing my comments, and I am happy to recommend this for publication. Congratulations!